



# Twenty years of ground-based NDACC FTIR spectrometry at Izaña Observatory - overview and long-term comparison to other techniques

Omaira E. García[1], Matthias Schneider[2], Eliezer Sepúlveda[1], Frank Hase[2], Thomas Blumenstock[2], Emilio Cuevas[1], Ramón Ramos[1], Jochen Gross[2], Sabine Barthlott[2], Amelie N. Röhling[2], Esther Sanromá[1,a], Yenny González[1,3], Ángel J. Gómez-Peláez[1,b], Mónica Navarro-Comas[4], Olga Puentedura[4], Margarita Yela[4], Alberto Redondas[1], Virgilio Carreño[1], Sergio F. León-Luis[1], Enrique Reyes[1], Rosa D. García[1,5], Pedro P. Rivas[1], Pedro M. Romero-Campos[1], Carlos Torres[1], Natalia Prats[1], Miguel Hernández[6], and César López[7]

[1]Izaña Atmospheric Research Centre (IARC), State Meteorological Agency of Spain (AEMet), Santa Cruz de Tenerife, Spain.
[2]Karlsruhe Institute of Technology (KIT), Institute of Meteorology and Climate Research (IMK-ASF), Karlsruhe, Germany.
[3]Cimel Electronique, Paris, France.
[4]Atmospheric Research and Instrumentation Branch, National Institute for Aerospace Technology (INTA), Madrid, Spain.
[5]Group of Atmospheric Optic, University of Valladolid, Valladolid, Spain.
[6]Canarias Delegation, State Meteorological Agency of Spain (AEMet), Santa Cruz de Tenerife, Spain.
[7]Sieltec Canarias S.L., La Laguna, Spain.
[a]Now at: Employment Observatory of the Canary Islands (OBECAN), Santa Cruz de Tenerife, Spain.
[b]Now at: Asturias Delegation, State Meteorological Agency of Spain (AEMet), Oviedo, Spain.

**Correspondence:** ogarciar@aemet.es

**Abstract.** High-resolution Fourier Transform InfraRed (FTIR) solar observations are particularly relevant for climate studies, as they allow atmospheric gaseous composition and multiple climate processes to be monitored in detail. In this context, the present paper provides an overview of 20 years of FTIR measurements taken in the framework of the NDACC (Network for the Detection of Atmospheric Composition Change) from 1999 to 2018 at the subtropical Izaña Observatory (IZO, Spain). Firstly,

long-term instrumental performance is comprehensively assessed, corroborating the temporal stability and reliable instrumental characterisation of the two FTIR spectrometers installed at IZO since 1999. Then, the time series of all trace gases contributing to NDACC at IZO are presented (i.e. $C_2H_6$, $CH_4$, $ClONO_2$, CO, HCl, HCN, $H_2CO$, HF, $HNO_3$, $N_2O$, $NO_2$, NO, $O_3$, OCS, and water vapour isotopologues $H_2^{16}O$, $H_2^{18}O$, and $HD^{16}O$), reviewing the major accomplishments drawn from these observations. In order to examine the quality and long-term consistency of the IZO FTIR observations, a comparison of those NDACC

products for which other high-quality measurement techniques are available at IZO has been performed (i.e. $CH_4$, CO, $H_2O$, $NO_2$, $N_2O$, and $O_3$). This quality assessment was carried out on different timescales to examine what temporal signals, and to what extent, are captured by the FTIR records. After 20 years of operation, the IZO NDACC FTIR observations have been found to be very consistent and reliable over time, demonstrating great potential for climate research. Long-term NDACC FTIR data sets, such as IZO, are indispensable tools for the investigation of atmospheric composition trends, multi-year phenomena

and complex climate feedback processes, as well as for the validation of past and present space-based missions and chemistry climate models.



# 1 Introduction

The recognition that changes in the composition of the Earth's atmosphere are occurring, on both long and short timescales and thereby modifying our environment and climate, has resulted in scientific debate, as well as public concern in the last

decades (Gottwald et al., 2006). Established examples, such as depletion of ozone layer, warming of air and oceans, rising sea level or melting cryosphere, have widely been reported in literature (Stocker et al., 2013; WMO, 2014a; Masson-Delmotte et al., 2018; WMO, 2018, and references therein). In order to assess the significance of such changes and to better understand the physical and chemical processes involved, continuous, consistent, long-term monitoring of the atmospheric composition is indispensable. These observational data sets are also fundamental to testing the ability of current climate models to provide

reliable projections of future climate, and thus, they are the basis for design and implementation of efficient climate-change mitigation and adaptation policies.

Among different atmospheric monitoring measurement techniques, Fourier Transform InfraRed (FTIR) spectrometry is of particular interest for climate research. With this technique, the source radiation (typically the sun for atmospheric ground-based measurements) is modulated by an interferometer and all optical frequencies are recorded simultaneously in the measured

interferogram (Griffiths and de Haseth, 2007). Then, a mathematical Fourier transform is used to retrieve the atmospheric absorption spectrum from the interferogram. By analysing the pressure broadening effect on these measured solar spectra through inversion schemes, the FTIR technique can provide atmospheric concentrations of many different trace gases simultaneously and with a high degree of precision (e.g. Rinsland et al., 1982, 1998; Hase et al., 2004; Schneider et al., 2005, 2012; Kohlhepp et al., 2012; García et al., 2012b; Sepúlveda et al., 2014; Barthlott et al., 2015; Vigouroux et al., 2015; Wunch et al., 2015;

Vigouroux et al., 2018; De Mazière et al., 2018).

The first continuous or semi-continuous records of ground-based FTIR spectrometers started in the late 1970s and early 1980s in just a few stations around the world. Nowadays, high-resolution FTIR instruments mainly operate at a global scale in the framework of two international networks for atmospheric composition monitoring: NDACC (Network for the Detection of Atmospheric Composition Change, https://www.ndaccdemo.org) and TCCON (Total Carbon Column Observing Network,

https://tccon-wiki.caltech.edu). While NDACC aims mainly to establish a long-term database to detect changes and trends in atmospheric composition and to understand their impact on the Earth's atmosphere (De Mazière et al., 2018), TCCON focuses more on research on greenhouse gases, improving our understanding of the carbon cycle and providing reference validation data sets for climate models and space-based observations (Wunch et al., 2011). Recently, these high-resolution FTIR observations have been extended by COCCON (COllaborative Carbon Column Observing Network, Frey et al. (2019)), a

research infrastructure of portable, compact, low-resolution ground-based FTIR instruments set up as a supplement to TCCON.

Given its strategic location, one of the most relevant ground-based FTIR stations is Izaña Observatory (IZO), where FTIR observations have been carried out since 1999 coincidentally with other high-quality atmospheric measurements (Cuevas et al., 2019). IZO is located in the subtropical belt ($\sim$ 30ºN), in the descending branch of the Northern Hadley atmospheric circulation cell and within the so-called subtropical transport barrier (Schneider et al., 2005, and references therein). This area,

the transition between tropics and mid-latitudes, plays a crucial role in the chemical and dynamical transport processes in the



atmosphere and is a direct tracer of climate change. Recent studies have evidenced, for example, that the tropical belt has expanded over the past few decades, meaning that the descending limb of the Hadley cells is shifting towards the poles in both hemispheres (Heffernan, 2016, and references therein). This poleward movement of large-scale atmospheric circulation systems, such as storm tracks and jet streams, and their associated subtropical dry zones, may lead to profound changes in the

global climate system, affecting natural ecosystems, biodiversity and water resources (Seidel et al., 2008). Together with the so-called tropical bloating, climate models predict a speed-up in the stratospheric Brewer-Dobson circulation in response to current global warming, boosting an ozone recovery in the extratropics at the expense of a delay in the tropics and subtropics (Hegglin and Shepherd, 2009; Li et al., 2009; WMO, 2014a, 2018, and references therein). Nevertheless, these complex phenomena and their implications on the Earth's climate system and, in particular, on tropical and subtropical regions, are still

poorly understood (Seidel et al., 2008; Heffernan, 2016). Unfortunately, due mainly to geographical and political factors, these areas suffer from a great lack of observations that allow their atmospheric structure and composition to be comprehensively investigated. Hence, the high-quality long-term FTIR measurements acquired at IZO provide excellent potential for climate research.

In this context, the present paper gives an overview of the FTIR measurement programme at IZO, going over its history

during its first 20 years of operation (1999-2018) and its current status, as well as exploring its great value for long-term climate research. Although the IZO FTIR station currently operates in the framework of NDACC, TCCON, and COCCON, this review work focuses on NDACC FTIR activities throughout the entire 20-year period. For this purpose, the current paper has been structured as follows: Sections 2 and 3 describe the measurement site and NDACC FTIR products (solar measurements, retrieval principles and strategies and product characterisation), respectively. Section 4 assesses the long-term performance of

the IZO FTIR instruments. Sections 5 and 6 present the time series of all products contributing to NDACC (total column -TC-amounts and volume mixing ratio -VMR- vertical profiles), illustrating the potential of the IZO long-term records with some research examples, while Section 7 briefly reviews other scientific applications of the IZO NDACC data. In Section 8, quality assessment of the different NDACC products is carried out by comparing them to other high-quality measurement techniques available at IZO. Finally, Section 9 summarises the most significant results and conclusions drawn from this work.

## 75  2   Izaña Observatory and FTIR Programme

IZO is a high-mountain station located on the Island of Tenerife (Canary Islands, Spain) in the subtropical North Atlantic Ocean (28.3ºN, 16.5ºW) at an altitude of 2.37 km a.s.l. (Figure 1). The observatory is managed by the Izaña Atmospheric Research Centre (IARC, https://izana.aemet.es/), which belongs to the State Meteorological Agency of Spain (AEMet).

IZO is located on the top plateau of Izaña mountain normally above a well-established thermal inversion layer and is char-

acterised by the quasi-permanent subsidence regime typical for subtropical regions (Figure 1 (b)). Locally, diurnal insolation generates a slight up-slope flow of air originating from below the inversion layer that can disturb the free troposphere conditions at IZO. But, during night-time, the subsidence regime prevails and the atmospheric observations taken at IZO are well-representative of the subtropical North Atlantic free troposphere (Cuevas et al., 2019, and references therein). This, to-





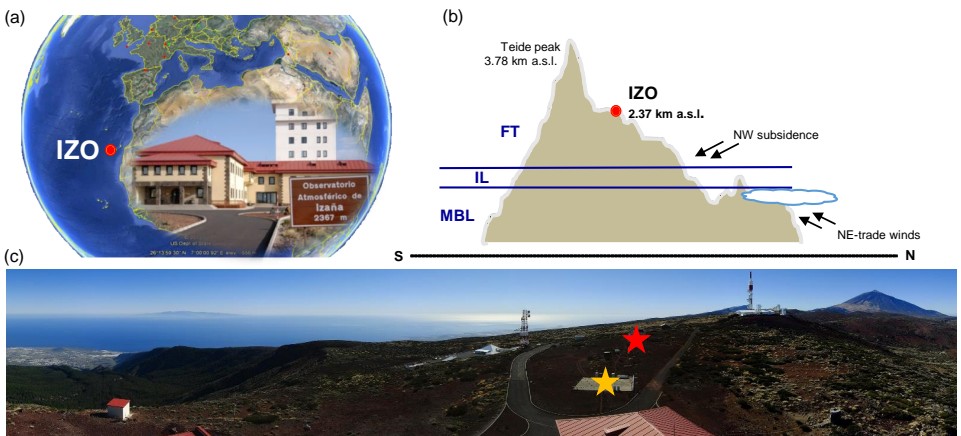

**Figure 1.** (a) Location of IZO in the subtropical North Atlantic Ocean (28.3ºN, 16.5ºW, on the Island of Tenerife); (b) transect of Tenerife island N-S showing the position of IZO and the vertical stratification of Tenerife: MBL-marine boundary layer, IL-inversion layer, and FT-free troposphere; and (c) northern and eastern panoramic views of IZO, as observed from the terrace of the observatory's instrumentation tower. Orange and red stars in (c) represent the location of the IFS 120M (1999-2005) and IFS 120/5HR spectrometers (2005-present), respectively.

gether with the fact that IZO is far from principal pollutant emission sources, means there are very clean, dry atmospheric

conditions at the observatory during most of the year. These environmental conditions account for the historical importance of this observatory and its comprehensive measurement programme for atmospheric composition monitoring. IZO was inaugurated in its present location on $1^{st}$ January 1916, initiating uninterrupted meteorological and climatological observations until the present day (Cuevas et al., 2019). Since 1984, IZO has contributed to the GAW-WMO (Global Atmospheric Watch-World Meteorological Organization) programme and to multiple international networks and databases (WDCGG, WOUDC,

NDACC, TCCON, AERONET, BSRN, MPLNET, E-GVAP, NOAA/ESRL/GMD CCGG,...). Refer to Cuevas et al. (2019) for more details about IZO and its atmospheric monitoring programmes.

Within IZO's atmospheric research activities, the FTIR programme was established in 1999, in the framework of a collaboration between the AEMet and the KIT (Karlsruhe Institute of Technology), with the main goals being the long-term monitoring of atmospheric gas composition and the validation of satellite remote sensing measurements and climate models (Schneider

et al., 2005). Since then, two Bruker high-resolution FTIR systems have been operated at IZO: an IFS 120M from 1999 to 2005 and an IFS 120/5HR from 2005 until present day. These activities have routinely contributed to NDACC and TCCON since 1999 and 2007, respectively. Since 2018 NDACC and TCCON activities have been complemented by a portable, low-resolution FTIR spectrometer (a Bruker EM27/SUN), which operates within the COCCON research infrastructure.





## 3 NDACC FTIR Products

### 3.1 Solar Measurements

The IZO FTIR instrument records direct solar absorption spectra in the near and mid infrared spectral region (NIR and MIR, respectively), using a set of different fieldstops, narrow-bandpass filters and detectors. Within the NDACC activities, the solar spectra were acquired between 700 and 4500 $cm^{-1}$ with a spectral resolution of 0.0036 $cm^{-1}$ (250 cm of maximum optical path difference, $OPD_{max}$) until April 2000, and of 0.005 $cm^{-1}$ ($OPD_{max}$=180 cm) onward. These MIR spectra are recorded

using a potassium bromide (KBr) beamsplitter and two liquid-nitrogen cooled semiconductor detectors: a photovoltaic Indium Antimonide (InSb) detector from 1850 to 4500 $cm^{-1}$ and a photoconductive Mercury-Cadmiun-Telluride (MCT) detector between 700 and 1300 $cm^{-1}$. For routine FTIR operations several scans are co-added in order to increase the signal-to-noise ratio, thereby the acquisition of one spectrum takes several minutes.

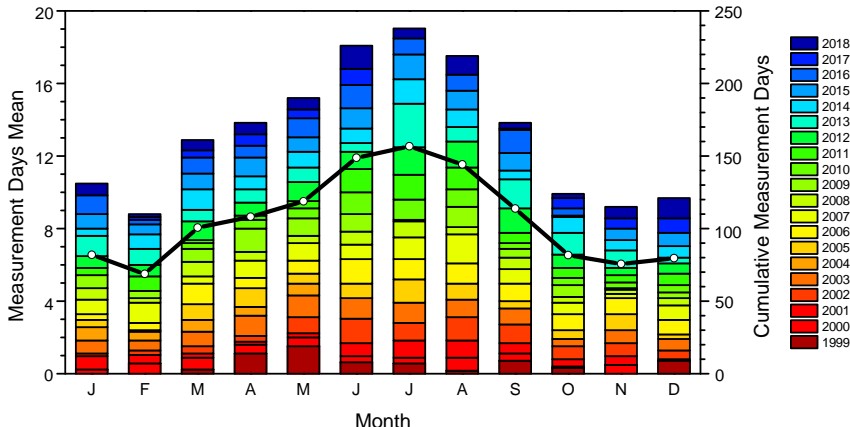

**Figure 2.** Monthly distribution of NDACC FTIR measurement days from 1999 to 2018. The left axis corresponds to monthly mean of measurement days over the entire period (black line), while the right axis represents cumulative measurement days per month from 1999 to 2018 (coloured scale).

FTIR solar spectra are only recorded when the line of sight between instrument and the sun is cloud-free. Given the strategic

location of IZO, these conditions are very common with an average of 179.5 days a year of clear days in the 1981-2010 climate period (García et al., 2019). Thus, FTIR solar measurements at IZO are typically taken about two or three times a week. Although the maximum number of measurement days is concentrated in the warmest months, as shown in Figure 2, the monthly distribution of sampling days over a year is quite uniform. The total number of NDACC measurement days amounts to 2056 for the 1999-2018 period, with an annual average of ∼100 measurement days a year.

An overview of the history of the FTIR instruments at IZO is given in Table 1. As previously mentioned, FTIR measurements started in 1999 with the installation of a Bruker IFS 120M. This spectrometer was replaced in 2005 by a more sophisticated





model, a Bruker IFS 120/5HR, which continues operating today. During March and April 2005 both instruments measured side-by-side, which allows the consistency of the spectrometers to be documented (García et al., 2012b; Sepúlveda et al., 2012). The two FTIR systems are placed in air-conditioned scientific containers (Figure 1 (c)), and been operated in vented

mode (i.e. the spectrometer not evacuated) due to the especially dry conditions at IZO.

**Table 1.** History of the FTIR instruments at IZO with the most important interventions and data gaps.

| Date | Intervention |
|------|-------------|
| January 1999 | Installation of Bruker IFS 120M, start of NDACC observations |
| June-December 2004 | Failure of entrance window |
| January 2005 | Installation of Bruker IFS 120/5HR |
| March-April 2005 | IFS 120M and IFS 120/5HR side-by-side observations |
| May 2007 | Installation of NIR detector (InGaAs), start of TCCON observations |
| June 2008 | Optic re-alignment and recording mode change from AC to DC |
| November 2009 | Installation of laser board ECL03 and optical long-pass filter |
| December 2009-March 2010 | Failure of scanner's motor |
| November 2010 | Installation of laser board ECL04 |
| April 2012 | Update of solar tracker hardware and installation of CamTracker software |
| February 2013 | Optic re-alignment |
| December 2015 | Installation of laser board ECL05 |
| August 2016 | Replacement of internal reference He-Ne laser due to frequency instability |
| October 2016-February 2017 | Failure of solar tracker controller |
| June 2017 | Replacement of internal reference He-Ne laser due to frequency instability |
| September 2017 | Replacement of pre-amplification electronics of MIR detectors |
| December 2018 | Replacement of MIR detectors (MCT and InSb) |

The IZO FTIR instruments have been very stable, especially the IFS 120/5HR system, and only two optic re-alignments have been required during the first 20 years of operation (in June 2008 and February 2013). Apart from that, the most relevant interventions have been the replacement of the internal reference laser used for controlling the sampling of the interferogram in 2016 and 2017, due to frequency instabilities, and the solar tracker upgrade in 2012, when the quadrant-diode set-up was

replaced by the CamTracker system (Gisi et al., 2011). By evaluating the image of the sun on the FTIR's entrance fieldstop acquired by a digital camera, the CamTracker system significantly improves the traditional tracking accuracies (at better than 10 arc seconds), and minimises FTIR pointing errors. In addition, some minor instrumental issues have occurred during these 20 years, causing short data gaps (see Table 1). For further details about the solar FTIR measurements at IZO, refer to Schneider et al. (2005), Sepúlveda et al. (2012), and García et al. (2012b).



## 3.2  Atmospheric Remote Sensing Retrieval Principles

By evaluating spectral signatures of vibrational-rotational transitions contained in the solar absorption spectra measured, the FTIR technique allows total column amounts and low-resolution vertical profiles of different atmospheric trace gases to be retrieved with a high degree of precision. For this purpose, refined FTIR retrieval strategies and inversion principles are used, based on the formalism given by Rodgers (2000). In summary, in the inversion procedure, the measurement (solar absorption spectrum) is assembled into a measurement vector $\mathbf{y}$, while the unknowns are described by a state vector $\mathbf{x}$ and a parameter vector $\mathbf{p}$, which define the state of the atmosphere and the auxiliary and instrumental parameters, respectively. These magnitudes are connected by a forward model $\mathbf{F}$ that describes the physics of the measurement process (interaction of solar radiation with the atmosphere):

$$\mathbf{y} = \mathbf{F}(\mathbf{x}, \mathbf{p}). \tag{1}$$

This is an ill-posed problem, i.e., there are many different atmospheric states ($\mathbf{x}$) that produce almost identical spectrum ($\mathbf{y}$). To overcome this, the solution state is constrained by setting up a cost function:

$$[\mathbf{y} - \mathbf{F}(\mathbf{x}, \mathbf{p})]^{\mathrm{T}} \mathbf{S}_{\mathbf{y}}^{-1} [\mathbf{y} - \mathbf{F}(\mathbf{x}, \mathbf{p})] + [\mathbf{x} - \mathbf{x}_{\mathrm{a}}]^{\mathrm{T}} \mathbf{S}_{\mathrm{a}}^{-1} [\mathbf{x} - \mathbf{x}_{\mathrm{a}}]. \tag{2}$$

The first term is a measure for the difference between the measured spectrum ($\mathbf{y}$) and that simulated for a given atmospheric state ($\mathbf{x}$), taking into account the part of the measurement signal which is not explained by the forward model assuming the state $\mathbf{x}$ and parameter values $\mathbf{p}$ ($\mathbf{S}_{\mathbf{y}}$ is the covariance matrix of $\mathbf{y} - \mathbf{F}(\mathbf{x}, \mathbf{p})$). The second term is the regularisation term. It constrains the atmospheric solution state ($\mathbf{x}$) towards an a priori most likely state ($\mathbf{x}_{\mathrm{a}}$), whereby the kind and strength of the constraint are defined by the a priori covariance matrix ($\mathbf{S}_{\mathrm{a}}$). The constrained solution is reached at the minimum of the cost function Eq. (2).

Due to the nonlinear behaviour of $\mathbf{F}(\mathbf{x}, \mathbf{p})$, the cost function, Eq. (2), is minimised iteratively by numerical methods. For the $(i+1)$th iteration it is:

$$\mathbf{x_{i+1}} = \mathbf{x_a} + \mathbf{G_i}[\mathbf{y} - \mathbf{F}(\mathbf{x_i}, \mathbf{p}) + \mathbf{K_i}(\mathbf{x_i} - \mathbf{x_a})], \tag{3}$$

where $\mathbf{K}$ is the Jacobian matrix (derivatives that capture how the measurement vector $\mathbf{y}$ will change for changes in the atmospheric state $\mathbf{x}$), and $\mathbf{G}$ is the gain matrix (derivatives that capture how the retrieved state vector $\hat{\mathbf{x}}$ will change for changes in the measurement vector $\mathbf{y}$).

Because the vertical resolution of a remote sensing FTIR instrument is limited, a proper description of the relation between retrieved and actual state must be provided. This information is theoretically characterised by the averaging kernel matrix ($\mathbf{A}$), which is calculated as $\mathbf{A}=\mathbf{KG}$ and samples the derivatives that capture changes in the retrieved state $\hat{\mathbf{x}}$ for changes in the actual atmospheric state $\mathbf{x}$. $\mathbf{A}$ links the retrieved and true state as follows:

$$\hat{\mathbf{x}} - \mathbf{x}_a = \mathbf{A}(\mathbf{x} - \mathbf{x}_a). \tag{4}$$



Therefore, $\mathbf{A}$ describes the smoothing of the real atmospheric distribution due to the use of a constrained retrieval, and thus, vertical resolution and sensitivity that can be achieved by a remote sensing FTIR system. While the columns of $\mathbf{A}$ provide the response of the retrieved profile to a perturbation in the state vector, the rows of $\mathbf{A}$ describe the altitude regions that mainly contribute to the retrieved profile and therefore the vertical distribution of the FTIR sensitivity. As a measure of the total sensitivity, the trace of $\mathbf{A}$ (also so-called "Degrees Of Freedom for Signal", DOFS) gives the number of independent layers

discernible by the remote sensing instrument.

     Rewriting Eq.(4) and considering potential errors, the retrieved state $\hat{\mathbf{x}}$ can be linearised about a reference profile $\mathbf{x}_a$ (the a priori profile), the estimated model parameters $\hat{\mathbf{p}}$, and the measurement noise $\epsilon$ as:

$$\hat{\mathbf{x}} = \mathbf{x_a} + \mathbf{A}(\mathbf{x} - \mathbf{x_a}) + \mathbf{GK_p}(\mathbf{p} - \hat{\mathbf{p}}) + \mathbf{G}\epsilon, \tag{5}$$

where $\mathbf{K_p}$ represents the model parameter Jacobian matrix (i.e. the sensitivity matrix to model parameters). Eq.(5) will be the

basis for the analytic error estimation of the retrieved NDACC products, where the first term corresponds to the smoothing error associated with the limited vertical sensitivity of the FTIR instrument, the second term accounts for errors due to uncertainties in the input/model parameters, and the third term provides the measurement noise. An extensive treatment of the atmospheric remote sensing retrieval principles is given in Rodgers (2000).

### 3.3   Retrieval Strategies

At IZO the FTIR programme routinely contributes to NDACC with TCs and VMR profiles of ethane ($C_2H_6$), methane ($CH_4$), chlorine nitrate ($ClONO_2$), carbon monoxide (CO), hydrogen chloride (HCl), hydrogen cyanide (HCN), hydrogen fluoride (HF), nitric acid ($NO_3$), nitrous oxide ($N_2O$), and ozone ($O_3$) (so-called "standard" products hereafter). All these compounds are retrieved with the non-linear least-squares fitting algorithm PROFFIT (PROFile FIT, Hase et al., 2004), considering the spectral regions and interfering gases given in Table 2. The inversion procedure is solved using a first order Tikhonov-Phillips

regularisation (L1, Tikhonov, 1963) for all NDACC products, with exception of $ClONO_2$ which is obtained using a scaling retrieval. The a priori VMR profiles for the target gas and interfering gases are taken from WACCM (Whole Atmosphere Community Climate Model, version 6, http://waccm.acd.ucar.edu) provided by NCAR (National Center for Atmospheric Research; J. Hannigan, private communication, 2014). It is important to remark that this a priori information does not vary over time (i.e. on a seasonal or yearly basis), whereby all variability observed in the retrieved NDACC FTIR products comes exclusively

from the measured solar spectra. Regarding spectroscopy parameters, these are mainly taken from the HITRAN spectroscopy database (HITRAN2008 with 2009 updates, Rothman et al., 2009, and references therein), with the exception of $C_2H_6$ and $ClONO_2$, for which specific cross sections or pseudolines are used (Birk and Wagner, 2000; Harrison et al., 2010). Finally, the NCEP (National Centres for Environmental Prediction) 12.00 UT daily temperature and pressure profiles are used for the forward simulations.

All of these settings are based on NDACC-IRWG recommendations (Infrared Working Group, IRWG, 2014) with small modifications. Most relevant changes are those related to $CH_4$, for which the spectral micro-windows are adopted from Sepúlveda et al. (2014), and the spectroscopy parameters correspond to the improved linelist provided by Dubravica et al.





**Table 2.** Summary of the spectral regions and interfering gases considered for standard and non-standard NDACC products at IZO. For details about specific retrieval strategies for the non-standard products refer to Vigouroux et al. (2018) for $H_2CO$, to Schneider et al. (2016) and Barthlott et al. (2017) for water vapour isotopologues, to Hase (2000) and Rinsland et al. (2003) for NO and $NO_2$, and to Lejeune et al. (2016) for OCS.

| Target Gas | Micro-Windows [$cm^{-1}$] | Interfering Gases | Gas | Micro-Windows [$cm^{-1}$] | Interfering Gases |
|---|---|---|---|---|---|
| | | **Standard NDACC Products** | | | |
| $C_2H_6$ | 2976.63-2977.06 | $H_2O$, $O_3$, $CH_4$ | HCN | 3268.00-3268.50 | $H_2O$, $N_2O$, $O_3$ |
| | 2983.10-2985.00 | | | 3287.00-3287.40 | $C_2H_2$, $CO_2$ |
| | 2986.46-2986.92 | | | 3299.40-3299.60 | |
| $CH_4$ | 2611.60-2613.35 | $H_2O$, HDO, $CO_2$ | | 3315.70-3315.86 | |
| | 2613.70-2615.40 | $NO_2$, $N_2O$, OCS | | 3331.40-3331.80 | |
| | 2835.55-2835.80 | HCl, $O_3$ | HF | 4000.90-4001.05 | $H_2O$, $O_3$, $CH_4$ |
| | 2903.82-2903.92 | | | 4036.85-4039.08 | |
| | 2914.70-2915.15 | | $HNO_3$ | 867.00-870.00 | $H_2O$, $CO_2$, OCS |
| | 2941.51-2942.22 | | | 872.25-875.20 | $C_2H_2$, $CFCl_2$ |
| $ClONO_2$ | 779.90-780.32 | $H_2O$, $C_2H_2$, $CO_2$ | $N_2O$ | 2481.30-2482.60 | $H_2O$, $O_3$, $CH_4$ |
| | 779.90-782.38 | $O_3$,$HNO_3$ | | 2526.40-2528.20 | $CO_2$, $N_2O$ |
| CO | 2057.50-2058.20 | $H_2O$, $O_3$, $CO_2$ | | 2537.85-2538.80 | |
| | 2069.40-2069.90 | $N_2O$, OCS | | 2540.10-2540.70 | |
| | 2140.40-2141.40 | | $O_3$ | 991.25-993.80 | $H_2O$, $CO_2$,$C_2H_4$ |
| | 2153.20-2160.00 | | | 1001.47-1003.04 | $^{668}O_3$, $^{686}O_3$ |
| HCl | 2727.73-2727.83 | $H_2O$, $O_3$, $CH_4$ | | 1005.00-1006.90 | |
| | 2775.60-2775.90 | HDO, $N_2O$, $NO_2$ | | 1007.348-1009.000 | |
| | 2821.40-2821.75 | OCS | | | |
| | 2925.75-2926.10 | | | | |
| | | **Non-Standard NDACC Products** | | | |
| $H_2CO$ | 2763.42-2764.17 | HDO, $CO_2$, $O_3$ | NO | 1900.00-1900.12 | $H_2O$, $CO_2$, $O_3$ |
| | 2765.65-2766.01 | $CH_4$ | | 1900.49-1900.54 | $N_2O$ |
| | 2778.15-2779.10 | | | 1903.10-1903.16 | |
| | 2780.65-2782.00 | | | 2152.00-2152.08 | |
| $H_2^{16}O$ | 2610.35-2610.80 | $CH_4$, $CO_2$, $O_3$, | $NO_2$ | 2914.55-2914.74 | $H_2O$, $CH_4$, $O_3$, |
| $H_2^{18}O$ | 2613.70-2615.40 | $N_2O$, HCl | | 2925.84-2925.95 | HCl |
| HDO | 2626.30-2627.00 | | OCS | 2030.75-2031.06 | $H_2^{16}O$,$H_2^{18}O$, CO |
| | 2658.70-2661.80 | | | 2047.85-2048.24 | $^{12}C^{16}O^{18}O$, $CO_2$, $O_3$ |
| | 2662.25-2664.35 | | | 2049.77-2050.18 | |
| | 2712.50-2714.10 | | | 2051.18-2051.46 | |
| | 2732.050-2732.875 | | | 2054.33-2054.67 | |
| | 2818.800-2820.125 | | | | |
| | 2878.70-2880.70 | | | | |
| | 2892.45-2893.45 | | | | |
| | 3019.575-3020.200 | | | | |
| | 3052.15-3052.75 | | | | |





(2013). Another minor modification affects the absorption lines used for $O_3$ retrievals, which are a simplification of the refined set-up presented by Schneider and Hase (2008). This strategy has been found to provide more precise $O_3$ estimations than

those retrieved from the traditional NDACC approach (1000-1005 cm$^{-1}$ broad micro-window) when comparing to independent measurements (Schneider et al., 2008a, b; García et al., 2021).

In addition to the standard NDACC products, the IZO FTIR programme also contributes to this database with other trace gases (not required by the network, and so-called "non-standard" products hereafter): nitrogen dioxide ($NO_2$), nitrogen oxide (NO), carbonyl sulphide (OCS), formaldehyde ($H_2CO$), and water vapour isotopologues ($H_2^{16}O$, $H_2^{18}O$ and $HD^{16}O$). These

non-standard NDACC gases are also retrieved with the PROFFIT code, using the settings and references listed in Table 2. Water vapour isotopologue observations have been centrally retrieved and quality-filtered in the framework of the MUSICA project (MUlti-platform remote Sensing of Isotopologues for investigating the Cycle of Atmospheric water, Schneider et al., 2012, 2016; Barthlott et al., 2017). Note that, for the water vapour products, the $\delta$-notation is used to express the relation of the observed isotopologue ratio to the standard ratio VSMOW (Vienna Standard Mean Ocean Water), whereby $\delta D = \frac{HD^{16}O/H_2^{16}O}{VSMOW-1}$

(the ratio for $H_2^{18}O$, $\delta^{18}$, has not been considered in the current work due to very weak signal at IZO, refer to Barthlott et al. (2017) for details about this MUSICA product).

All FTIR products presented here correspond to those publicly available from the NDACC archive (www.ndaccdemo.org). MUSICA water vapour isotopologues are also available at the NASA LaRC Airborne Science Data for Atmospheric Composition (www-air.larc.nasa.gov). The only quality filter applied on public FTIR products is that observations taken at high solar

zenith angles ($\geq$85°) have been excluded to avoid imprecise retrievals (mainly caused by misalignments of the solar tracker or spectroscopic issues). These data represent less than 1% of the total data set.

### 3.4  Product Characterisation: Vertical Sensitivity and Uncertainty Budget

The vertical sensitivity of an FTIR system changes significantly from gas to gas, since it depends on the target gas considered, geometry of observation and instrumental issues (e.g. the signal-to-noise ratio) or retrieval strategy. This fact can clearly be

observed in Table 3, which summarises the DOFS' statistics for all NDACC products. It can be seen that the FTIR vertical sensitivity ranges from roughly resolving four independent layers of $O_3$ vertical distribution (mean total DOFS of 4.12) to only retrieving information about the TCs of $H_2CO$, $NO_2$ and $ClONO_2$ (recall that these gases are retrieved using a scaling retrieval, thereby the total DOFS is theoretically equal to unity). Between two and three atmospheric layers are discernible for $CH_4$, CO, HCl, HCN, $HNO_3$, $H_2^{16}O$, $N_2O$, and OCS, while for $C_2H_6$, $\delta D$, HF, and NO, the sensitivity is limited to one or two

layers. These total DOFS values mean that the vertical resolution amounts to roughly 10 km (from the ground up to the middle stratosphere), except for $H_2O$ isotopologues. For the latter, the sensitivity is mainly confined to the troposphere and the vertical resolution ranges from 2-3 km in the lower troposphere and up to $\sim$8 km in the upper troposphere, as shown in Figure 3. This figure depicts the **A** rows for typical measurement conditions at IZO for all NDACC products, where the altitude of the layers that are well-resolved by the FTIR instrument are identified (and highlighted in coloured lines). For example, for $O_3$, the four

resolvable layers are the troposphere, tropopause region, the lower stratosphere and the middle stratosphere, while for trace





**Table 3.** Overview of standard and non-standard NDACC products: number of measured spectra (N), mean (M) and standard deviation ($\sigma$) of the total DOFS, M and $\sigma$ of the statistical uncertainty (Sta. Unc., in %), and M and $\sigma$ of the systematic uncertainty (Sys. Unc., in %). Note that $H_2^{16}O$ values correspond to the simple MUSICA water vapour product (refer to Section 4.3 of Barthlott et al. (2017)), while $\delta D$ products are taken from the quasi optimal estimation of $\{H_2O, \delta D\}$-pair data (refer to Section 4.4 of Barthlott et al. (2017)). (*) refers to those trace gases evaluated using a scaling retrieval.

| Gas | N | DOFS | | Sta. Unc. [%] | | Sys. Unc. [%] | |
|---|---|---|---|---|---|---|---|
| | | M | $\sigma$ | M | $\sigma$ | M | $\sigma$ |
| **Standard NDACC Products** | | | | | | | |
| $C_2H_6$ | 13625 | 1.48 | 0.15 | 1.81 | 0.39 | 5.44 | 0.17 |
| $CH_4$ | 13625 | 2.42 | 0.14 | 0.55 | 0.09 | 3.22 | 0.06 |
| $ClONO_2$ | 4323 | 1.00 | 0.00* | 105 | 1792 | 94 | 1584 |
| CO | 5255 | 3.08 | 0.12 | 0.51 | 0.06 | 2.11 | 0.03 |
| HCl | 13625 | 1.98 | 0.19 | 1.84 | 0.44 | 5.06 | 0.16 |
| HCN | 3872 | 2.13 | 0.14 | 11.7 | 2.30 | 16.1 | 1.79 |
| HF | 4128 | 1.73 | 0.13 | 1.39 | 0.28 | 5.10 | 0.07 |
| $HNO_3$ | 5885 | 2.17 | 0.36 | 2.18 | 0.51 | 8.76 | 0.24 |
| $N_2O$ | 13625 | 2.87 | 0.16 | 0.44 | 0.04 | 2.33 | 0.04 |
| $O_3$ | 5797 | 4.12 | 0.18 | 1.79 | 0.14 | 5.05 | 0.01 |
| **Non-Standard NDACC Products** | | | | | | | |
| $H_2CO$ | 11124 | 1.00 | 0.00* | 52 | 19 | 54 | 18 |
| $H_2^{16}O$ | 12237 | 3.13 | 0.16 | 0.94 | 0.27 | 1.21 | 0.21 |
| $\delta D$ | 12237 | 1.71 | 0.13 | 3.21 | 1.05 | 9.78 | 2.63 |
| NO | 5253 | 1.48 | 0.27 | 3.75 | 1.04 | 4.55 | 0.23 |
| $NO_2$ | 13551 | 1.00 | 0.00* | 7.56 | 2.07 | 11.7 | 0.64 |
| OCS | 5252 | 2.74 | 0.15 | 1.37 | 0.16 | 2.84 | 0.07 |

gases with a total DOFs of about two, such as $CH_4$ or HCN, the FTIR system basically distinguishes between signals from the troposphere and the stratosphere.

The characterisation of the retrieved FTIR products is completed by a theoretical assessment of expected uncertainties according to Eq.(5), which evaluates how different sources of errors can be propagated into the retrieved products. At IZO, the error budget, analytically performed by PROFFIT software, includes the impact of measurement noise and the model parameter sources accounting for instrumental/model aspects (baseline parameters, Instrumental Line Shape -ILS- function, solar pointing, atmospheric temperature profiles, solar lines, and spectroscopic parameters), which are split into statistical and systematic contributions. Further details about the uncertainty analysis are given in Appendix A.



**Figure 3.** Example of the averaging kernels (**A**) rows, on a logarithmic scale, for standard and non-standard NDACC products for typical measurement conditions at IZO (spectra taken on $20^{th}$ July 2013 at a solar zenith angle of ∼30º). Note that for $H_2O$, only the **A** rows of the main water vapour isotopologue ($H_2^{16}O$) are shown. Coloured lines represent **A** rows at altitudes representative of the layers discernible by the FTIR instrument. For a better representation, the ordinate limit for each product has been adapted depending on whether the trace gas has a significant contribution in the middle/upper stratosphere (y-limit of 40, 50, or 60 km) or is predominantly distributed in the troposphere/lower stratosphere (y-limit of 10, 15, or 25 km).





Statistics on the uncertainties for all NDACC products are also included in Table 3. The statistical uncertainty mean over the
20-year period (one standard deviation, $\sigma$, in brackets) ranges from $\sim$0.4% ($\sim$0.04%) for $N_2O$ up to $\sim$50% ($\sim$20%) and $\sim$100%
($\sim$1800%) for $H_2CO$ and $ClONO_2$, respectively. The latter, as great values point out, are particularly difficult to retrieve from
the solar absorption spectra at IZO due to their weak spectral signature and the relatively low TCs recorded at a subtropical
station under background conditions (Kohlhepp et al., 2012; Vigouroux et al., 2018). Overall, statistical uncertainties are
dominated by measurement noise, baseline, and atmospheric temperature errors (e.g. Schneider et al., 2006, 2008a, 2012;
García et al., 2012b; Sepúlveda et al., 2012; Vigouroux et al., 2018; García et al., 2021).

A similar pattern is found for systematic uncertainty contributions: $H_2CO$ and $ClONO_2$ present the maximal errors, $\sim$50 %
($\sim$20%) and $\sim$100 % ($\sim$1600%) respectively, while the main water vapour isotopologue $H_2^{16}O$ shows a mean bias lower
than 1.5 % ($\sim$0.2%). For all NDACC gases, the systematic uncertainty budget is led by spectroscopic errors. In the case of
MUSICA water vapour isotopologue products, these are retrieved using an improved spectroscopy based on HITRAN2012,
but modifying line intensities (S) and broadening parameters ($\gamma$) by about 5-10 % (Schneider et al., 2016; Barthlott et al.,
2017). This modification is introduced to correct the bias in the water vapour profile products documented by Schneider et al.
(2015), whereby very small systematic errors are expected.

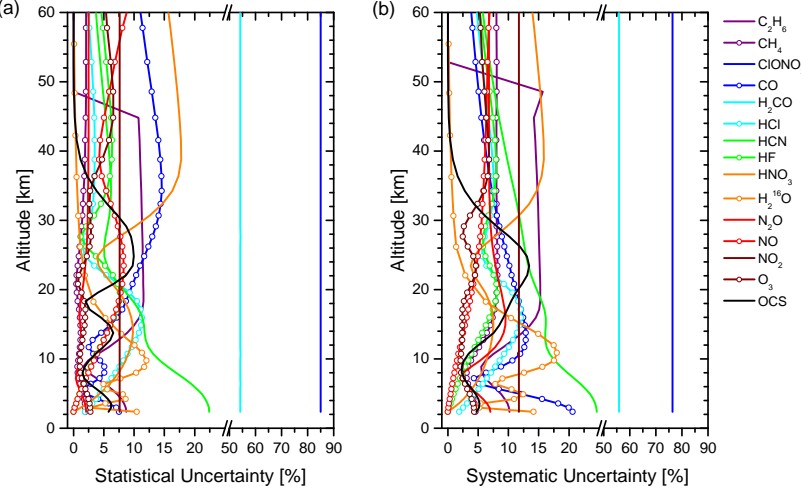

**Figure 4.** Example of estimated uncertainty profiles for all standard and non-standard NDACC products for typical measurement conditions
at IZO (same spectrum as Figure 3): (a) total errors due to statistical uncertainty sources, and (b) total errors due to systematic uncertainty
sources. Total errors are computed as the square root of the quadratic sum of all statistical and systematic error sources considered according
to Appendix A. Note that for water vapour only the error profiles of the main isotopologue ($H_2^{16}O$) are shown.

Figure 4 illustrates how uncertainties in TCs are vertically distributed for all NDACC products. Typically, uncertainty vertical
profiles are largely linked to the vertical distribution of each trace gas and FTIR vertical sensitivity (recall Figure 3). Hence,
depending on the target gas, the predominant errors are mainly located in the troposphere, upper troposphere/lower stratosphere
(UTLS) region, or middle/upper stratosphere. Statistical uncertainties between 5-10% are expected in the lower, middle and





upper troposphere for $C_2H_6$, CO, $H_2^{16}O$, and OCS, and as high as 20% for HCN tropospheric VMR estimates. HCl, HF, HNO, NO and OCS are found to be especially sensitive to uncertainties in the UTLS region, while OCS and $O_3$ exhibit the major error impacts in the middle/upper stratosphere. Large values are also detected in the upper stratosphere for CO and $HNO_3$, but

a subtle impact on the TCs might be expected given the low concentrations of these gases at those altitudes. For $CH_4$ and $N_2O$, the error values are mostly limited to ∼2.5% throughout atmosphere. The systematic uncertainties behave similar to statistical vertical profiles, although in general, the error values are slightly higher. Particularly, high error profiles are estimated for $H_2CO$ and $ClONO_2$, and for both statistical and systematic contributions, due to their weak spectral signatures and low abundances at IZO as mentioned above.

## 4 Long-term Performance

The long-term performance of ground-based FTIR instruments can be assessed through indirect tests. Here, the evolution of the Instrumental Line Shape (ILS) function and solar pointing are analysed, as well as the total column-averaged amount of dry air ($X_{air}$) and of carbon dioxide ($XCO_2$) retrieved from NDACC FTIR spectra in order to identify instrumental inconsistencies and to document temporal stability of the long-term IZO FTIR time series.

### 4.1 Instrumental Line Shape Function

A precise knowledge of the ILS function is essential to properly characterise instrument performance, since the ILS affects the absorption line shape on which the retrieved information is based. This is of particular importance when stratospheric gases are concerned due to the full width at half maximum of their sharp absorption lines and ILS have similar magnitudes (Schneider et al., 2008a; Schneider and Hase, 2008; Sun et al., 2018). Therefore, the ILS function at IZO has been routinely

monitored about every two months since 1999 using re-filled $N_2O$ cells at a pressure of 10 Pa. The ILS is then retrieved from the $N_2O$ absorption lines using LINEFIT software (v14.5), as described in Hase (2012), and applied in the NDACC atmospheric retrievals. Note that the ILS function depends on instrumental configuration (i.e. fieldstops, detectors, optical filters, ...), thereby at IZO the ILS information is estimated independently for each detector. For the MCT configuration, two broad micro-windows combining saturated and un-saturared $N_2O$ absorbing lines between 1235.0-1279.5 and 1291.8-1301.9 $cm^{-1}$ are used, while

for the InSb detector one micro-window between 2173.2 and 2210.0 $cm^{-1}$ is considered (Hase, 2012). In addition, sealed HBr cell measurements have been taken occasionally since 1999.

Continuous monitoring of the ILS function through independent cell measurements ensures that the actual instrumental status is taken into account in NDACC retrievals, but it also allows instrumental alignment and temporal stability to be verified. As an example, Figure 5 depicts the time series of the ILS's modulation efficiency amplitude (MEA) and phase error (PE)

parameters for the NDACC filter 4 measurement settings (InSb detector) for the IZO FTIR instruments between 1999 and 2018. This figure documents that, in addition to suffering from a higher level of spectral noise in the cell and atmospheric measurements, the ILS of the IFS 120M spectrometer is less stable over time than the ILS of the IFS 120/5HR. It further illustrates how punctual interventions on the spectrometers can properly correct the instrumental issues detected: the MEA





temporal degradations were mitigated by two punctual optic re-alignments in 2008 and 2013, while the PE asymmetries were
minimised by replacing the internal reference laser in 2016 (recall Table 1). In the last years, the IFS 120/5HR has been very
stable with a loss of MEA not exceeding 2% and PE limited to ±0.04 rad throughout the OPD range. As documented by
García et al. (2021), the ILS time series for the MCT configuration is very consistent with that reported for the InSb detector,
corroborating proper instrumental characterisation of the IZO FTIR instruments.

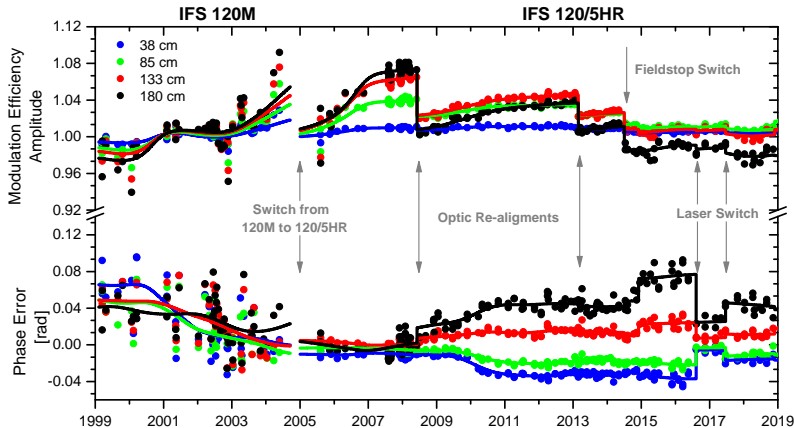

**Figure 5.** Time series of the modulation efficiency amplitude (MEA), and phase error (PE, in rad) at four OPDs (38, 85, 133 and 180 cm)
for NDACC filter 4 measurement settings (InSb detector) of the IZO FTIR spectrometers between 1999 and 2018. Data points represent
individual low-pressure $N_2O$ cell measurements and solid lines show smoothed MEA and PE curves. The solid grey arrows indicate punctual
interventions on the instruments: switch from the IFS 120M to the IFS 120/5HR system in 2005, optic re-alignments in June 2008 and
February 2013, change of fieldstop in October 2014, and internal reference laser replacements in August 2016 and June 2017.

### 4.2 Solar Pointing

Mispointing of the solar tracker can generate a Doppler shift of solar lines with respect to telluric spectral features due to the
solar rotation (Gisi et al., 2011). This effect is considered in the operational NDACC retrievals by fitting a separate shift for
solar background lines, whereby effects on trace gas observations are minor. Nevertheless, analysing the Doppler shift also
gives a useful method to estimate the solar tracking accuracy. Figure 6 (a) shows the time series of the Doppler spectral scaling
factor $\Delta\nu/\nu$, which has been retrieved by observing the solar line shifts in the measured MIR spectra around 2104 cm$^{-1}$
using PROFFIT software. After the quadrant-diode set-up with a semitransparent mirror was installed in February 2005 at
the IZO FTIR instrument and further realignments were made in May 2007, an averaged scaling factor of -0.18·10$^{-7}$ with a
scatter of 4.48·10$^{-7}$ was reached. The latter translates into a mispointing precision along the solar equator of ∼35 arc seconds
(Gisi et al., 2011). The Doppler scaling values after 2012 clearly indicate the significant improvement induced from the more
accurate CamTracker system. The precision is a factor of about two lower, and within the range of 20 arc seconds. This ensures
a minor impact of pointing errors on trace gas retrievals, specially at high observing zenith angles (Gisi et al., 2011).



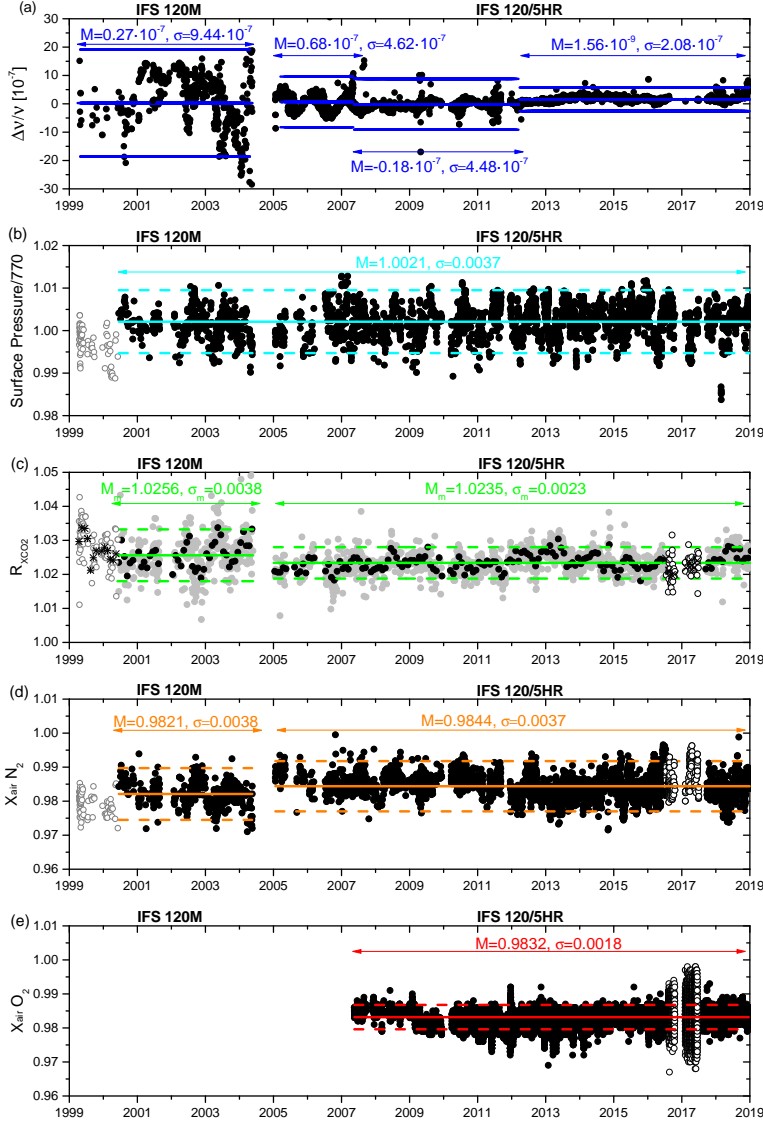

**Figure 6.** Time series of (a) Doppler spectral scaling factor $\Delta\nu/\nu$ estimated from measured MIR spectra around 2104 $cm^{-1}$ solar line; (b) surface pressure (normalised by typical surface pressure at IZO of 770 hPa); (c) $R_{XCO_2}$ ratio; (d) and (e) $X_{air}N_2$ and $X_{air}O_2$ parameters, calculated using $N_2$ TCs from the NDACC FTIR spectra and $O_2$ TCs from the TCCON FTIR spectra, respectively. $M_m$ and $\sigma_m$ stand for mean and standard deviation values computed from the monthly time series (in (c)), while M and $\sigma$ refer to the same statistics, but based on single measurements (in (d) and (e)). The period marked with grey-white dots (in (b), (c), and (d)) denotes instrumental issues on surface pressure records and has been ruled out to compute the statistics displayed in the sub-plots. Black-white dots in (d) correspond to 2016-2017 period, when frequency instabilities were detected in FTIR internal reference laser. Solid and dashed lines represent mean values and $\pm 2\sigma$ ranges, respectively.





### 4.3 $XCO_2$ and $X_{air}$

Two approaches based on gas retrievals from the measured MIR spectra have been examined to assess the long-term consistency of NDACC FTIR data sets. The first approach, the $XCO_2$ method, is based on Schneider et al. (2012), and further elaborated by Barthlott et al. (2015), who demonstrated that the $XCO_2$ retrievals from NDACC MIR spectra (referred to as NDACC $XCO_2$ hereafter) can be used as a proxy for the assessment of the network consistency of the NDACC FTIR measurements. This approach compares retrieved NDACC $XCO_2$ data to a multi-regression $XCO_2$ model that provides information on long-term, seasonal, and latitudinal behaviour of $XCO_2$ (Barthlott et al., 2015). To quantify this relationship, the $R_{XCO_2}$ parameter is defined here as follows:

$$R_{XCO_2} = \frac{\text{NDACC } XCO_2}{\text{Modeled } XCO_2}. \tag{6}$$

The $XCO_2$ model is based on CarbonTracker results and Mauna Loa $CO_2$ in situ records, and adapted to the FTIR measurement site using only latitude and surface pressure as local inputs. On the other hand, the NDACC $CO_2$ TCs were retrieved by analysing four isolated $CO_2$ absorption lines between 2620.55 and 2629.95 $cm^{-1}$, considering a scaling retrieval with a fixed WACCM a priori VMR profile, and using PROFFIT software. Then, the $XCO_2$ is calculated by dividing the $CO_2$ TC by the dry pressure column (DPC) parameter. The DPC is obtained by converting surface pressure ($P_S$, in Pascals) to column air concentration (Sepúlveda et al., 2012; Barthlott et al., 2015), as follows:

$$DPC = \frac{P_S}{g(\varphi)\mu_{dryair}} - TC_{H_2O}\frac{\mu_{H_2O}}{\mu_{dryair}}, \tag{7}$$

where $\mu_{dryair}$ is the molecular mass of dry air ($\sim 28.96 \times 10^{-3} N_A$ kg molecule$^{-1}$), $\mu_{H_2O}$ the molecular mass of water vapour ($\sim 18.02 \times 10^{-3} N_A$ kg molecule$^{-1}$), $N_A$ Avogadro's constant ($\sim 6.022 \times 10^{23}$ molecules mol$^{-1}$), $g(\varphi)$ is the latitude-dependent column-averaged gravitational acceleration, and $TC_{H_2O}$ is the water vapour TC. The $TC_{H_2O}$ data are a result of the MUSICA retrieval (Schneider et al., 2012, 2016, and references therein), and surface pressure is taken from NCEP data used in the retrievals. Refer to Barthlott et al. (2015) for further details about the $XCO_2$ approach.

The second method is based on the $X_{air}$ parameter, which can be used as a sensitive test of the temporal stability of an FTIR instrument because, for $X_{air}$, there is no compensation of possible instrumental problems (Frey et al., 2019). This quantity compares the measured TC of a well-known, very stable reference gas with surface pressure measurements (Eq.(8)). Therefore, for an ideal FTIR instrument $X_{air}$, values should be close to unity and large deviations ($\sim 1\%$) from this threshold might indicate instrumental problems (Wunch et al., 2015; Frey et al., 2019). Here, the nitrogen ($N_2$) absorption signatures measured by the NDACC FTIR spectra have been considered as reference gas.

$$X_{air}N_2 = \frac{F_{N_2}}{TC_{N_2}}DPC, \tag{8}$$

where $F_{N_2}$ is the dry-air mole fraction of nitrogen in the atmosphere (0.7808) and $TC_{N_2}$ is the $N_2$ TC. The latter is retrieved by evaluating four $N_2$ spectral micro-windows between 2403.00 and 2426.3 $cm^{-1}$, considering a scaling retrieval with a fixed WACCM a priori VMR profile, and using PROFFIT software (Goldman et al., 2007).





Figure 6 also shows the time series of the $P_S$ (normalised by the typical surface pressure at IZO of 770 hPa), the $R_{XCO_2}$ and the $X_{air}N_2$ parameters. Consistently, both $R_{XCO_2}$ and $X_{air}N_2$ data are found to be biased by $\sim 2\%$ with respect to unity (upward and downward, respectively), which is very likely due to errors in the MIR spectroscopic parameters of the $CO_2$ and

$N_2$ absorption signatures analysed (Goldman et al., 2007; Barthlott et al., 2015). Therefore, considering $R_{XCO_2} \sim 1.02$ and $X_{air}N_2 \sim 0.98$ as reference, the $XCO_2$ and $X_{air}N_2$ approaches provide consistent results. Anomalous values are detected only in the period 1999-2000, which are attributed to the surface pressure records (marked as grey-white dots in Figure 6). In 2000, there was a change of type and location of the IZO surface pressure sensor (until June 2000 a Thyas sensor with a precision of $\pm 1$hPa has been used, followed by a Setra sensor with a precision of $\pm 0.3$ hPa), leading to a jump of 0.30% in both the $R_{XCO_2}$

and $X_{air}N_2$ parameters. By ruling out this period, the mean $X_{air}N_2$ values ($1\sigma$ in brackets) are 0.9821 (0.0038) and 0.9844 (0.0037) for the IFS 120M and IFS 120/5HR, respectively, while for $R_{XCO_2}$ the mean values are 1.0256 (0.0038) and 1.0234 (0.0023) for the IFS 120M and IFS 120/5HR, respectively. These results agree well with the reference $X_{air}N_2$ and $R_{XCO_2}$ values of 0.98 and 1.02. Note that the reported $R_{XCO_2}$ mean values are computed from the monthly time series as the modeled $XCO_2$ data cannot capture the synoptic time scale variation (i.e. day-to-day variations) (Barthlott et al., 2015).

The switch of spectrometer from IFS 120M to IFS 120/5HR in 2005 entails the most important change identified in both the $X_{air}N_2$ and $R_{XCO_2}$ time series, leading to a mean bias of $\sim 0.20\%$ between both FTIR instruments. In addition, and consistent with the ILS analysis, the IFS 120/5HR system is found to be more stable than the IFS 120M spectrometer (the $R_{XCO_2}$ scatter is reduced by $\sim 65\%$ for the 2005-2018 period and $\sim 3\%$ for the $X_{air}N_2$). However, these differences lie clearly within the estimated confidence ranges for both FTIR systems and it is, therefore, not expected that they will influence the long-term IZO

NDACC time series (e.g. García et al., 2012b; Sepúlveda et al., 2012). The other minor instrumental issues or interventions on the FTIR instruments (recall Table 1) do not seem to affect the $X_{air}N_2$ and $R_{XCO_2}$ time series, since some of them can be partially post-corrected during NDACC gas retrieval processing. That is the case of, for example, the frequency instabilities detected in the internal reference laser in the period 2016-2017. As illustrated in Figure 6 (b)-(c) (black-white dots), this issue has an unestimated impact on the $R_{XCO_2}$ and $X_{air}N_2$ values, because the spectral shift of the measured MIR spectra

is simultaneously fitted by PROFFIT software when retrieving the different NDACC products. No significant temporal drifts were detected in either reference time series (at 95% confidence level), corroborating the long-term temporal stability of the IZO FTIR instruments.

Figure 6 also includes the $X_{air}O_2$ time series, which is estimated similarly to $X_{air}N_2$ (Eq.(8)) but using the oxygen ($O_2$) TCs retrieved from TCCON NIR spectra as the reference gas (Wunch et al., 2015). The $X_{air}O_2$ parameter also suffers from a $\sim 2\%$

bias due to $O_2$ spectroscopic inconsistencies, and a mean typical value of 0.9832 is found for the IZO FTIR instrument. This value is consistent with results reported for other TCCON sites (Wunch et al., 2015). It is worth highlighting that the dispersion for $X_{air}N_2$ ($\sim 0.37\%$) duplicates that found for the IZO FTIR instrument when using the TCCON $X_{air}O_2$ retrievals (0.18%). This different behaviour could, in part, be due to the fact that $N_2$ TCs are retrieved from a few weak $N_2$ absorption lines $\sim 2400$ cm$^{-1}$, whereby they are more sensitive to spectral measurement noise (and disturbing effects). In addition, discrepancies in the

$N_2$ spectroscopic linelists (e.g. inconsistencies and/or airmass dependencies) could contribute to the reported higher variability. Indeed, the NDACC $X_{air}N_2$ time series exhibits a kind of seasonal signal, not observed in the TCCON $X_{air}$ values. Note also





that the $X_{airO_2}$ and $R_{XCO_2}$ results are very coherent, indicating that the $R_{XCO_2}$ parameter can be successfully used to assess reliability and stability of the NDACC FTIR data.

To sum up, the long-term performance analysis indicates that the IZO FTIR spectrometers do not suffer from major in-
strumental issues apart from those already identified. In addition, both instruments have been shown to be stable over time (especially the IFS 120/5HR) and well-characterised during their first 20 years of operation. Therefore, the NDACC FTIR trace gas concentrations measured at IZO can be reliably used for long-term climate research.

## 5   Time Series of NDACC Total Columns

Figure 7 depicts TCs time series for all NDACC products recorded at IZO from 1999 to 2018. Together with all the individual
FTIR measurements, a time series model fitting the different observations is also displayed. This model combines a linear function with a Fourier time series in order to account for variations on different timescales (linear trend, intra-annual, and inter-annual variations) and is calculated according to Eq.(B1) (see details in Appendix B).

In general, the abundances of trace gases observed at IZO are relatively lower than at middle/high latitudes and in polluted areas due to the special measurement conditions of the observatory. As mentioned above, IZO is a high-altitude station, isolated
from local and regional pollution contributions, and located in the descending branch of the northern subtropical Hadley cell. Therefore, very low water vapour or pollution-related gas concentrations are typically measured and, for some trace gases, IZO records are close to the FTIR limit of detection (e.g. $ClONO_2$). These typical background conditions are only sporadically disturbed by long-range transport of pollution and/or biomass-burning events from Europe and North America (Cuevas et al., 2013; García et al., 2017, and references therein), and intrusions of polar or tropical streamer airmasses causing punctual
downward and upward shift of the UTLS region, respectively (Schneider et al., 2005; Cuevas et al., 2013). These episodes are typically observed in winter when, generally, large variations are also detected in the FTIR TCs time series due to a more disturbed atmosphere (see Figure 7). In addition, direct stratospheric air mass intrusions are occasionally detected in spring and summer (Cuevas et al., 2013), leading to an enhancement of tropospheric concentrations of some trace gases (e.g. $O_3$). Chemically long-lived gases, such as HF or $N_2O$, could be used to track these UTLS vertical movements and stratosphere-
troposphere-exchange (STE) events (Schneider et al., 2005). Figure 7 shows the anti-correlation between the extreme TCs of HF and $N_2O$, which are more abundant in the stratosphere and troposphere, respectively. This cross-relationship is also noticeable for other stratospheric gases, such as $O_3$, $NO_2$, or $HNO_3$, and tropospheric compounds, like $CH_4$. On the other hand, during summer, African air masses are frequently advected westwards over the Atlantic Ocean, modifying the atmospheric composition of lower and middle troposphere of the subtropical North Atlantic region. During these episodes, African boundary
layer air is strongly injected into the free troposphere, whereby large mineral dust concentrations are detected at IZO along with industrial pollutants, as well as rather humid, enriched water vapour, and relatively low tropospheric $O_3$ concentrations (García et al., 2012a; Cuevas et al., 2013; González et al., 2016; García et al., 2017, and references therein).

Given the IZO location, the FTIR time series allow changes in the subtropical atmosphere to be investigated, as well as its dynamical and chemical characteristics on different timescales. Apart from the aforementioned short-term episodes, the



**Figure 7.** Time series of standard and non-standard NDACC TCs at IZO from 1999 to 2018. Black dots represent all FTIR measurements, while the multi-regression fit of the time series model (see Eq.(B1)) is depicted as red lines. Data gaps are due to instrumental issues (recall Table 1).





intra-annual variations are somewhat smooth and are largely dominated by the dynamical shift in the height of the subtropical tropopause, which is associated with stratospheric general circulation. The higher the tropopause, the smaller the relative contribution of the stratosphere to the TCs. This results in minimum (maximum) TCs in summer (winter) for long-lived stratospheric gases (e.g. $ClONO_2$, HCl, HF, $HNO_3$) and the opposite behaviour for long-lived tropospheric gases (e.g. OCS, $N_2O$, and $CH_4$), as seen in Figure 7. For photochemically active species, such as $O_3$, NO, and $NO_2$, seasonal variations are

controlled by the joint effect of the height of the tropopause and photochemical reactions, strongly linked to the annual cycle of tropical insolation and middle-stratospheric temperature. These phenomena produce, for example, peak $O_3$ values in spring and minimum in autumn–winter at subtropical latitudes. Overall, the range of these averaged seasonal cycles could expand as higher levels are emitted into the atmosphere from biogenic processes, biomass/fossil fuel burning or other urban/industrial activities.

On a long-term scale, the IZO FTIR time series indicate there is no clear $O_3$ recovery in the TCs with no significant linear trend of -0.04±0.17%$yr^{-1}$ for the period 1999-2018 (here, and hereafter, the trend values are calculated using the monthly time series in order to mitigate sampling effects, according to Eq.(B1), and the error range represents the 95% confidence level, see details in Appendix B). This steady, long-term behaviour agrees with previous studies as summarised by the latest WMO/UNEP ozone assessment (WMO, 2018, and references therein). This report concluded that no significant trend has

been detected in global (60ºS–60ºN) $O_3$ TCs over the 1997–2016 period and statistically significant increases in $O_3$ had been observed only in the upper stratosphere. The reported stratospheric $O_3$ recovery of the last decades is directly attributed to the reduction in the production and release of ozone depleting substances (ODSs), as a response to the Montreal Protocol and its Amendments and Adjustments (WMO, 2014a; Steinbrecht et al., 2017; WMO, 2018, and references therein).

The reduction of ODSs, such as anthropogenic chlorine reservoir species (e.g. HCl and $ClONO_2$), has already been ob-

served in the IZO FTIR time series. As documented in Kohlhepp et al. (2012), a consistent decrease in these compounds by ∼1%$yr^{-1}$ in the period 2000-2009 has been found using globally-distributed NDACC FTIR sites, IZO among them. However, recently, an annual increase has been reported of up to 3% in stratospheric HCl content over the Northern Hemisphere in the period 2007-2012 in contrast with the ongoing monotonic decrease of near-surface emissions (Mahieu et al., 2014). This trend anomaly, also observed in the IZO HCl time series (Figure 8 (a)), has been attributed to a slowdown in Northern Hemisphere

atmospheric circulation. Other stratospheric gases, such as NO and $NO_2$, seem also to be affected by dynamical changes in the lower/middle stratosphere, showing burden accumulations around the period 2007-2012 (Figure 8 (a)). This result is consistent with the positive $NO_2$ trends observed at IZO by Yela et al. (2017), using a refined multi-regression model and a multi-instrument database (ground-based DOAS and FTIR instruments, and space-based MIPAS, OMI, and SCIAMACHY sensors). After the 2011 peak, the updated IZO time series suggest a consistent drop for all the stratospheric gases followed

by a stabilisation of the TCs in the last years. As recently pointed out by Strahan et al. (2020) using NDACC HCl and $HNO_3$ records at a global scale (including IZO among them), this oscillating behaviour can be caused by an extratropical dynamical variability with a 5-7 year period driven by interactions between transport circulation and the quasi-biennial oscillation in tropical winds. This work also reveals that the amplitude of this short-term dynamical variability is large in relation to the long-term trend records, whereby it may have a strong impact trend estimates when using shorter than multi-decadal data





records. Consistently with other northern NDACC stations (Strahan et al., 2020), the IZO records point to a decline in HCl
($-0.21\pm0.19\%\mathrm{yr}^{-1}$) and an increase in $HNO_3$ ($+0.44\pm0.33\%\mathrm{yr}^{-1}$) over the 20-year period. However, when considering the
two decades separately, trends become less apparent and could likely be affected by these short-term dynamical variations as
previously mentioned ($+0.41\pm0.75\%\mathrm{yr}^{-1}$ and $+0.19\pm0.58\%\mathrm{yr}^{-1}$ for the periods 1999-2008 and 2009-2018, respectively, for
$HNO_3$, and $-0.17\pm0.37\%\mathrm{yr}^{-1}$ and $-0.14\pm0.33\%\mathrm{yr}^{-1}$ for the periods 1999-2008 and 2009-2018, respectively, for HCl). Note

that for a better interpretation of the long-term evolution, Figure 8 represents the time series of annual TC anomalies relative
to the 1999-2018 background signal, which are computed from the measured time series according to temporal decomposition
as detailed in Appendix B.

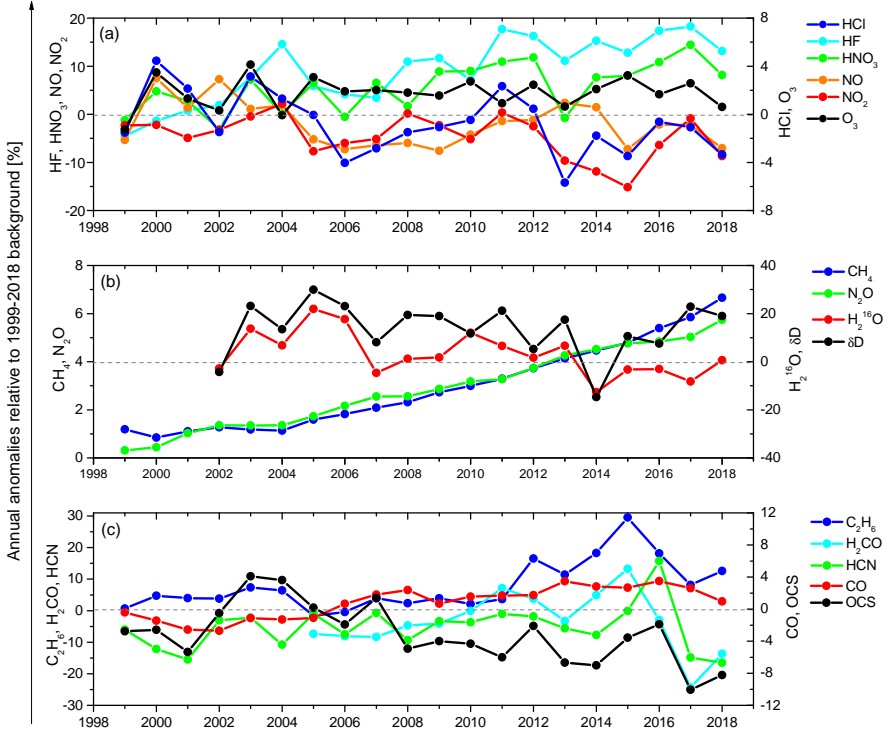

**Figure 8.** Time series of annual anomalies relative to the 1999-2018 background [in %] for NDACC TCs of (a) stratospheric gases (HCl,
HF, $HNO_3$, NO, $NO_3$, and $O_3$), (b) greenhouse gases ($CH_4$, $H_2O$, and $N_2O$) and $\delta D$, and (c) tropospheric and air quality gases ($C_2H_6$, CO,
$HC_2O$, HCN, and OCS). Annual anomalies are computed according to temporal decomposition as detailed in Appendix B.

The stabilisation of TCs in the last decade is also remarkable for HF. Although fluorine emissions were not explicitly re-
stricted by the Montreal Protocol, and many CFC substitutes contain fluorine, which accounts for the observed increase at the

beginning of the 2000's (Figure 8 (a)), the atmospheric fluorine content has stabilised in response to the progressive implemen-
tation of the Montreal Protocol requirements (Kohlhepp et al., 2012; WMO, 2014a, 2018). As a result, the linear HF TC trend is





not significant in the last decade at IZO, $+0.30\pm0.41\%\mathrm{yr}^{-1}$ in the period 2009-2019, compared to a strong positive increment of $+1.84\pm0.54\%\mathrm{yr}^{-1}$ for the period 1999-2008 (the linear trend encompassing the entire period is $+0.98\pm0.27\%\mathrm{yr}^{-1}$).

As controlled ODSs decline, recovery of the stratospheric ozone layer and its long-term evolution largely depends on green-
house gas concentrations as key modulators of stratospheric temperatures (WMO, 2014a; Steinbrecht et al., 2017; De Mazière et al., 2018; WMO, 2018). The NDACC IZO FTIR time series consistently confirm increasing trends of atmospheric green-house gas concentrations widely reported in literature at a global scale (Stocker et al., 2013; Masson-Delmotte et al., 2018) (see Figure 8 (b)). TC data sets show a linear trend between 1999 and 2018 of $+0.28\pm0.01\%\mathrm{yr}^{-1}$ for $CH_4$ and of $+0.29\pm0.02\%\mathrm{yr}^{-1}$ for $N_2O$. The length of the NDACC IZO records also allows a speed-up in the emission rates for $CH_4$ and $N_2O$ to be observed
in the last decade, which is likely caused by the increase in antropogenic emissions. For $CH_4$, for example, Bader et al. (2017) found that anthropogenic activities (such as coal mining, gas and oil transport and exploration) act as secondary contributors to the $CH_4$ global budget and have played a major role in the increase in atmospheric $CH_4$ observed since 2005 at a global scale. This acceleration is remarkable for $CH_4$ with increasing trends of $+0.13\pm0.03\%\mathrm{yr}^{-1}$ and $+0.43\pm0.03\%\mathrm{yr}^{-1}$ for the periods 1999-2008 and 2009-2019, respectively. In case of $N_2O$ this phenomenon is also detectable, but to a lesser extent, with linear
trends of $+0.26\pm0.03\%\mathrm{yr}^{-1}$ and $+0.31\pm0.03\%\mathrm{yr}^{-1}$ for the periods 1999-2008 and 2009-2019, respectively.

In relation to air quality analysis, the IZO FTIR records of tropospheric pollutants (e.g. CO, $C_2H_6$ or $H_2CO$), acquired under background conditions, are well-suited for investigating the effectiveness of the implementation of measures to improve regional and global air quality. In this sense, the long-term behaviour of these gases shows that, while CO records signifi-cantly decrease in the period 1999-2018 with a linear rate of $-0.32\pm0.21\%\mathrm{yr}^{-1}$, the $H_2CO$ content increases until 2015 at
$+2.2\pm0.57\%\mathrm{yr}^{-1}$, which agrees with the findings of Vigouroux et al. (2018). Then, a smooth drop has been detected from 2016 on, leading to a negative linear trend for the entire period ($-0.71\pm0.44\%\mathrm{yr}^{-1}$ in 2005-2018), consistent with long-term CO behaviour (Figure 8 (c)). Specific anthropogenic activities can be further monitored through the measurement of related-gas abundances in the atmosphere. This is the case of non-methane hydrocarbons, such as $C_2H_6$, which shows rather steady values until 2009 when an upturn is detected (Figure 8 (c)). The causes of this sharp rise, also observed at other globally-distributed
NDACC FTIR sites, are similar to those documented for the $CH_4$ increase (i.e. the oil and natural gas production boom in the Northern Hemisphere, particularly in North America) (Franco et al., 2016; Mahieu et al., 2018, and references therein). As with $H_2CO$, after the 2015 peak the IZO time series suggests a stabilisation of the $C_2H_6$ TCs, at least at subtropical latitudes.

In addition to ozone, greenhouse and air quality gases, a key element in the Earth's climate is the water cycle. Ground-based FTIR observations of water vapour isotopologue composition have proven to provide valuable information for understanding
the different water cycle processes (moisture source, transport, cloud processes, and precipitation) and their relation to climate (e.g. Risi et al., 2012; Schneider et al., 2012; Barthlott et al., 2017; Schneider et al., 2012, 2016, and references therein). Figure 7 also includes column-integrated $H_2O$ and $\delta D$ time series recorded at IZO, while Figure 8 displays the corresponding annual anomalies. By analysing the $\{H_2O, \delta D\}$-pair distributions on Figure 7, distinct lower/middle tropospheric moisture pathways can be identified in agreement with multi-year surface references (Schneider et al., 2016; González et al., 2016): air masses that
are clearly affected by dry convection over the African continent (associated to depleted $\delta D$ and high $H_2O$ values, and more concentrated in summer months); Atlantic dry air masses from high altitudes and high latitudes, as the result of condensation





events occurring at low temperatures (characterised by enriched $\delta D$ and low $H_2O$ concentrations); and Atlantic humid air masses resulting from the mixing with lower-level and more humid air during transport (associated to intermediate $\delta D/H_2O$ situations).

## 6 Time Series of NDACC VMR Vertical Profiles

One of the most valuable potentials of NDACC FTIR MIR spectra is the capability of retrieving, albeit roughly, information about the vertical distribution of many trace gases. This is fundamental to monitoring and investigating, for example, the sources/sinks of greenhouse gases and their transport throughout the atmosphere, STE mechanisms, any possible changes of tropospheric and stratospheric chemistry, or the evolution of the ozone layer. Figures 9 and 10 display time series of the

IZO FTIR VMR vertical profiles for all NDACC products from 1999 to 2018, where some of the results drawn from the TC analysis are even more clearly observed. This is the case of, for example, the seasonal shift in the subtropical tropopause altitude, resulting in maximum heights in late spring and summer, and minimum values in winter. As previously outlined, vertical profiles of chemically stable gases, such as HF or $N_2O$, can track these seasonal upward/downward movements, and they can identify sporadic episodes on short-term timescales due to the presence of polar or tropical streamer air masses and

subsidence stratospheric transport.

Figure 11 illustrates how the vertical information provided by the NDACC FTIR data can be used for long-term analysis at different altitudes, as an example, for $O_3$, $CH_4$, and $N_2O$. In case of $O_3$, the steady recovery in TCs found at IZO is likely due to the increase in stratospheric $O_3$ possibly being partially compensated for by a decrease in $O_3$ in the troposphere and tropopause regions at subtropical latitudes (e.g. García et al., 2012b; Vigouroux et al., 2015; Steinbrecht et al.,

2017; Gaudel et al., 2018; WMO, 2018, and references therein). In the framework of the SPARC/LOTUS (Stratosphere-troposphere Processes And their Role in Climate/Long-term Ozone Trends and Uncertainties in the Stratosphere, www.sparc-climate.org/activities/ozone-trends) activity, Steinbrecht et al. (2017) found significant $O_3$ increases in the upper stratosphere ($\sim$42 km) by +0.1 to +0.3%yr$^{-1}$ outside the polar regions by analysing $O_3$ profile trends at a global scale over the period 2000 to 2016 from several merged satellite $O_3$ data sets and from different ground-based NDACC techniques (including the IZO

FTIR records). At tropospheric levels this work has also documented that $O_3$ trends have been smaller and not statistically significant since 2000.

In addition, within the first TOAR (Tropospheric Ozone Assessment Report, www.igacproject.org/TOAR) and encompassing all current tropospheric $O_3$ measurement techniques (surface, sondes, aircraft, lidar, FTIR, Umkehr Dobson and Brewer, satellite, and IZO FTIR records among them), Gaudel et al. (2018) assessed the present-day distribution and trends of tropo-

spheric $O_3$ relevant to climate and evidenced that today there is no consistent picture of $O_3$ tropospheric changes around the world. Although some techniques have pointed to a tropospheric $O_3$ enhancement since the 1970s in the free troposphere, when focusing on the period from 2000 on, significant uncertainties and discrepancies show up. These findings are consistent with the latest WMO/UNEP report (WMO, 2018), and the results obtained from $O_3$ profile trends at IZO for the period 1999-2018: a decline in $O_3$ is detected in the troposphere and tropopause regions (-0.20$\pm$0.38%yr$^{-1}$ and -0.22$\pm$0.43%yr$^{-1}$ in the 2.37-





5.6 km and 12-22 km layers, respectively), while an enhancement in $O_3$ is observed in the stratosphere (+0.12±0.15%yr$^{-1}$ and +0.03±0.15%yr$^{-1}$ in the 22-29 km and 29-45 km layers, respectively). However, all these linear trends are not found to be significant at the 95% confidence level. This may, in part, be induced by the smooth $O_3$ recovery observed and expected at tropical and subtropical regions (when compared to higher latitudes, Steinbrecht et al. (2017); WMO (2018)), but also by the large year-to-year $O_3$ fluctuations exhibited in the decade of the 2000s (Figure 11 (a)). As the figure also illustrates, stratospheric

$O_3$ recovery at IZO is clearly visible from 2010 on, consistent with the long-term analysis reported in literature (Steinbrecht et al., 2017; WMO, 2018; De Mazière et al., 2018). The onset of tropospheric $O_3$ increases also becomes apparent from the end of the 2000s (+0.68±0.54%yr$^{-1}$ in the 2009-2018 period). Finally, it should be pointed out that stratospheric $O_3$ seems to reflect the observed long-term behaviour of the pivotal species controlling the chemical cycles of $O_3$ destruction/formation (e.g. chlorine reservoir species, nitrogen oxides, recall Figure 8 (a)).

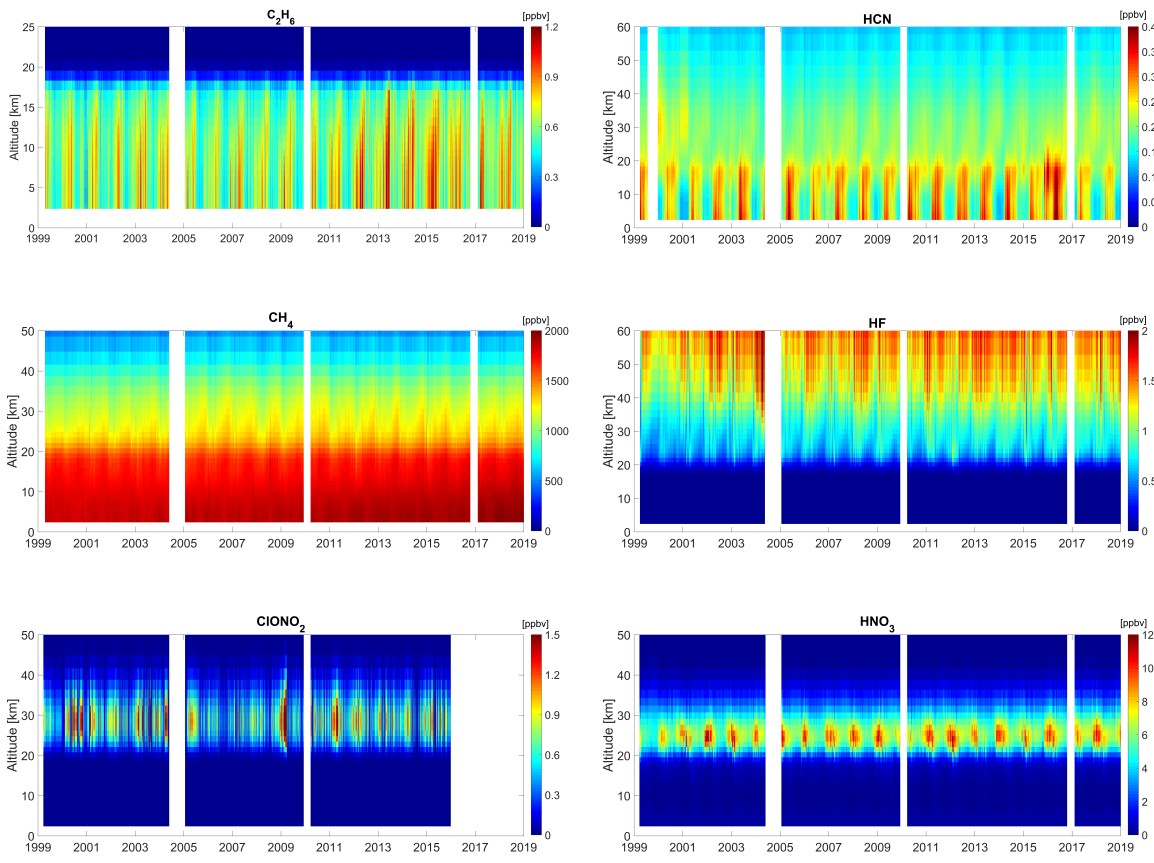

**Figure 9.** Time series of NDACC VMR vertical profiles at IZO from 1999 to 2018: $C_2H_6$, $CH_4$, $ClONO_2$, HCN, HF, and $HNO_3$. Ordinate limits as in Figure 3. Also note that the range of the coloured scale, showing VMR concentrations, has also been adapted for each trace gas. White areas correspond to data gaps due to instrumental issues (recall Table 1).





**Figure 10.** As for Figure 9, but for, CO, HCl, $\delta$D, $H_2^{16}O$, $H_2CO$, $N_2O$, $O_3$, NO, $NO_2$, and OCS.





For greenhouse gases, the long-term TC tendency is largely the result of the increase in the tropospheric concentrations, where the NDACC products provide similar positive rates in the 1999-2018 period: $+0.30\pm0.02\%\mathrm{yr}^{-1}$ and $+0.29\pm0.01\%\mathrm{yr}^{-1}$ in the 2.37-5.6 km layer for $CH_4$ and $N_2O$, respectively. This monotone increment is also reported in the lower/middle stratosphere (see Figure 11 (b)), although marked year-to-year fluctuations are detected, likely due to atmospheric transport processes and dynamical mechanisms, as pointed out in Section 5. Although greenhouse gas concentrations in the stratosphere

are significantly lower than in the troposphere, stratospheric accumulation rates have been found to be significantly greater ($+0.52\pm0.08\%\mathrm{yr}^{-1}$ and $+0.50\pm0.16\%\mathrm{yr}^{-1}$ in the 22-29 km layer for $CH_4$ and $N_2O$, respectively). This result is expected since vertical transport and mixing mechanisms are considerably faster than their respective destruction processes in the stratosphere (i.e. mainly photodissociation for $N_2O$, and oxidation by reaction with hydroxyl radical OH for $CH_4$). This long-term behaviour, and possible short-term variations, may play an important role in modulating stratospheric temperatures, and thus,

affecting the stratospheric chemical cycles (e.g. enhancing the recovery of stratospheric $O_3$, Steinbrecht et al. (2017)).

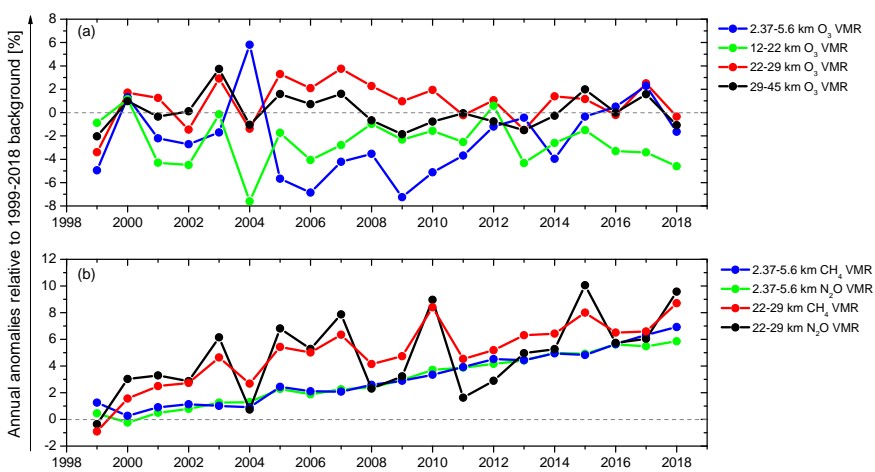

**Figure 11.** Time series of annual anomalies relative to the 1999-2018 background [in %] for NDACC VMR means of (a) $O_3$ (2.37-5.6, 12-22, 22-29, and 29-45 km layers), and (b) $CH_4$, and $N_2O$ (2.37-5.6, and 22-29 km layers). Annual anomalies are computed according to temporal decomposition as detailed in Appendix B.

For other trace gases, the information drawn from the TC analysis only would cover up the vertical distribution of long-term patterns. This is the case of the tropospheric CO records, whose decrease rate is almost three times that for TC abundances: $-0.96\pm0.26\%\mathrm{yr}^{-1}$ in the 2.37-5.6 km layer for the 1999-2018 period. This example, together with those described above, further emphasises the added value of the vertical information provided by the NDACC FTIR data. Nonetheless, it is fair to admit that

the trend estimations of short-lived gases, such as CO, might be influenced by the FTIR sampling effects (as addressed in detail in Section 8.4).





## 7 Other Climate Research Applications

Examples of scientific applications of the IZO FTIR time series are described in detail in Sections 5 and 6, especially those
of use in investigating greenhouse gas budgets and long-term changes in key atmospheric gases, such as ozone, chlorine
and fluorine compounds, air quality gases, etc., at both a regional and global scale. In addition, the validation of remote
observations measured by different satellite instruments has been one of the priorities of the IZO FTIR programme. The high-
quality NDACC FTIR data at IZO have been applied extensively for many years in the evaluation of, e.g., $ClONO_2$, $O_3$, $HNO_3$,
$H_2O$ and isotopologues measured by MIPAS (e.g. Höpfner et al., 2007; Steck et al., 2007; Blumenstock et al., 2013; Risi et al.,
2012); CO, $NO_2$, $H_2O$, and isotopologues retrieved by SCIAMACHY (e.g. De Laat et al., 2010; Risi et al., 2012; Scheepmaker
et al., 2015; Robles-González et al., 2016; Yela et al., 2017); CO, and $HNO_3$ observed by ACE-FTIR (e.g. De Mazière et al.,
2018, and references therein); $O_3$ and $NO_2$ measured by OMI and GOME (e.g. Viatte et al., 2011; Robles-González et al.,
2016); CO, $H_2CO$, and $CH_4$ measured by TROPOMI (e.g. Borsdorff et al., 2018; Vigouroux et al., 2020; Sha et al., 2021);
$O_3$, $N_2O$, $CH_4$, CO, $H_2O$, and isotopologues measured by IASI (e.g. Schneider and Hase, 2011; Viatte et al., 2011; Wiegele
et al., 2014; Schneider et al., 2016; García et al., 2016, 2018). It is worth highlighting that IZO FTIR products, acquired in
the framework of TCCON, have also been widely used for satellite validation applications (e.g. Wunch et al., 2017; Borsdorff
et al., 2018; Hedelius et al., 2019; Sha et al., 2021, and references therein).

Another important contribution of the FTIR observations is the evaluation and development support of global atmospheric
chemistry climate models (CCMs). NDACC IZO data have allowed the performance of many different CCMs to be assessed
(e.g. KASIMA, SLIMCAT, EMAC, SOCOL, LMDZ, IsoGSM, GEOS-Chem, CCMVal initiative, ..., Schneider et al., 2005;
Kohlhepp et al., 2012; Risi et al., 2012; Schneider et al., 2012; Douglass et al., 2014; Mahieu et al., 2014; Bader et al., 2017;
Steinbrecht et al., 2017; Gaudel et al., 2018; Zhou et al., 2019). The attribution of sources/sinks and projected trends of key
atmospheric compounds such as $O_3$, ODSs, and greenhouse gases, or the representation of moist processes have been evaluated,
giving insights into the seasonal cycle, small-scale variations and effects of large-scale circulations in the atmosphere. Although
the model estimates and measurements are generally consistent, there are still significant discrepancies, mainly due to the poor
representation of the dynamical transport, from local to hemispheric scales, in CCMs. Therefore, the synergy of long-term
FTIR observations and simulations may allow for better, deeper understanding of dynamical transport, and chemical processes
in the atmosphere.

## 8 Comparison to other Reference Observations

Quality assessments of different IZO NDACC FTIR products have been partially carried out by previous works (e.g. Schneider
et al., 2008a, 2010b; García et al., 2012b; Sepúlveda et al., 2012, 2014; Robles-González et al., 2016; García et al., 2021).
Nevertheless, such activities have focused on specific time periods, trace gases and the application of different approaches.
Here, the documentation of the quality and long-term consistency of the IZO NDACC FTIR products is presented by using all
reference observations available at IZO within the period 1999-2018 and the same comparison strategy, whereby the results





are directly comparable. Note that this comparison exercise is performed only for those NDACC trace gases for which other
high-quality measurement techniques are available at IZO, i.e., $CH_4$, CO, $H_2O$, $NO_2$, $N_2O$, and $O_3$.

## 8.1   IZO Trace Gas Observations

The following is a brief description of the reference observations used for comparison to the FTIR records. Refer to Cuevas
et al. (2019) for further details about the different measurement techniques and data sets.

1. **Ozone Total Columns**: $O_3$ TC observations are routinely performed with Brewer spectrometers from 1991 on, con-
tributing to NDACC since 2001. Since 2003, IZO has acted as the Regional Brewer Calibration Centre for Europe within
the WMO, which guarantees the high-performance of its $O_3$ activities. By measuring solar ultraviolet absorption spectra,
the IZO RBCC-E reference instruments provide $O_3$ TCs with an uncertainty of ~1 Dobson Unit (DU) (Gröbner et al.,
2017). Together with Brewer data, $O_3$ TCs have also been measured at IZO with the DOAS (Differential Optical Ab-
sorption Spectroscopy) technique since 1993, and within NDACC since 1999, by INTA (Spanish National Institute for
Aerospace Technology, www.inta.es). Based on this technique, a RASAS instrument was installed at IZO from 1998 to
2010, when it was replaced by a MAX-DOAS (Multi AXis DOAS) spectrometer which is still in operation (RASAS-II,
Robles-González et al., 2016; Yela et al., 2017, and references therein). By analysing scattered light at zenith during
twilight (solar zenith angle range of 89-91°), the DOAS approach provides $O_3$ TCs with an expected precision of ~5%
(Bassford et al., 2005). Nevertheless, comparisons between Brewer and DOAS $O_3$ TC records at IZO show an agreement
between techniques within 1% with a standard deviation of only 3% (Gil-Ojeda et al., 2012).

2. **Nitrogen Dioxide Total Columns**: $NO_2$ TCs have also been acquired with the DOAS technique since 1993, and in the
framework of NDACC since 1998, by INTA. The estimated overall uncertainty in the individual $NO_2$ TC measurements
is expected to be, on average, ~12 % (Gil et al., 2008).

3. **Methane, Carbon Monoxide, and Nitrous Oxide Total Columns**: $CH_4$, CO and $N_2O$ TCs have been routinely re-
trieved from the FTIR NIR spectra acquired in the framework of TCCON since 2007 (Blumenstock et al., 2017). The
TCCON FTIR products are retrieved with the GGG Suite software package (current version GGG2014), whose core
part is the non-linear least-squares fitting algorithm GFIT (Wunch et al., 2015). It basically performs scaling retrievals
with respect to the a priori VMR profiles to compute TCs of $CH_4$, CO and $N_2O$, together with $CO_2$, $H_2O$, HDO, and
HF. In order to minimise the impact of instrumental issues on the precision of the TCCON products, retrieved column
abundances are converted to total column-averaged dry-air mole fractions (XGas) by using simultaneously retrieved $O_2$
TCs. The XGas mole fractions are then calibrated onto the WMO's gas standard scale maintained by NOAA (National
Oceanic and Atmospheric Administration, www.noaa.gov) and provided as standard TCCON products (Wunch et al.,
2011, 2015). The calibration factors with respect to the WMO references are estimated by comparing the TCCON re-
trievals to coincident airborne in-situ profiles, leading to values of 0.9765±0.0020 for XCH4, 1.0672±0.0200 for XCO,
and 0.9638±0.0100 for XN2O (Wunch et al., 2015). Although TCCON and NDACC FTIR observations are performed





with the same instrument at IZO, the measurement settings (spectral range, spectral resolution, detectors, filters, field-stop, etc.) and posterior analysis (retrieval software and procedure) are completely different. Thereby, both FTIR data can be considered as independent data sets, and thus, TCCON FTIR observations can be a valid reference for the quality assessment of NDACC FTIR data.

4. **Water Vapour Total Columns**: $H_2O$ TC measurements, or equivalent Precipitable Water Vapour (PWV) amounts, are taken at IZO by many different measurement techniques. In fact, in 2014, IZO was appointed by the WMO as a CIMO (Commission for Instruments and Methods of Observation) Testbed for Aerosols and Water Vapour Remote Sensing Instruments, recognising the high-quality and long-term experience of its water vapour measurement programme (Cuevas et al., 2019). The current study only uses those measurements obtained from CIMEL sunphotometers within AERONET (since 2003), FTIR within TCCON (since 2007), Vaisala RS92 radiosondes (since 2008), and Global Positioning System (GPS) receivers (precise orbits since 2009). Although meteorological radiosonde measurements have been routinely launched on Tenerife since 1994, here we work with the Vaisala RS92 data that have been processed by the GRUAN (GCOS - Global Climate Observing System Reference Upper-Air Network, www.gruan.org) lead centre, which are only available from 2008 on. This limits the comparison data set, however GRUAN data processing ensures that the obtained humidity, pressure and temperature profiles are well-calibrated and highly accurate (Borger et al., 2018, and references therein). The estimated PWV precision of these techniques is 10% for CIMEL, 0.7 mm ($\sim$10–20% for IZO typical conditions) for GPS, and $\sim$6% for the GRUAN Vaisala RS92 sondes (Schneider et al., 2010a; Weaver et al., 2017, and references therein), while for TCCON FTIR $H_2O$ products ($XH_2O$) a scale factor of 1.0180$\pm$0.0040 has been determined using collocated meteorological radiosoundings at globally-distributed TCCON sites (Wunch et al., 2015). Recently, at IZO, further studies have corroborated these results, obtaining a mean bias of -0.006 mm (-1.3%) when comparing TCCON FTIR PWV values to those obtained from meteorological radiosondes launched on Tenerife and processed according to the GRUAN scheme (Almansa et al., 2020). Note that all these PWV techniques are located at IZO, with the exception of meteorological radiosondes, which have been launched twice daily (at 11.15 UT and 23.15 UT) very close to IZO: from the Santa Cruz de Tenerife station (WMO station 60020, 35 km northeast of IZO) until 2002 and from the Guimar station since then (WMO station 60018, 15 km south of IZO).

5. **Ground-level Greenhouse and Carbon Cycle Gases**: Continuous surface measurements of $CH_4$ (since 1984), and $N_2O$ (since 2007) have been routinely carried out at IZO in the framework of the GAW-WMO programme. CO (since 2008) has also been monitored as it affects the $CH_4$ cycle. All of these gases are measured by gas chromatography (GC) techniques ($CH_4$ with a GC DANI 3800, $N_2O$ with a GC Varian 3800, and CO with a GC Trace Analytical RGA-3), with expected uncertainties of $\pm$2 ppbv for $CH_4$, $\pm$0.2 ppbv for $N_2O$, and $\pm$2 ppbv for CO (Gómez-Peláez et al., 2013, 2019, and references therein).

6. **Water Vapour Profiles**: The $H_2O$ profiles are obtained from the Vaisala RS92 meteorological radiosondes launched on Tenerife and evaluated according to the GRUAN scheme, with uncertainties typically from 5% near the ground to values of 5-20% at around 10 km altitude (Borger et al., 2018, and references therein).



7. **Ozone Profiles**: The $O_3$ sonde programme on Tenerife started in November 1992, managed by AEMet, and since March 2001 these activities have formed part of NDACC. The Electrochemical Concentration Cell, ECC, sondes (Scientific Pumps 5A and 6A) were launched once per week from Santa Cruz de Tenerife until 2010 and since then from Puerto de la Cruz (12 km north of IZO). The expected uncertainty of the ECC sondes is $\pm 5$–15% in the troposphere and $\pm 5$% in the stratosphere (WMO, 2014b).

## 8.2  Comparison Strategy

The comparison methodology used here is based on previous works carried out at IZO (e.g. Sepúlveda et al., 2012; García et al., 2012b; Sepúlveda et al., 2014; García et al., 2016, 2018) and can be briefly summarised as follows:

1. **Considerations for comparison of remote sensing techniques**: When comparing different remote sensing retrievals, there are multiple factors affecting the comparison, namely, spectral ranges, spectral resolution, retrieval strategies, vertical sensitivity, etc. This is especially critical for the DOAS and FTIR methods since, together with the aforementioned factors, they are quite different measurement techniques sampling different air masses and with different observing geometries (Gil-Ojeda et al., 2012; Robles-González et al., 2016; Yela et al., 2017). On one hand, the DOAS and NDACC FTIR retrievals use a profile retrieval, however while DOAS considers varying a priori VMR profiles for each target gas, they are kept fixed for the NDACC FTIR retrievals. TCCON FTIR concentrations are obtained using a scaling retrieval assuming a priori VMR profiles varying from day-to-day. On the other hand, the Brewer, CIMEL, and GPS techniques use completely different measurement approaches, which are not based on inverse methods. Other discrepancies or biases arise from the different spectral regions and spectral resolutions used in the different remote sensing techniques. The different measurement approaches and instrument capabilities also lead to different vertical sensitivity. Even using the same instrument and technique, NDACC and TCCON averaging kernels tend to peak at different altitudes and so, NDACC and TCCON FTIR products may reflect concentration variations from different atmospheric layers. All these aspects can introduce significant differences in the retrieved products and must be considered when interpreting the comparison results (e.g. Barthlott et al., 2015; Robles-González et al., 2016; Kiel et al., 2016; Zhou et al., 2019, and references therein).

2. **FTIR Products**: The FTIR TCs of $NO_2$ and $O_3$ are directly compared to the reference IZO data. For $H_2O$, since the reference techniques considered here do not distinguish between $H_2O$ isotopologues, the MUSICA optimal estimation of $H_2^{16}O$ data (so-called "Type 1" products, Barthlott et al. (2017)) were used. The TCCON $H_2O$ TCs were used instead of the standard $XH_2O$ products, and both TCCON and NDACC $H_2O$ TCs were transformed into PWV values to be consistent with the other $H_2O$ techniques. Note that the TCCON PWV values are not post-calibrated with the aforementioned scale factor, since the latter has been determined for $XH_2O$, not for $H_2O$ TCs.

   For $CH_4$, CO and $N_2O$, the NDACC TC data were converted to total column-averaged dry-air mole fractions by using the DPC parameter (Eq.(7)) in order to be compared to the standardised, WMO-calibrated TCCON XGas retrievals. This transformation also allows the analysis of the capability of the NDACC XGas products to capture tropospheric con-





centration variations by comparing them to IZO ground-level concentrations. In addition, the IZO ground-level records are compared to the NDACC tropospheric CH$_4$, CO, and N$_2$O concentrations (CH$_{4_{TRO}}$, CO$_{TRO}$, and N$_2$O$_{TRO}$, respectively), which are obtained as the mean of retrieved NDACC VMR profiles between IZO altitude (2.37 km a.s.l.) and middle troposphere (5.6 km a.s.l.) (Sepúlveda et al., 2012; García et al., 2014).

In order to compare the FTIR vertical profiles to the in-situ highly-resolved profiles (meteorological and ECC sondes), the latter have been vertically-degraded by applying the averaging kernels obtained in the FTIR retrieval procedure, following Eq.(4). By doing so, the limited sensitivity of the FTIR data is properly taken into account in the comparison (Rodgers, 2000). To homogenise the reference data sets, only those sondes with continuous measurements up to 12 km for H$_2$O, and up to 29 km for O$_3$ have been considered. Beyond these altitudes, the sonde profiles have been completed using the corresponding FTIR a priori VMR profiles to compute the smoothed humidity and O$_3$ profiles.

3. **Temporal Criteria**: Temporal collocation depends on the natural variability of each target gas, FTIR uncertainty and characteristics of each reference technique, therefore it varies from gas to gas.

For CH$_4$, CO and N$_2$O, the daily night-time means (20.00-08.00 UT) of the IZO ground-level records and the daily day-time means of the FTIR products are paired (Sepúlveda et al., 2012; García et al., 2018). As previously mentioned, the IZO night-time surface data represent the background regional signal of the free troposphere well, while during day-time local air circulations may disturb the ground-level data. In addition, these trace gases show rather small intra-day variations, so a pairing of daily means could be more meaningful than a comparison of individual measurements. Similar temporal criterion is then applied for the CH$_4$, CO and N$_2$O column retrievals from the NDACC and TCCON data sets, i.e., the daily means of XCH$_4$, XCO and XN$_2$O are matched.

Given the large natural variability of H$_2$O, and in order to ensure that the different techniques observe similar air masses, temporal coincidence criterion has been restricted to 1 hour for all the PWV products (Schneider et al., 2010a). For the radiosonde data, the time at a half of the observation is chosen as reference time (a sonde takes approximately one hour between the launch and burst in the UTLS).

For the Brewer O$_3$ TC the 1-hour collocation is also applied, since the precision of both the FTIR and Brewer techniques is able to properly resolve the intra-day O$_3$ concentration variations (Schneider et al., 2008b; García et al., 2016). For the DOAS-FTIR intercomparison, because the DOAS technique measures only during twilight and O$_3$ and NO$_2$ are photochemically active species, the FTIR observations are averaged before and after 12 UT (a.m. and p.m., respectively) (Robles-González et al., 2016).

Finally, for O$_3$ profile comparison the strategy described by Schneider et al. (2008a) and García et al. (2012b) was followed, where the ECC sondes are corrected daily by means of coincident Brewer data. These works illustrated that by applying this correction the quality and long-term stability of the ECC sonde data can be significantly improved. Given that intra-day O$_3$ variability is much lower than that for H$_2$O, the temporal collocation window is extended to 3 hours around the O$_3$ sonde launch.





To avoid redundant data, for all the intercomparisons, each FTIR measurement is only paired once to that reference observation with the minimal time difference within the temporal collocation window.

4. **Temporal Decomposition**: Once the FTIR and reference IZO observations have been temporally collocated, quality assessment of the FTIR products is addressed, both directly comparing the measured data sets and at different timescales
by means of a temporal decomposition of the measured time series. This temporal decomposition, explained in detail in Appendix B, provides an added value for comparison as it allows the temporal signals discernible by the FTIR system to be properly identified (i.e. single measurements, daily and seasonal averages, and long-term variations) (Sepúlveda et al., 2012, 2014; García et al., 2016, 2018).

## 8.3    Direct and Timescale Comparison

Table 4 summarises the direct comparison between the NDACC FTIR products and the reference data sets considered: TCs of $O_3$, $NO_2$, $H_2O$ (PWV), column-averaged amounts of $CH_4$, CO, and $N_2O$, as well as tropospheric $CH_4$, CO, and $N_2O$ records. In addition, Figures 12 and 13 display the time series of paired FTIR and reference observations and the corresponding relative differences (FTIR-Reference), as well as the multi-year seasonal cycles, while Figure 14 synthesises the comparisons on different timescales.

1. **Ozone:** Similarly to previous works, an excellent agreement is found between FTIR and Brewer $O_3$ TCs at IZO. More than 98% of the $O_3$ TC variance obtained for both techniques agrees (Pearson correlation coefficient, R, of 0.99, i.e. $R^2$=0.98) and the dispersion between techniques is smaller than 1% ($1\sigma$, $\sigma$ stands for the standard deviation of the relative differences with respect to reference data, i.e., Brewer data). This scatter accounts for the precision of both techniques, thereby it could be considered a very conservative value of FTIR precision. A large contribution to this dispersion can be attributed to the impact of atmospheric temperature profile uncertainties on the FTIR $O_3$ retrievals (Schneider
et al., 2008a; Schneider and Hase, 2008; García et al., 2012b, 2021). When considering more refined $O_3$ products (not the standard NDACC product), including a simultaneous temperature profile fit in the $O_3$ retrieval procedure, the scatter between Brewer and FTIR can be significantly reduced by up to 0.5-0.7% (Schneider and Hase, 2008; García et al., 2012b, 2021). Although the agreement is very satisfactory, the Brewer and FTIR products differ in their absolute quantification of $O_3$ TCs: the FTIR is upward biased by ∼3.6%. This well-known systematic discrepancy is likely
introduced by inconsistencies between ultraviolet and infrared spectroscopic parameters. As presented in Section 3.4, uncertainties of 5% in the HITRAN $O_3$ spectroscopy may explain a bias of ∼5% between FTIR and Brewer data.

Although a direct comparison between collocated DOAS and FTIR instruments is challenging, a satisfactory consistency has been documented for $O_3$ TCs: the FTIR-DOAS mean difference is ∼8-10% with a $\sigma$ of ∼3%. These values agree
with the expected precision for DOAS (∼5%) and with previous DOAS-Brewer comparisons carried out at IZO (Gil-Ojeda et al., 2012). Analogously to Brewer, the bias between techniques is likely due to spectroscopic inconsistencies between the spectral regions.





**Table 4.** Summary of intercomparison between the FTIR products and the IZO reference data sets: reference technique, mean bias (MB, in % and absolute units), standard deviation of the differences ($\sigma$, in % and absolute units), slope and intersect of the least-squares fit (Y=FTIR and X=Reference), Pearson correlation coefficient (R), number of coincidences (N), and coincident period. The differences are computed with respect to the reference IZO observations: (FTIR - Reference) in absolute units and (FTIR-Reference)/Reference in %. (*) means $10^{14}$ molec/cm$^2$.

| Product | Reference | MB [%] | MB [unit] | $\sigma$ [%] | $\sigma$ [unit] | Slope | Intersect [unit] | R | N | Period |
|---|---|---|---|---|---|---|---|---|---|---|
| $O_3$ TC | Brewer | 3.57 | 10.5 DU | 0.95 | 2.90 DU | 1.04±0.01 | -1.54±1.29 DU | 0.99 | 5104 | 1999-2018 |
| $O_3$ TC | DOAS a.m. | 7.78 | 22.3 DU | 3.07 | 8.71 DU | 0.88±0.03 | 55.6±8.0 DU | 0.89 | 949 | 2000-2018 |
| $O_3$ TC | DOAS p.m. | 9.88 | 27.9 DU | 3.38 | 9.44 DU | 0.87±0.03 | 62.7±7.3 DU | 0.88 | 1215 | 2000-2018 |
| $NO_2$ TC | DOAS a.m. | -0.10 | -0.11* | 10.8 | 2.91* | 0.67±0.03 | 8.90±0.95* | 0.75 | 1100 | 1999-2018 |
| $NO_2$ TC | DOAS p.m. | -21.7 | -7.77* | 10.0 | 3.82* | 0.56±0.03 | 9.44±1.04* | 0.77 | 1243 | 1999-2018 |
| PWV | CIMEL | 37.9 | 1.66 mm | 11.6 | 0.95 mm | 1.34±0.01 | 0.37±0.02 mm | 0.98 | 11469 | 2003-2018 |
| PWV | GPS | 27.1 | 1.02 mm | 23.2 | 0.60 mm | 1.08±0.01 | 0.65±0.02 mm | 0.98 | 11186 | 2009-2018 |
| PWV | Sonde | 17.2 | 0.72 mm | 18.2 | 0.79 mm | 1.02±0.04 | 0.62±0.22 mm | 0.97 | 154 | 2008-2017 |
| PWV | TCCON | 9.56 | 0.51 mm | 7.63 | 0.48 mm | 1.09±0.01 | 0.05±0.05 mm | 0.99 | 1127 | 2007-2018 |
| $XCH_4$ | TCCON | 0.23 | 4.14 ppbv | 0.37 | 6.58 ppbv | 0.98±0.02 | 38.7±38.7 ppbv | 0.96 | 635 | 2007-2018 |
| $XN_2O$ | TCCON | -0.53 | -1.66 ppbv | 0.66 | 2.05 ppbv | 0.71±0.03 | 88.8±9.9 ppbv | 0.87 | 633 | 2007-2018 |
| XCO | TCCON | 6.08 | 4.61 ppbv | 2.70 | 2.20 ppbv | 1.09±0.02 | -2.18±1.55 ppbv | 0.97 | 629 | 2007-2018 |
| $CH_{4_{TRO}}$ | GAW | 2.56 | 47.8 ppbv | 1.02 | 19.2 ppbv | 1.03±0.03 | -1.91±52.03 ppbv | 0.87 | 1634 | 1999-2018 |
| $N_2O_{TRO}$ | GAW | 0.10 | 0.35 ppbv | 0.54 | 1.77 ppbv | 1.15±0.04 | -49.2±12.1 ppbv | 0.89 | 1013 | 2007-2018 |
| $CO_{TRO}$ | GAW | -13.4 | -11.8 ppbv | 8.31 | 7.65 ppbv | 0.80±0.03 | 7.40±2.65 ppbv | 0.87 | 984 | 2008-2018 |

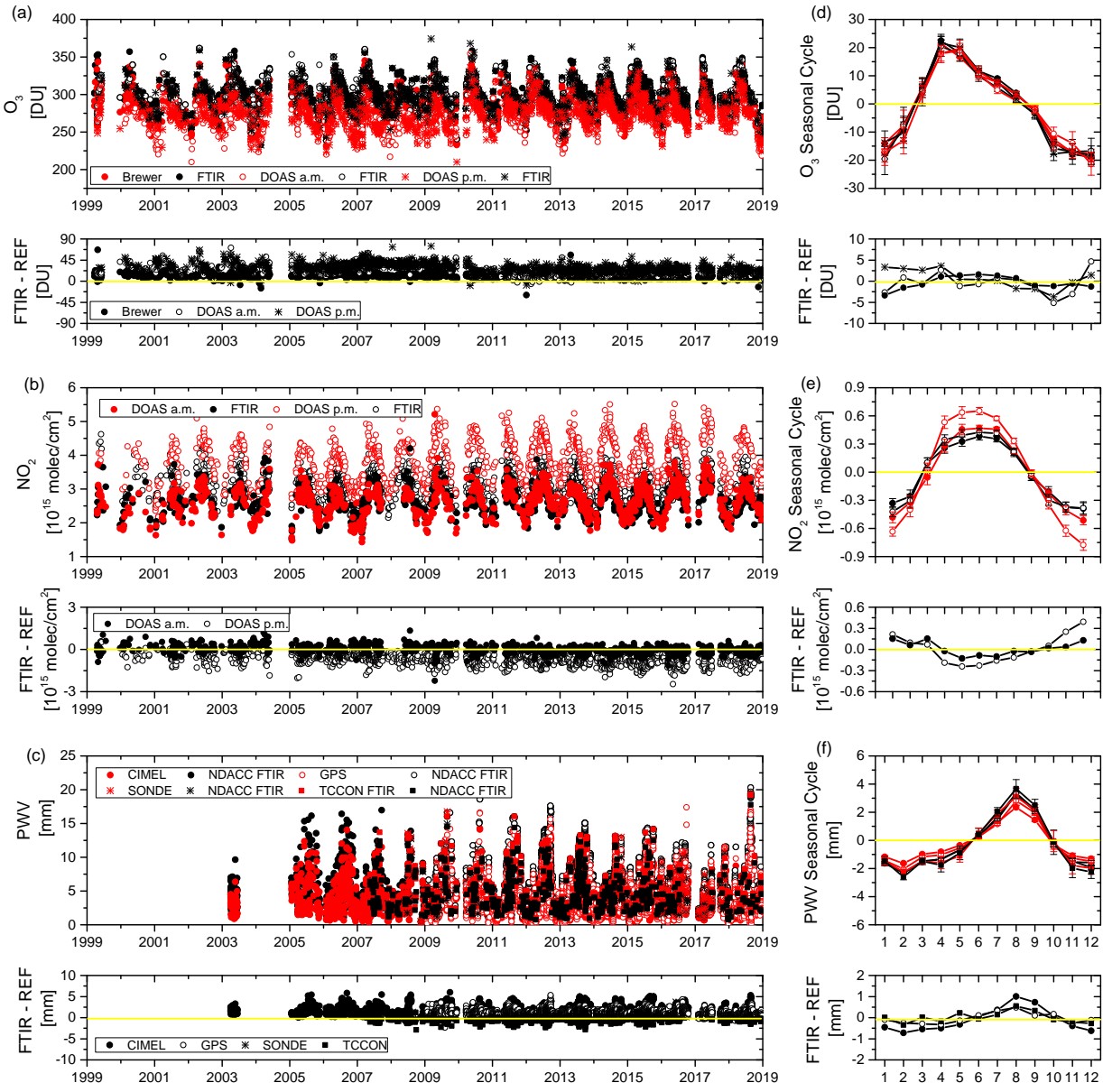

**Figure 12.** Time series of coincident NDACC FTIR TCs and IZO reference data sets, and of differences (FTIR - Reference, in absolute units) on left panels for: (a) $O_3$, (b) $NO_2$, and (c) PWV. Right panels show the averaged annual cycle of variations relative to the long-term background, from temporal decomposition analysis, and of differences for: (d) $O_3$, (e) $NO_2$, and (f) PWV. Error bars in annual cycles are standard errors of the monthly mean: $2 \times \sigma/\sqrt{n}$, with $\sigma$ the standard deviation and $n$ the number of monthly measurements. Yellow line indicates zero value (i.e. no difference or no annual cycle signal).





**Figure 13.** As for Figure 12, but for tropospheric and column-averaged NDACC products: (a) and (d) CH$_4$, (b) and (e) N$_2$O, (c) and (f) CO.





When examining the different timescale signals, the $O_3$ TC variations are similarly captured by all techniques at the short-term (measurement-to-measurement) and seasonal scales, as shown in Figure 12 (right panels) and summarised in Figure 14. For long-term signals, while Brewer and FTIR observations similarly reproduce the $O_3$ evolution with a correlation of ∼95%, DOAS $O_3$ observations show a poorer agreement with the FTIR data (R∼60%). This is likely due to a systematic bias introduced by the switch of DOAS instrument in 2010 (the change-point is clearly recognisable in the difference time series displayed in Figure 12 (a)). As a result, the DOAS long-term trends (14 (e)) are found to overestimate the Brewer and FTIR values. Note that the linear trends shown in Figure 14 (e) have been computed considering the coincident observations and so could differ from the values presented in Sections 5, which are based on the FTIR monthly means.

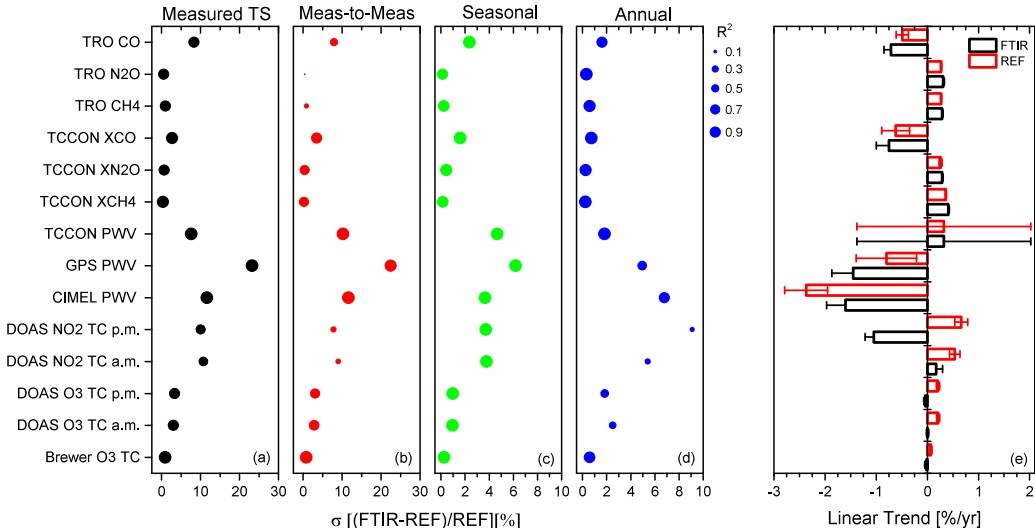

**Figure 14.** Summary of timescale comparison between FTIR and IZO reference data sets: standard deviation of the relative differences ($\sigma$, in %) is displayed on x-axis, and the size of dots represents the determination coefficient, $R^2$. These statistics are shown for the comparison of (a) measured and decomposed time series: (b) measurement-to-measurement, (c) seasonal, and (d) long-term variations (annual means). (e) Linear trends (in %/yr) for coincident FTIR and reference data sets, which are calculated by fitting a linear function combined with a Fourier time series to the data according to Eq.(B1). Errors represent the 95% confidence interval and were determined using the bootstrap method (see details in Appendix B).

2. **Nitrogen Dioxide:** Moderate performance is found when comparing the DOAS and FTIR $NO_2$ products. The scatter obtained for the relative differences, ∼10%, was found to agree well with the expected precision of DOAS $NO_2$ data (∼12%), with the FTIR theoretical uncertainty budget (recall Table 3), and with previous works (∼9-11%, Robles-González et al. (2016)). However, a remarkable asymmetry in the mean differences has been documented between a.m. and p.m. FTIR-DOAS comparisons for all timescales analysed (Figure 12 and Figure 14). In general, both for $O_3$ and $NO_2$, the DOAS a.m. observations compare better with the FTIR data than p.m. values. This pattern is largely introduced





by diurnal variations in the TCs due to photochemical processes, especially important for active species like NO$_2$, which are captured differently by the two techniques. A photochemical correction (solar zenith angle-dependent) can be applied to the FTIR data to refer the measurements to the DOAS acquisition time, which reduces the bias between the techniques and the a.m.-p.m. asymmetry (Robles-González et al., 2016). In addition, part of these differences can be attributed to the different vertical sensitivities of the DOAS and FTIR techniques: while the DOAS method, with zenith-sky measurements at twilight, is almost insensitive to troposphere and tropopause regions (Gil-Ojeda et al., 2012; Robles-González et al., 2016), the FTIR system can detect UTLS contributions (recall Figure 4).

The diurnal asymmetry might be the reason for the inconsistency between the FTIR a.m. and p.m. records for the long-term signals. While the FTIR a.m. observations point to a significant NO$_2$ increase over IZO (similar to that for DOAS data), the FTIR p.m. records seem to indicate the opposite long-term behaviour (Figure 14 (e)). However, further studies would be recommendable to better understand what drives the difference in these linear trends and reconcile them.

3. **Water Vapour:** The consistency between the NDACC PWV product and the other PWV techniques is very satisfactory, with variances of more than 95% in agreement. As expected, the best performance is found for FTIR PWV products acquired within TCCON (correlation higher than 0.99 and scatter of ∼8%), consistent with previous comparisons at globally-distributed FTIR sites (e.g. Weaver et al., 2014). The degree of comparability with the other sun-photometer technique considered (AERONET CIMEL) is also high with a dispersion of ∼12%, while larger scatter values are obtained for the other rather different PWV techniques (18% and 23% for sondes and GPS, respectively). These values are very close to the estimated precision of each measurement technique (recall Section 8.1) and are found to be very consistent with previous PWV comparison works carried out at IZO (Schneider et al., 2010a; Almansa et al., 2020; García et al., 2021). These studies comprehensively compared FTIR PWV retrievals with different techniques (MFRSR, CIMEL, EKO, GPS, and sondes), highlighting the different observing geometries (i.e. sampling different air masses), atmospheric conditions (humid or dry), and the significant clear sky dry bias of sun-spectrometers may account for the differences reported among techniques. Regarding the absolute PWV quantification, the NDACC PWV products show a wet bias with respect to all the PWV reference techniques, ranging from ∼10% to ∼38% for TCCON and CIMEL data, respectively. Part of this overestimation is introduced by the NDACC PWV retrievals, ∼12% (Tu et al., 2020), which agrees with the bias obtained with respect to the calibrated TCCON data. The large bias with CIMEL is likely attributed to calibration issues of the standard AERONET CIMEL PWV products. As recently pointed out by Almansa et al. (2020), a dedicated calibration of the CIMEL H$_2$O channel (centred at 940 nm) at IZO reduces the reported bias by ∼20%.

In relation to PWV seasonality, although the comparability of the different techniques is excellent, the differences among them depend on the PWV values: maximal differences are observed for extreme PWV conditions (i.e. summer and winter months, Figure 12), which may be due in part to the different seasonal sensitivities of the different measurement techniques. At longest timescales, the NDACC FTIR, CIMEL and GPS consistently suggest that PWV values over IZO have been significantly slowing down over the last two decades, but the magnitude of this decrease varies among





techniques (Figure 14 (e)). However, the NDACC-TCCON PWV comparison offers contrary results due in part to a more reduced coincident data set (∼1100 pairs for NDACC-TCCON versus ∼11000 coincidences for NDACC FTIR-CIMEL-GPS).

4. **Carbon Monoxide, Methane, and Nitrous Oxide:** An excellent consistency between the NDACC and TCCON XCO, and XCH$_4$ records is observed for both the original measured time series and the signals on different timescales. The direct correlation is higher than 0.95 for both gases with a scatter of the relative differences of 0.7% and 2.7% for XCH$_4$ and XCO, respectively. These values lie within the expected NDACC systematic uncertainties budget (recall Table 3) and the random uncertainty of the TCCON retrievals (Wunch et al., 2015).

Although the overall comparability of the FTIR XN$_2$O products is satisfactory, it is found to be poorer than the agreement between tropospheric NDACC N$_2$O and ground-level records. This behaviour can be due to different vertical sensitivities of the NDACC and TCCON products, leading to the NDACC XN$_2$O data being more influenced by the seasonality of the UTLS region, as seen in Figure 13 and reported by previous studies (García et al., 2014; Zhou et al., 2019). This figure also highlights the remarkable decoupling (even anti-correlation for XCH$_4$ and XN$_2$O) between the annual cycles

of the XGas and tropospheric products. While the seasonality of XGas observations is mostly dominated by the annual shift in the UTLS region, tropospheric concentrations of CO, CH$_4$, and N$_2$O are determined by the source emission patterns. As indicated by the comparison to ground-level records, the NDACC tropospheric products properly capture the tropospheric seasonal signals, demonstrating great potential for source/sink attribution studies. The performance of the NDACC tropospheric products can be further improved by means of more sophisticated retrieval strategies, allowing

for a reduction of the stratospheric contribution (Sepúlveda et al., 2012, 2014; García et al., 2014). Likewise, the influence of the stratospheric signal on the XGas products can be partially corrected by using co-retrieved XHF data or XN$_2$O (for XCH$_4$) (Sepúlveda et al., 2012; Saad et al., 2014; Wang et al., 2014, and references therein).

In relation to systematic differences, the mean bias is lower than 0.6% for XCH$_4$ and XN$_2$O, indicating no significant spectroscopy inconsistencies between the NIR and MIR spectral regions used for TCCON and NDACC retrievals, re-

spectively. This result is corroborated by comparing the tropospheric NDACC N$_2$O product to the ground-level records. However, for CH$_4$, a bias of ∼2.6% with respect to surface measurements appears, which is compatible with the assumed error of 3% in the spectroscopy intensity parameter (Table 3). Concerning CO, the NDACC and TCCON XCO products generally differ by ∼5-6%, which is likely attributed to TCCON post-calibration, as reported by previous works (Kiel et al., 2016; Zhou et al., 2019). The TCCON XCO calibration factor is 1.0672±0.0200 (recall Section 8), which coincides

with the bias obtained between the TCCON and NDACC XCO products. Note that the comparison with ground-level records points to an underestimation of the NDACC tropospheric CO values (unlike to those observed for the TCs), which further emphasises the presence of a systematic inconsistency.

The timescale analysis was found to be a very useful tool, particularly, for tropospheric NDACC comparisons. As illustrated by Figure 14, the agreement observed between ground-level and tropospheric NDACC products is mainly the

result of seasonal and long-term signals. Tropospheric measurement-to-measurement variations, especially for long-lived





CH$_4$ and N$_2$O gases, are smaller than the FTIR precision and are, therefore, scarcely captured by the remote sensing system (no correlation has been found between FTIR and ground-level observations). Given that CO presents more variable concentrations in the atmosphere, the NDACC FTIR product is able to capture part of its tropospheric variations (correlation of ∼0.60 on a daily scale).

5. **Water vapour and ozone profiles:** Figure 15 summarises the comparison between the FTIR O$_3$ and H$_2$O VMR profiles and the reference observations (smoothed ECC and humidity sondes). The vertical distribution of differences exhibit different patterns for each gas. For O$_3$, the difference profile is almost constant until the UTLS region with a mean bias of ∼10-15% and scatter of ∼6-7%. Beyond this region, the discrepancies considerably decrease to below ∼5% (4.0% at 29 km) likely due to a better sensitivity of the FTIR system to O$_3$ variations (recall Figure 3). Part of the observed dis-
crepancies are introduced by including both FTIR instruments in the comparison. As shown in Section 4, the ILS of the IFS 120M spectrometer is noisier and less stable over time than the ILS of IFS 120/5HR, leading to greater uncertainties in the 120M O$_3$ vertical distribution retrievals. Note that instrumental performance is especially critical for stratospheric gases, like O$_3$, since the ILS affects the absorption line shape on which the retrieved information is based. When considering only the more stable, well-aligned IFS 120/5HR instrument, García et al. (2021) documented an improvement
in comparability between FTIR and ECC sondes by ∼1% up to the UTLS and ∼0.5% in the middle stratosphere. In addition, as with the total columns, refined FTIR retrieval strategies (i.e. including a simultaneous temperature fit) can further improve the FTIR performance (Schneider et al., 2008a; García et al., 2012b, 2021).

On the other hand, the H$_2$O comparison exhibits a strong vertical stratification with largest discrepancies located in the lower troposphere (i.e. just above the island). This pattern is likely due to the substantial impact of the local diurnal up-
slope flow on FTIR observations (Schneider et al., 2016). As stated above, diurnal insolation at IZO generates a thermal up-slope flow from the lowermost humid layers, causing a strong H$_2$O diurnal cycle and thus affecting the H$_2$O signals measured by the FTIR system (Schneider et al., 2010a; González et al., 2016). To examine this effect and ensure pure free conditions, the comparison has been also restricted to those FTIR observations taken at low solar elevation angles (between 25º and 45º) (Schneider et al., 2016), with the resulting difference profile also included in Figure 15 (in red).
Despite the number of coincidences considerably decreases, this restriction ensures an optimal comparison and that quite similar air masses are detected by both FTIR and meteorological sondes. As a result, the comparability is significantly improved. Until the middle troposphere, both the mean bias and scatter range from 15-20%, while they increase in the upper troposphere (mean bias of 45.6% and $\sigma$ of 31.0% at 8 km). At these altitudes, larger temporal and spatial variability of the humidity fields are expected (Schneider et al., 2016, and references therein), which makes the comparability of
the remote sensing and in-situ profiles difficult.

For both gases, the scatter values found agree well with the FTIR error estimation (recall Section 3.4), with the expected uncertainty for the ECC and meteorological sondes (recall Section 8.1), as well as with previous works (Schneider et al., 2006, 2008a, 2010a, b, 2016; García et al., 2012b; Duflot et al., 2017; García et al., 2021, and references therein). As stated in these studies, although the reference sondes have been smoothed by FTIR averaging kernels, the limited vertical





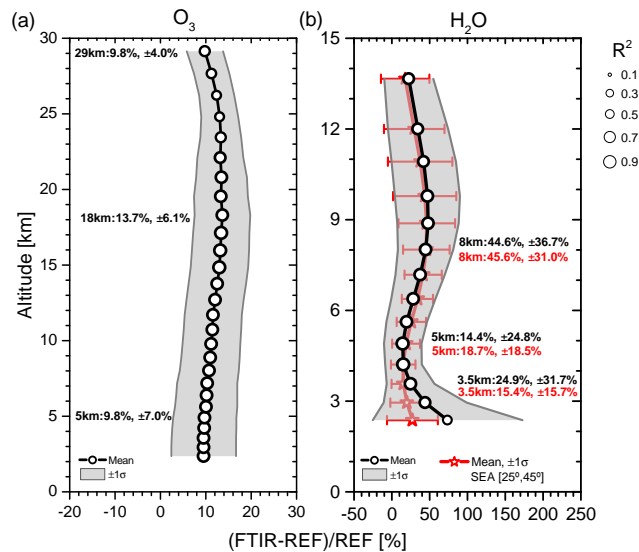

**Figure 15.** Vertical profile of relative differences between NDACC FTIR VMR profiles and IZO reference data sets (mean and $\pm 1\sigma$ in %) for (a) $O_3$ and (b) $H_2O$. For the latter, the comparison considering only coincidences for FTIR solar elevation angle between 25°-45° is also included (in red). The dotted area represents determination coefficient, $R^2$. Statistics of differences at the altitude levels, well-distinguishable by the FTIR instrument, are included in plots (5, 18, and 29 km for $O_3$, and 3, 5, and 8 km for $H_2O$). The number of coincident measurements is 276 in the period 1999-2018 for $O_3$, and 154 and 32 in the period 2008-2017 for $H_2O$ without and with limiting the FTIR solar zenith angle, respectively.

sensitivity of the FTIR profiles could account for part of the dispersion observed between both data sets. Other sources of discrepancies, as mentioned, might be the different observing geometries. Note that the temporal decomposition analysis has not been carried out for profile comparisons given the reduced number of coincidences.

## 8.4   Influence of Sampling

Although the weather conditions at IZO are very favourable for solar measurements, the sampling of FTIR data may not be
regular enough and contain gaps that affect the reliability of the FTIR results, particularly on long-term timescales. In order to examine this effect, Figure 16 compares the linear trends for coincident FTIR-Reference data sets and those computed from the complete reference time series, without pairing to the FTIR observations, in the coincident period shown in Table 4. Note that only those products, for which the sampling of the reference data sets is uniform and continuous, have been considered (i.e. GPS PWV observations and ground-level CO, $CH_4$, and $N_2O$ records). Although some of the other reference techniques
have a higher measurement frequency than the FTIR system, they present their own sampling issues (the Brewer and CIMEL data are also biased towards cloud-free days, the DOAS technique measures only during twilight, and the TCCON data have similar sampling to the NDACC FTIR observations). In addition, as in a similar way to Section 8.3, the linear trends shown





in Figure 16 have been computed considering individual observations to assess the sampling effects and they could, therefore, differ from the values presented in Sections 5 and 6, which are based on FTIR monthly means.

The overall agreement between the different linear trend estimations is quite high. FTIR sampling has been found to have a minor impact, especially, on the stable and long-lived $CH_4$, and $N_2O$ gases. However, for the more variable CO records, a significant discrepancy is found between considering the entire ground-level time series and those paired with the FTIR observations. This artefact could be, in part, attributed to the sparse FTIR sampling and large dynamical variability in the winter months. As shown in Figure 13, the greatest differences between FTIR and ground-level data are concentrated in

winter, which could induce a bias in the trend values. For PWV, the agreement between the coincident and entire GPS time series demonstrates that the uncertainties induced by FTIR sampling might also be considered negligible. Although there are remarkable differences in the magnitude of the obtained PWV linear trends, they lie within the respective confidence error intervals.

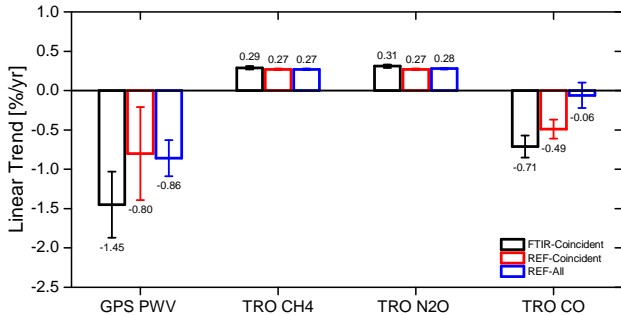

**Figure 16.** Linear trend (in %/yr) for coincident FTIR and reference data sets (FTIR-Coincident and REF-Coincident, respectively) for PWV GPS, and ground-level CO, $CH_4$, and $N_2O$ records. For the reference, the linear trends for the entire time series are also shown, without pairing to FTIR observations, in the coincident period given in Table 4 (REF-All). Trends are calculated by fitting a linear function combined with a Fourier time series to the data according to Eq.(B1) (see details in Appendix B). Trend values are included in the plot for each data set. Error bars represent the 95% confidence interval.

## 9    Summary and Conclusions

Long-term ground-based observations of atmospheric composition are essential to monitor the evolution of the Earth's atmosphere system. Within the NDACC framework, FTIR spectrometry provides abundances of many trace gases simultaneously and with a high degree of precision, which can be used to understand tropospheric and stratospheric chemistry and transport. In this context, the current paper provides an overview of the first 20 years of NDACC FTIR measurements taken at the subtropical Izaña Observatory (IZO, Spain) between 1999 and 2018.

The great potential of the IZO NDACC FTIR records for climate research is internationally recognised, contributing to more than one hundred peer-reviewed scientific papers and to numerous international research activities and projects. The





major accomplishments drawn from these works are briefly reviewed in the current paper, especially those for investigating the greenhouse gas budgets and long-term changes of pivotal atmospheric gases, or the evaluation of space-based observations and climate model estimations. In addition, a comprehensive assessment of the long-term instrumental performance of the IZO

FTIR systems is presented, and of the quality and long-term consistency of the different NDACC FTIR products in a coherent manner.

Together with the long-term monitoring of key atmospheric trace gases, analysing the temporal frequency, duration, and extent of the vertical phenomena over IZO (e.g. STE episodes or UTLS vertical movements) could provide useful insights into long-term changes in the dynamics and chemistry of the subtropical atmosphere. Furthermore, the evaluation of possible links

to dynamical mechanisms or teleconnection patterns and hemispheric phenomena, such as ENSO (El Niño and the Southern Oscillation), NAO (North Atlantic Oscillation) or QBO (Quasi-Biennial Oscillation), might also serve as tracer of climate change. Dedicated studies using the IZO trace gas time series would be of great use in better understanding these drivers and connections on short-term and long-term scales. The NDACC FTIR multidecadal data sets, such as those produced by IZO, are therefore indispensable to address the major challenges of current climate research.

**Appendix A: NDACC Uncertainty Budget**

The NDACC uncertainty analysis includes the impact of measurement noise and the different model parameter sources. Particularly, the error contribution of model parameters can be analytically estimated through the respective error covariance matrix $\mathbf{S_{x,p}}$:

$$\mathbf{S_{x,p}} = \mathbf{G}\mathbf{K_p}\mathbf{S_p}\mathbf{K_p}^T\mathbf{G}^T, \tag{A1}$$

where $\mathbf{S_p}$ is the covariance matrix of the uncertainties $\mathbf{\Delta p}$. In this work, $\mathbf{S_p}$ is estimated considering the error sources, values, and partitioning between random and systematic contributions listed in Table A1. They have been identified as the leading error sources and typical values affecting the different FTIR products (Hase, 2007).

The error covariance matrix for measurement noise ($\mathbf{S_{x,\epsilon}}$) is analytically calculated by

$$\mathbf{S_{x,\epsilon}} = \mathbf{G}\mathbf{S_{y,\epsilon}}\mathbf{G}^T, \tag{A2}$$

where $\mathbf{S_{y,\epsilon}}$ is the covariance matrix for noise in the measurement.

The total statistical and systematic uncertainties (listed in Table 3) are then calculated as the square root sum of the squares of all statistical and systematic errors considered, respectively. Note that the measurement noise is considered as purely random, while the spectroscopy parameters are purely systematic.





**Table A1.** List of sources and values used for the uncertainty analysis. (MEA: modulation efficiency amplitude; $\nu$-scale: spectral position; $S$: intensity; $\gamma$: pressure broadening parameter). The third column provides the partitioning of the error values between statistical (ST) and systematic (SY) contributions.

| Uncertainty Source | Uncertainty Value | ST/SY |
|---|---|---|
| Baseline (channeling and offset) | 0.2 and 0.1 % | 50/50 |
| ILS (MEA and phase error) | 1 % and 0.01 rad | 50/50 |
| Pointing offset | 0.1° | 90/10 |
| Solar lines (intensity and $\nu$-scale) | 1 % and $10^{-6}$ | 80/20 |
| Temperature profile (2.37-10, 11-36, 37-120 km) | 1, 2, 5 K | 70/30 |
| Spectroscopy $\gamma$ | 5 % | 0/100 |
| Spectroscopy $S$ for CO, $H_2^{16}O$, $H_2^{18}O$, $HD^{16}O$, $N_2O$, OCS | 2 % | 0/100 |
| Spectroscopy $S$ for $CH_4$ | 3 % | 0/100 |
| Spectroscopy $S$ for $C_2H_6$, HF, HCl, NO, $O_3$ | 5 % | 0/100 |
| Spectroscopy $S$ for $ClONO_2$, HCN, $H_2CO$, $HNO_3$, $NO_2$ | 10 % | 0/100 |

## Appendix B: Multi-annual Evolution

The multi-annual evolution of the measured time series has been modeled by using a multi-regression fit of different coefficients that consider a mean concentration gas value and variations on different timescales (Sepúlveda et al., 2014; García et al., 2018, and references therein):

$$
\begin{aligned}
x_m(t) \quad = \quad & A_0 + A_1 t + \sum_{1 \leq i \leq N-1} \left\{ A_{\mathrm{sin},i} \sin\left(\frac{2\pi i}{\Delta t} t\right) + A_{\mathrm{cos},i} \cos\left(\frac{2\pi i}{\Delta t} t\right) \right\} \\
& + \sum_{1 \leq i \leq P} \left\{ B_{\mathrm{sin},i} \sin\left(\frac{2\pi i}{\Delta j} j(t)\right) + B_{\mathrm{cos},i} \cos\left(\frac{2\pi i}{\Delta j} j(t)\right) \right\}
\end{aligned} \tag{B1}
$$

Coefficients $A_i$ capture the long-term variations: $A_0$ and $A_1$ are related to the linear changes, while coefficients $A_{\mathrm{sin},i}$ and $A_{\mathrm{cos},i}$ define the amplitude and phases of a Fourier series that considers all frequencies between 1 and $N-1$. Here $N$ is the total number of years covered by the whole time series series and $\Delta t = \max\{t\} - \min\{t\}$ is the time period covered by the whole time series. The coefficients $B_{\mathrm{sin},i}$ and $B_{\mathrm{cos},i}$ capture the intra-annual variation (season cycle) by fitting amplitude and phases of a Fourier series that considers all frequencies between 1 and $p$. We consider frequencies up to $2 yr^{-1}$ ($P=2$) and
$\Delta j = \max\{j(t)\} - \min\{j(t)\}$ as the intra-annual Julian day period covered by the data, and $j(t)$ the intra-annual Julian day (intra-annual Julian day means Julian day starting each year with 0, i.e, $j(t)$ is between 0 and 366). Note that $N = \frac{\Delta t}{\Delta j}$.

The uncertainty ranges of the fit parameters, including the linear trends, is calculated using the bootstrap resampling method (Gardiner et al., 2008). This approach is based on recurrently estimating the fit parameters on a modified time series, which





results from randomly disturbing the original time series with the residues between the multi-regression fit and the original
time series. Thereby, it does not assume that residues follow any specific distribution (i.e. Gaussian) and are uniform over time,
which could occur if the modeled fit is not able to properly capture the measured time series.

This multi-regression fit has been also used to temporally decompose the FTIR and reference time series, when the quality
assessment of the FTIR products is addressed (Section 8). For the hourly and daily means comparison (so-called measurement-
to-measurement), we work with the de-seasonalised and de-trended time series in order to ensure that the comparison between
the measured inter-day variabilities is not affected by the seasonal and long-term signals. This time series is calculated by
subtracting from the measured time series (reference or FTIR) the corresponding linear trend, inter-annual and intra-annual
signals obtained from the multi-regression fit. For the seasonal comparison, an averaged annual cycle is computed from the
multi-annual averaged monthly means of the de-trended time series (measured time series minus the linear trend and the
inter-annual variations). Finally, the long-term signals given by the annual means are computed from the de-seasonalised time
series, which is computed by subtracting the intra-annual variations from the measured time series. Note that to obtain the linear
trends in percentage the measured time series is transformed on a logarithmic scale. Since the trace gas short-term variations
are usually much smaller than the climatological or long-term background values, the variations on the logarithmic scale can
be interpreted as the variations relative to the long-term background reference.

*Data availability.* The NDACC FTIR and DOAS products, as well as the ozone sondes, are available from the NDACC archive (www.ndaccdemo.org).
The TCCON FTIR data are accessible via the TCCON Data Archive, hosted by CaltechDATA (https://tccondata.org). The CIMEL PWV data
can be downloaded from AERONET database (https://aeronet.gsfc.nasa.gov/), while the water vapour sondes are available from the GRUAN
archive (www.gruan.org). The Brewer ozone and GPS PWV data are available by request from the corresponding authors. The ground-
level greenhouse and carbon cycle gases are archived in the World Data Centre for Greenhouse Gases (WDCGG) under the WMO-GAW
programme (https://gaw.kishou.go.jp/).

*Author contributions.* O.G., and M.S. designed and wrote the structure and methodology of the paper, and computed the calculations re-
quired. F.H., T.B., J.G., S.B., and A.R. discussed the results and provided the NDACC and TCCON retrievals. E.C. and T.B. ensured the
provision of funds for the IZO FTIR measurement programme. O.G., M.S. E.S., E.S., C.L., and Y.G. performed the maintenance and quality-
control of the FTIR measurements. R.R., A.G.-P., E.R., P.P.R., and R.D.G. are responsible for the maintenance, quality-control, and data
generation of the WMO-GAW greenhouse gases programme at IZO. M.N., O.P., and M.Y. are in charge of the NDACC DOAS observations
at IZO. A.R., V.C., and S.L.-L. are responsible for the maintenance of RBCC-E Brewer spectrometers as well as for providing the NDACC
Brewer ozone observations. P.R.-C. and M.H. are responsible for the PWV data and GRUAN humidity profiles, respectively. C.T. and N.P.
are in charge of the ozone sonde programme at IZO. Finally, all authors discussed the results and contributed to the final paper.

*Competing interests.* The authors declare no conflict of interest.





*Acknowledgements.* The Izaña FTIR station has been supported by the German Bundesministerium für Wirtschaft und Energie (BMWi)
via DLRunder grants 50EE1711A and by the Helmholtz Society via the research program ATMO. In addition, this research was funded by
the European Research Council under FP7/(2007-2013)/ERC Grant agreement nº256961 (project MUSICA), by the Deutsche Forschungs-
gemeinschaft for the project MOTIV (GeschaFTIRzeichen SCHN 1126/2-1), by the Ministerio de Economía y Competitividad from Spain
through the projects CGL2012-37505 (project NOVIA) and CGL2016-80688-P (project INMENSE), and by EUMETSAT under its Fel-
lowship Programme (project VALIASI). This work has been developed within the framework of the activities of the World Meteorological
Organization (WMO) Commission for Instruments and Methods of Observation (CIMO) Izaña test bed for aerosols and water vapour remote
sensing instruments.





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
