# Peer review of "Twenty years of ground-based NDACC FTIR spectrometry at Izaña Observatory - overview and long-term comparison to other techniques"

_Atmospheric Chemistry and Physics, 2021_

## Author Comment (AC1)

**Comment on acp-2021-307**

**Anonymous Referee #1**

**Referee comment on "Twenty years of ground-based NDACC FTIR spectrometry at Izaña Observatory – overview and long-term comparison to other techniques" by Omaira E. García et al., Atmos. Chem. Phys. Discuss., https://doi.org/10.5194/acp-2021-307-RC1, 2021**

**Response to Referee#1**

The authors would like to thank the Referee for reviewing carefully this paper and provide valuable and constructive comments that have improved this work. In the following text, the Referee suggestions (in blue italics) are addressed in detail (the authors' responses are in plain text).

*This is an extensive and well written review of the measurements and analyses programs in place at the Izaña Observatory over the past twenty years. It is a review paper, differing from most in that the focus is on the FTIR measurements at this site rather than a broad overview of some subject matter. This is not a problem as the measurement and science record is extensive. This paper and the references therein provide a useful compendium of the research carried out at Izaña by the FTIR group and their colleagues, and it will provide a good starting point for researchers looking to understand the context of the work carried out there. It discusses some of the more significant results and provides the references to the supporting papers should a more in-depth look be of interest to the reader. The assessment of instrument performance over the 20 years is comprehensive and fairly well presented. It is difficult to judge this paper in terms of "substantial new concepts, ideas, methods, or data" as presented herein, but nevertheless I feel the paper does have merit in presenting in one place the extensive data record and the use of that data record to address many of the more prominent areas of atmospheric research.*

*Some general remarks:*

*The paper refers to the IFS120/5 HR. Is this to denote that the instrument is of the family of the IFS120HR and the later IFS125HR, or is it to denote that it entered service as an IFS120HR and was later modified to the specification of the newer IFS125HR?*

The denotation of the second Izaña FTIR instrument as IFS 120/5HR is due to the interferometer belongs indeed to the family of the IFS120HR, however, before installing at Izaña observatory (IZO), it was upgraded to the IFS125HR electronics. This clarification will be included in Section 3.1 of the revised manuscript.

*With some of the figures that have many data points, the use of circles and filled circles (dots) of similar or the same colour can be problematic. It would be better to use clearly different symbols. Looking specifically at Figure 6, the use of gray circles and black circles is not a good idea.*

The authors agree with the Referee in that the symbols and colours used in some figures can make them difficult to evaluate. Therefore, the format of Figure 4, 6, 8, 14 and 15 will be modified, following Referee#1's comments as well as the suggestions from Referee#2 and #3.

*For Figure 2, could some indication of the mean number of measurements per day (binned per month) be given?*

The spectral range covered by the NDACC activities (700 and 4500 cm⁻¹) is routinely measured at IZO using six optical filters in a sequential manner (so-called SA, SB, SC, SD, SE, and SF or S1, S2, S3, S4, S5, and S6). Therefore, the number of measured spectra is highly variable depending on each NDACC filter, as illustrated in Table 3 of the preprint and Figure 1 below. This figure shows the monthly distribution of all FTIR observations taken from 1999 to 2018 for the abovementioned six NDACC filters. Usually, two measurements per day are routinely taken for each NDACC filter, except for SC, for which greater frequent observations are recorded given these spectra are used for water vapour and greenhouse gas studies (Schneider et al., 2012, 2016, García et al., 2016, and references therein).

The authors agree with the Referee in that this information would be useful for the paper and will be included in Section 3.1 of the revised manuscript. However, it will not be included in Figure 2 of the preprint to avoid making the figure more confusing.

[Figure]

Figure 1. Monthly distribution of all FTIR measurements taken at IZO from 1999 to 2018 for the six NDACC optical filters (SF-SA). The left axis corresponds to cumulative measurements per month, while the right axis represents monthly mean of all measurements.

*Line 120: Was the IFS 120/5HR operated in "vented mode" at all times, or just for the time period of the comparison with the IFS 120M?*

Both FTIR instruments have always been operated in ventilated mode given the especially dry conditions at IZO. This will be make clearer in the revised manuscript.

*Table 2: As formatted, this table is difficult to read. Is there a difference between the Target Gas and Gas columns? Perhaps the "Gas" should also be Target Gas. It would improve readability significantly if a few more spaces or a vertical bar were added to more clearly separate the left-hand grouping of gases from the right-hand grouping*

Effectively, the term "Gas" should be "Target Gas". It will be corrected in the revised Table 2, which will be formatted following the Referee's comment.

*Line 223: This would be more readable if it was written: This figure depicts the rows of the averaging kernel matrix A …"*

This statement will be modified following the Referee's suggestion.

*Table 3: For readability rather than spacing the columns equally, I would suggest adding more space between the columns for the combined pairs, and removing a space between the M and σ, so the first line would look like:*

*C2H6 13625   1.48 0.15    1.81 0.39    5.44 0.17*

Table 3 will be modified following the Referee's suggestion.

*Figure 3: As there are many similar averaging kernels, it would be of interest to see some explanation of why the particular kernel was selected for display in the figure.*

Figure 3 depicts, as an example, all the rows of the averaging kernel matrix (A) for all NDACC products for typical measurement conditions at IZO (spectra taken on 20$^{th}$ July 2013 at a solar zenith angle of $\sim$30º). Note that, for each trace gas, the A rows describe the altitude regions that mainly contribute to the retrieved profile and, therefore, the vertical distribution of the FTIR sensitivity. Given that the retrieved altitude information is not independent (Rodgers, 2000), only those A rows at altitudes representative of the layers discernible by the FTIR instrument were highlighted in Figure 3 as coloured lines. Therefore, for each trace gas (i.e. the frame in Figure 3) the number of highlighted A rows corresponds to the number of independent layers discernible by the remote sensing instrument, which is equivalent to the trace of A or Degrees Of Freedom for Signal (DOFS), as summarised in Table 3 of the preprint. This will be further explained in Section 3.4 of the revised manuscript.

*Line 236: Are "great values" to mean those values that show large statistical uncertainty?*

Yes, this statement refers those values that show large statistical uncertainty. It will be re-written in the revised manuscript to avoid confusions.

*Line 241: maximum instead of maximal*

This will be modified following the Referee's suggestion.

*Line 243: The systematic uncertainty budget is dominated by spectroscopic errors (instead of "led").*

This statement will be modified following the Referee's suggestion.

*Figure 4: The choice of lines and lines + circles in the same colour makes the plot difficult to evaluate for some constituents. Making the plot wider would help. Also, use just symbols for those constituents that have sufficient points and are fairly "straight" such that the lines are not necessary. H2CO and HCl are both shown in cyan with the former being a thinner line. I can't distinguish a difference in thickness in the lines in the plot.*

Figure 4 will be modified, according to the Referee's comments, to make it easier to evaluate.

*Line 250: Do the authors mean to say that the predominant errors are located in one of the troposphere, upper troposphere/lower stratosphere or middle/upper stratosphere, or that the errors are in all three regions? If the latter, they do not appear to be equally concentrated in the regions and that should be discussed.*

The estimated uncertainty vertical profiles are largely linked to the vertical distribution of each trace gas and FTIR vertical sensitivity, as shown in Figure 4 of the preprint. Therefore, depending on each gas, they are predominantly located in only one of the layers (troposphere, upper troposphere/lower stratosphere, or middle/upper stratosphere), but not in all three regions. This statement will be clarified in the revised manuscript.

*Line 257: Particularly, high error profiles … would read better as "large"error profiles unless the authors mean to suggest that there is also an altitude dependent component which does not seem to be the case as the lines are straight in Figure 4.*

Effectively, this statement should refer to large values in the estimated error profiles. It will be modified following the Referee's suggestion.

*Figure 6: Caption refers to 'grey-white dots" which is, I think, a grey circle that is not filled? See earlier comment about the use of circles and dots. Also the grey circle is easily confused with a black circle. Note also that the RXCO2 quantity is used here but not yet defined in the text.*

The term "grey-white dots" refers to grey circles with a white centre. The symbols and their colours will be changed in Figures 4 and 6 in order to make them clearer, following the Referee's comments.

In addition, $R_{XCO2}$ will be defined in the caption of Figure 6.

*Line 312: I don't think that 'considering' is the proper word here. I would suggest 'using'. For example: "… using a scaling retrieval with a fixed WACCM a piori VMR profile, and PROFFIT software."*

This statement will be modified following the Referee's suggestion.

*Line 366: At the resolution the data are plotted with, it is not possible to discern a seasonal cycle in Xair N2 . If the authors consider this to be a significant finding, then overlay a trace on the data that would display it. Otherwise, the statement could be left out without weakening the section.*

Effectively, the Xair $N_2$ seasonal cycle is not clearly discernible in Figure 6. However, to avoid make this figure more confusing, the authors prefer to keep it as is and remove this statement in the discussion.

*Line 384: "… causing punctual downward and upward shift of the UTLS region,…" The use of punctual doesn't make much sense. Punctual implies arriving on schedule. Is that what the authors mean to say?*

Effectively, the term "punctual" has been wrongly used here. It will be replaced by "sporadic".

*Line 390/Figure 7: It would be a good idea to move the HF and N2O frames to be next to each other to make the anti-correlation of the two easier to detect.*

The frames of the trace gases are displayed in alphabetical order in Figure 7. However, as the Referee suggests, placing the HF and $N_2O$ time series together could help reader to better follow the discussion. This will be changed in the revised manuscript.

*Line 559: There is a set of ellipses between CCMVal initiative and Schneider et al. Are there more missing from the list? There are 8 models listed and 10 papers referenced. It might make it clearer to list the CCM followed directly by the reference.*

As far as the authors know, there are no more studies dealing with the evaluation and development support of global atmospheric chemistry climate models at Izaña observatory, thereby the ellipsis will be removed in the revised manuscript and the statement will be modified following the Referee's suggestion.

*Lines 561-562: This is an awkward and confusing sentence and the details are lacking. There is mention of "attribution of sources/sinks", but no comment on whether or not there is agreement. Similarly for the "representation of moist processes" evaluation. As the paper is already quite long, it might be best to make a general statement about the comparison to the CCMs and referring the reader to the above references for details rather than to try to call out these areas without further discussion.*

The authors agree with the Referee in that this explanation is vague and does not provide helpful information. Therefore, it will be removed in the revised manuscript.

*Line 693 and below: Generally, when speaking of time the word used is coincident rather than collocation.*

The word "collocation" will be changed by "coincident" following the Referee's suggestion.

*Line 690: It is unclear whether a actual single temporal criterion is applied here as multiple gases are being discussed. Sentence should read either "Similar temporal criteria …" or "A similar temporal criterion…." as the authors see fit.*

For $CH_4$, CO and $N_2O$, the same temporal criterion has been applied for both NDACC-GAW and NDACC-TCCON comparisons (i.e. matching the daily means). This statement will be changed following the Referee's suggestion.

*Line 699: Over what time interval are the FTIR observations averaged? Are the total columns derived from a series of sequential spectra, averaged?*

For the DOAS-FTIR intercomparison, all available FTIR measurements taken before and after 12 UT were averaged with the only restriction of not considering FTIR data at very high solar elevation angles (≥85°) to avoid imprecise retrievals (mainly caused by misalignments of the solar tracker or spectroscopic issues) (see Section 3.3 of the preprint). As shown in Figure 2 below for $O_3$, the FTIR measurements at IZO are mostly concentrated (≥90% of the total observations) in the interval 8:00-16:00 UT, i.e., ±4 hours around noon. Therefore, the latter can be considered as a representative time interval for FTIR averages.

The FTIR total column means are computed from single total column retrievals, not from an averaged solar spectra.

[Figure]

Figure 2. Hourly distribution of the FTIR $O_3$ measurements taken at IZO in the period 1999-2018. The number of measurements (left axis) and the cumulative percentage (right axis) are shown.

*Line 703: Need a comma after "correction"*

This statement will be modified following the Referee's suggestion.

*Line 706: If I understand what is being said, this might read better as "... each FTIR measurement is only paired once to the reference observation that minimizes the time difference within the temporal collocation window." And I would use coincidence rather than collocation.*

This statement will be modified following the Referee's suggestion.

*Figure 14: Presumably the "TRO" quantities refer to the tropospheric portions?*

Effectively, the term "TRO" refers to the comparison of the tropospheric quantities (GAW in-situ records and FTIR VMR averages). This clarification will be included in the caption of Figure 14.

*Line 785: In what manner are the PWV values over IZO "slowing down"?*

Both climate models and observations suggest that an upward trend in water vapour is expected to appear as a response to the surface temperature increase (Chen and Liu, 2016, and references therein). Nevertheless, it has also been shown that water vapour scales not everywhere to temperature as expected and that large regional differences exist (Bernet et al., 2020, and references therein). Over continental areas, correlations between surface temperature and integrated water vapour changes are smaller than over oceans, showing in some regions even opposite trends (Wagner et al., 2006). In this sense, as mentioned above, albeit the net response of climate system points to an increase in the atmospheric water vapour concentrations, numerous studies report large trend variabilities, especially at a regional scales (e.g. Nilsson and Elgered, 2008; Alshawaf et al., 2017, Bernet et al., 2020). Another key factor largely contributing

to trend variabilities is altitude (Bernet et al., 2020). Given the less water vapour content present at higher altitudes, the trend estimation is more sensitive to uncertainties.

The FTIR water vapour observations at IZO suggest that water vapour total column amounts have been significantly slowing down over the last two decades. This finding is consistent with the downward trends obtained from coincident CIMEL and GPS measurement techniques. This progressive loss of humidity in the subtropical free troposphere might be associated with the expansion of the tropical belt, meaning that the descending limb of the Hadley cells is shifting towards the poles in both hemispheres (Heffernan, 2016, and references therein). As a consequence of this poleward movement, their associated subtropical dry zones are expected to move as well (Seidel et al., 2008). In addition, dynamical variability of atmospheric transport circulation may affect atmospheric composition of subtropical regions and influence trend assessments (Li et al., 2009; Strahan et al. 2020). Although dedicated studies would be of great use in better understanding these drivers and connections on short-term and long-term scales, they are beyond the scope of this review work.

Finally, the authors should also admit that, although the estimated water vapour trends are found to be statistically significant, the period analysed is relatively short to draw robust conclusions for a trace gas, like water vapour, with a so large spatial and temporal variability. These results will be revisited as the water vapour records at IZO extend over time.

*Line 823: Do the authors mean to say that the NDACC FTIR product is able to capture "only" a part of its tropospheric variations?*

The timescale analysis presented in the current work is a very powerful tool to examine what temporal signals are captured by the FTIR observations, and to what extent. For example, as illustrated in Figure 14 of the preprint, the agreement observed between ground-level GAW records and tropospheric NDACC products is mainly the result of seasonal and long-term signals. Therefore, the NDACC FTIR products are able to properly monitor the tropospheric variations at those time scales. On a daily basis, the correlation found is almost nil for long-lived gases, such as $CH_4$ and $N_2O$, likely due to their daily variations being smaller than the FTIR precision. However, for CO, the agreement obtained significantly increases, leading to the NDACC FTIR CO product also being able to capture significant information of tropospheric CO signals on a daily basis.

*Line 842: What is meant by "pure free conditions"?*

This statement was included to point out that the comparison between radiosonde humidity and FTIR profiles is carried out for those layers where the subtropical free tropospheric conditions are reached. As discussed in detail in Schneider et al. (2016), a fair comparison of in-situ profile measurements (free troposphere) with ground-based FTIR measurements made at IZO (on a mountain ridge) is difficult due to the local thermal circulation that starts on the island during the morning hours. To ensure that both measurement techniques are detecting air masses with the same atmospheric characteristics and representative of the free tropospheric signals, these authors found the optimal coincidences correspond to those FTIR observations taken at low solar elevation angles (between 25º and 45º). For that reason, this criterion was applied in the current work.

However, we agree with the Referee in that this short statement as it is in the preprint is vague and confusing. Therefore, it will be modified as follows: "To examine this effect and ensure the comparison is carried out for free tropospheric conditions, the FTIR observations have also been

restricted to low solar elevation angles (between 25º and 45º) (Schneider et al., 2016), with the resulting difference profile also included in Figure 15 (in red)."

*Line 845: I suggest the wording be: Despite a considerable decrease in the number of coincidences, …*

This statement will be modified following the Referee's suggestion.

*Line 849: I suggest the wording: ...which makes the comparison of the remote sensing and in-situ profiles difficult.*

This statement will be modified following the Referee's suggestion.

**References**

Alshawaf, F., Balidakis, K., Dick, G., Heise, S., and Wickert, J.: Estimating trends in atmospheric water vapor and temperature time series over Germany, Atmos. Meas. Tech., 10, 3117–3132, https://doi.org/10.5194/amt-10-3117-2017, 2017.

Bernet, L., Brockmann, E., von Clarmann, T., Kämpfer, N., Mahieu, E., Mätzler, C., Stober, G., and Hocke, K.: Trends of atmospheric water vapour in Switzerland from ground-based radiometry, FTIR and GNSS data, Atmos. Chem. Phys., 20, 11223–11244, https://doi.org/10.5194/acp-20-11223-2020, 2020.

Chen, B., and Z. Liu, Global water vapor variability and trend from the latest 36 year (1979 to 2014) data of ECMWF and NCEP reanalyses, radiosonde, GPS, and microwave satellite, J. Geophys. Res. Atmos., 121,11,442–11,462, doi:10.1002/2016JD024917, 2016.

García, O. E., Sepúlveda, E., Schneider, M., Hase, F., August, T., Blumenstock, T., Kühl, S., Munro, R., Gómez-Peláez, A. J., Hultberg, T., Redondas, A., Barthlott, S., Wiegele, A., González, Y., and Sanromá, E.: Consistency and quality assessment of the Metop-A/IASI and Metop-B/IASI operational trace gas products (O3, CO, N2O, CH4, and CO2) in the subtropical North Atlantic, Atmospheric Measurement Techniques, 9, 2315–2333, https://doi.org/10.5194/amt-9-2315-2016, 2016.

Heffernan, O.: The mystery of the expanding tropics, Nature, 530, 20–22, https://doi.org/10.1038/530020a, 2016.

Li, Q., Palmer, P. I., Pumphrey, H. C., Bernath, P., and Mahieu, E.: What drives the observed variability of HCN in the troposphere and lower stratosphere?, Atmos. Chem. Phys., 9, 8531–8543, https://doi.org/10.5194/acp-9-8531-2009, 2009.

Nilsson, T. and Elgered, G.: Long-term trends in the atmospheric water vapor content estimated from ground-based GPS data, J. Geophys. Res. Atmos., 113, https://doi.org/10.1029/2008JD010110, 2008.

Rodgers, C.: Inverse Methods for Atmospheric Sounding: Theory and Praxis, World Scientific Publishing Co., Singapore, 2000.

Schneider, M., Barthlott, S., Hase, F., González, Y., Yoshimura, K., García, O. E., Sepúlveda, E., Gomez-Pelaez, A., Gisi, M., Kohlhepp, R., Dohe, S., Blumenstock, T., Wiegele, A., Christner, E., Strong, K., Weaver, D., Palm, M., Deutscher, N. M., Warneke, T., Notholt, J., 1210 Lejeune, B., Demoulin, P., Jones, N., Griffith, D. W. T., Smale, D., and Robinson, J.: Ground-based remote sensing of tropospheric water vapour isotopologues within the project MUSICA, Atmospheric Measurement Techniques, 5, 3007–3027, https://doi.org/10.5194/amt-5- 3007-2012, 2012

Schneider, M., Wiegele, A., Barthlott, S., González, Y., Christner, E., Dyroff, C., García, O. E., Hase, F., Blumenstock, T., Sepúlveda, E., Mengistu Tsidu, G., Takele Kenea, S., Rodríguez, S., and Andrey, J.: Accomplishments of the MUSICA project to provide accurate, long-term, global and high-resolution observations of tropospheric {H2O,δD} pairs – a review, Atmospheric Measurement Techniques, 9, 2845–2875, https://doi.org/10.5194/amt-9-2845-2016, 2016.

Seidel, D., Fu, Q., Randel, W., and Reichler, T.: Widening of the tropical belt in a changing climate, Nature Geoscience, 1, 21–24, https://doi.org/10.1038/ngeo.2007.38, 2008.

Strahan, S. E., Smale, D., Douglass, A. R., Blumenstock, T., Hannigan, J. W., Hase, F., Jones, N., Mahieu, E., Notholt, J., Oman, L. D., Ortega, I., Palm, M., Prignon, M., Robinson, J., Schneider, M., Sussmann, R., and Velazco, V.: Observed hemispheric asymmetry in stratospheric transport trends from 1994 to 2018, Geophysical Research Letters, 47, https://doi.org/10.1029/2020GL088567, 2020.

Wagner, T., Beirle, S., Grzegorski, M., and Platt, U.: Global trends (1996-2003) of total column precipitable water observed by Global Ozone Monitoring Experiment (GOME) on ERS-2 and their relation to near-surface temperature, J. Geophys. Res. Atmos., 111, D12 102, 740 https://doi.org/10.1029/2005JD006523, 2006

---

## Author Comment (AC2)

**Comment on acp-2021-307**

**Anonymous Referee #3**

**Referee comment on "Twenty years of ground-based NDACC FTIR spectrometry at Izaña Observatory – overview and long-term comparison to other techniques" by Omaira E. García et al., Atmos. Chem. Phys. Discuss., https://doi.org/10.5194/acp-2021-307-RC2, 2021**

**Response to Referee#3**

The authors would like to thank the Referee for reviewing carefully this paper and provide valuable and constructive comments that have improved this work. In the following text, the Referee suggestions (in blue italics) are addressed in detail (the authors' responses are in plain text).

*"Twenty years of ground-based NDACC FTIR spectrometry at Izana Observatory - overview and long-term comparison to other techniques" Omaria Garcia et al., 2021*

*The paper meticulously describes the methodology of FTIR retrievals and draws on previous work in the field. The writing style is clear and descriptive. It gives a very detailed description of the 20-years FTIR-related measurements from 120/5 HR at Izana. Such kind of overview paper is valuable for the global users (modelers, satellite developers, atmospheric scientists... ) to use their data. Izana is located in the subtropical region, which is crucial to understand the change of atmospheric compositions. The 20-years FTIR measurements have already been used in many scientific studies, leading to more than 100 peer-reviewed papers. Overall, I recommend this paper to publish in ACP, and I only have a few minor comments:*

*P8 line 182: WACCM model used in NDACC-IRWG is v4 instead of v6*

As far as the authors know, the NDACC-IRWG guideline recommends the versions 5 or 6 of the climatological WACCM model, depending on the target gas, to be used as a priori information (IRWG, 2014). In this respect, the authors would appreciate it if the Referee could provide us another reference supporting the usage of WACCM version 4.

*P8 line 188: why only use the temperature and pressure at 12:00 UT? How about the H2O? The temperature and H2O variation can be very large even on one day. Would you like to address such uncertainty on your retrievals?*

As shown in Figure 1 below for $O_3$, the FTIR measurements at Izaña observatory (IZO) are mostly taken around noon. In particular, more than 90% of the total observations during the 1999-2018 are concentrated in the interval 8:00-16:00 UT, i.e., ±4 hours around the NCEP temperature and pressure profiles used as reference in the NDACC FTIR retrievals (12 UT). Therefore, the 12 UT NCEP profiles can be considered a reliable proxy of the atmospheric state at IZO for the radiative transfer calculations. Nevertheless, as the Referee suggests, greater frequent NCEP profiles might improve the overall quality of FTIR retrievals, and it will be taken into consideration in the next re-evaluation of the NDACC IZO database that is expected to be carried out in 2021/2022. In this sense, a previous work analysing the effect of the intra-day variability of the pressure and temperature profiles (3-hourly profiles) on different FTIR products at IZO found that

the differences among FTIR products did not show significant dependence on the altitude (±0.5%), except for $H_2O$, for which the differences are overall within ±2% (García et al., 2014a). These values are overall within the estimated theoretical uncertainty of the FTIR products.

[Figure]

Figure 1. Hourly distribution of the FTIR $O_3$ measurements taken at IZO in the period 1999-2018. The number of measurements (left axis) and the cumulative percentage (right axis) are shown.

Regarding water vapour, its treatment in the operational NDACC FTIR retrievals strongly depends on its interference on the NDACC target gas and, therefore, it varies from gas to gas. For example, the NDACC $CH_4$ products are retrieved following Sepúlveda et at. (2012, 2014) that proposes six micro-windows, which contain strong, not saturated, and well-isolated $CH_4$ absorption lines as well as $H_2O$ lines, in order to better account for the $H_2O$ interferences. The $H_2O$ profiles are simultaneously retrieved with $CH_4$ using a dedicated profiling retrieval. As documented by Hase et al. (2011), the Sepúlveda approach could be less dependent on humidity conditions as it minimises perturbing $H_2O$ absorptions and it handles the problematic interference species $H_2O$/HDO in a rigorous manner. For other gases for which the $H_2O$ interference shows a minor impact, such as $N_2O$, the $H_2O$ profile is simultaneously scaled from the WACCM climatological profiles during the inversion procedure. Finally, for other gases, if possible, the spectral regions used for retrievals are selected so that $H_2O$ absorption lines are not presented like for $O_3$ (García et al., 2021).

*P9 table 2. do you want to add N2 also here?*

Given that Table 2 only lists those FTIR products available at the NDACC archive, the authors prefer not to include $N_2$ in this table and keep it as an auxiliary product to test the long-term performance of FTIR observations in Section 4.

*P14 line 261: "the total column-averaged amount of dry air (Xair) " is not appropriate. Xair is the ratio of o2 or n2 derived dry air to DPC, please use a better definition here.*

Effectively, as stated by the Referee, the Xair parameter is computed as the ratio of the total column of $O_2$ or $N_2$ to DPC. Nevertheless, the authors have adopted the definition of Xair widely used within the TCCON community, in which Xair is defined as the column-averaged amount of dry air (e.g. Wunch et al., 2011, 2015; Pollard et al., 2017; Frey et al., 2019). The authors would

appreciate it if the Referee could provide us another valid definition or reference for Xair parameter.

*P22 Figure 8. in the bottom panel, are you sure the colors are correct? because the CO is increasing, but you mention that in P23 line 464 that the CO is decreasing.*

Effectively, the coloured line assigned to the CO time series is wrong in Figure 8. The CO and OCS lines were exchanged. The figure will be corrected accordingly in the revised manuscript.

*P22 Figure 8 in the middle panel and Figure 11, I see that the CH4 and N2O long-term trends are similar. However, other in-situ measurements show that N2O is increasing continuously while the annual growth of CH4 is variable: 1999-2007 stable, and reincreased after 2007 (https://gml.noaa.gov/ccgg/trends_ch4/). Any explanation here? Why we get a different CH4 tend from Izana FTIR CH4 measurements, especially between 1999 and 2007, compared to other surface measurements?*

Figure 2 below shows the time series of the NDACC FTIR $CH_4$ products (total columns and tropospheric VMR means) along with the corresponding annual anomalies relative to the 1999-2018 background. Note that the anomalies are the same as those displayed in Figure 8 and Figure 11 of the preprint. The GAW ground-level $CH_4$ records have also been included for a better comparison with the FTIR tropospheric observations. These plots show more clearly the variable $CH_4$ annual growth remarked by the Referee: constant until about 2005-2007 and increasing after 2005-2007. As reported in the preprint, the NDACC IZO $CH_4$ data confirms a speed-up in the emission rates for $CH_4$ in the last decade, which is likely caused by the increase in anthropogenic emissions. This acceleration is found to be of $+0.13\pm0.03\%\mathrm{yr}^{-1}$ and $+0.43\pm0.03\%\mathrm{yr}^{-1}$ for the periods 1999-2008 and 2009-2019, respectively. In addition, as documented in Figure 2 below and in the detailed comparison results included in the current work (Section 8.3), there is excellent agreement between GAW and NDACC FTIR tropospheric data (excluding the well-known bias of about 2% in the FTIR products). In fact, as further illustrated in Section 8.4 of the preprint, the uneven sampling of the FTIR system results in a negligible effect on $CH_4$ trend estimations.

These findings are in total agreement with the results presented by Bader et al. (2017), who evaluated changes of $CH_4$ total columns since 2005 using FTIR observations carried out at ten globally-distributed NDACC sites (IZO among them). Particularly for IZO, they documented a close to statistical agreement with a mean annual increase of $0.33\pm0.01$ and $0.28\pm0.02$ % year$^{-1}$ for the NDACC FTIR total columns and GAW ground-level records, respectively, for the period 2005-2014. These values were close to those reported considering the ten NDACC sites, $0.31\pm0.03$ % year$^{-1}$, and the GEOS-Chem $CH_4$ simulations included in Bader's work as well.

Nevertheless, the authors should admit that the change of FTIR instrument in 2005 at IZO could slightly affect the change point (2005-2007) between the well-known steady and increasing evolution of $CH_4$. As illustrated in the bottom panel of Figure 2, there is a small jump in 2005 preceded and followed by a flat period between 1999-2005 and 2005-2007, respectively. Although this potential instrumental issue does not influence the trend estimations presented in the current work, as mentioned above, it will be analysed in detail in future studies.

[Figure]

Figure 2. Upper panel: time series of NDACC FTIR CH₄ total columns (left axis) and of annual anomalies relative to the 1999-2018 background (right axis). Bottom panel: time series of NDACC FTIR tropospheric CH₄ VMR and GAW ground-level records (left axis) and of annual anomalies relative to the 1999-2018 background (right axis).

*P39 line 811, the tropospheric XCH4 is compared to the surface measurements to found a bias of ~2.6%. I do not support such direct comparison, as they are sampling different vertical ranges still.*

Effectively, as stated by the Referee, when comparing different measurement techniques, there are multiple factors affecting the comparison, namely, approaches, sampled air masses, observing geometries, spectral ranges, spectral resolution, retrieval strategies, vertical sensitivity, etc. All these aspects can introduce significant differences and must be considered when interpreting the comparison results. This was briefly discussed in the current work in Section 8.2.

The authors assume that the Referee is referring to the comparison between the NDACC tropospheric CH₄ product and GAW ground-level records, albeit the Referee mentioned XCH₄. The latter is the term used to refer to the total column-averaged CH₄ mole fraction.

The NDACC tropospheric FTIR and ground-level observations can indeed be representative of different air masses (i.e. different vertical ranges). However, the comparison methodology used in the current work was designed to minimise these potential impacts. On the one hand, the NDACC tropospheric concentrations are obtained as the mean of retrieved VMR profiles between IZO altitude (2.37 km a.s.l.) and middle troposphere (5.6 km a.s.l.). This layer has proved to represent well the tropospheric signal detectable by the FTIR system (Sepúlveda et al., 2012, 2014; García et al., 2014b). On the other hand, the daily nighttime means (20.00-08.00 UT) of the IZO ground-level records and the daily daytime means of the FTIR products are paired. As mentioned in the preprint, given the strategic location of IZO, diurnal insolation generates a slight up-slope flow of air originating from below the inversion layer that can disturb the free troposphere conditions at IZO. However, during nighttime, the subsidence regime typical for subtropical regions prevails and the atmospheric observations taken at IZO are well-representative of the subtropical North Atlantic free troposphere (Cuevas et al., 2019, and references therein). Under these conditions,

the ground-level records are well comparable to the FTIR observations in the lower troposphere (Sepúlveda et al., 2012; 2014; García et al., 2014b, 2018). In particular, for $CH_4$, the bias of 2.6% of the NDACC FTIR products related to GAW ground-level data found at IZO is consistent with those reported at different NDACC FTIR stations ranging different latitudes and altitudes (Sepúlveda et al., 2014). Therefore, the authors presume that it is likely due to spectroscopic parameter issues in the mid-infrared spectral region, not being introduced by the comparison approach.

*P39 line 814, the reference "Zhou et al., 2019" is wrong. It should be https://amt.copernicus.org/articles/12/5979/2019/*

This reference will be changed following the Referee's comment.

**References**

Bader, W., Bovy, B., Conway, S., Strong, K., Smale, D., Turner, A. J., Blumenstock, T., Boone, C., Collaud Coen, M., Coulon, A., Garcia, O., Griffith, D. W. T., Hase, F., Hausmann, P., Jones, N., Krummel, P., Murata, I., Morino, I., Nakajima, H., O'Doherty, S., Paton-Walsh, C., Robinson, J., Sandrin, R., Schneider, M., Servais, C., Sussmann, R., and Mahieu, E.: The recent increase of atmospheric methane from 10 years of ground-based NDACC FTIR observations since 2005, Atmos. Chem. Phys., 17, 2255–2277, https://doi.org/10.5194/acp-17-2255-2017, 2017.

Cuevas, E., Milford, C., Bustos, J. J., R., García, O. E., García, R. D., Gómez-Peláez, A. J., Guirado-Fuentes, C., Marrero, C., Prats, N., Ramos, R., Redondas, A., Reyes, E., Rivas-Soriano, P. P., Rodríguez, S., Romero-Campos, P. M., Torres, C. J., Schneider, M., Yela, M., 1005 Belmonte, J., del Campo-Hernández, R., Almansa, F., Barreto, A., López-Solano, C., Basart, S., Terradellas, E., Werner, E., Afonso, S., Bayo, C., Berjón, A., Carreño, V., Castro, N. J., Chinea, N., Cruz, A. M., Damas, M., De Ory-Ajamil, F., García, M., Gómez-Trueba, V., Hernández, C., Hernández, Y., Hernández-Cruz, B., León-Luís, S. F., López-Fernández, R., López-Solano, J., Parra, F., Rodríguez, E., Rodríguez-Valido, M., Sálamo, C., Sanromá, E., Santana, D., Santo Tomás, F., Sepúlveda, E., and Sosa, E.: Izaña Atmospheric Research Center Activity Report 2017-2018, Eds. Cuevas, E., Milford, C. and Tarasova, O., State Meteorological Agency (AEMET), Madrid, Spain, 1010 and World Meteorological Organization (WMO), Geneva, Switzerland, WMO/GAW Report No. 247, 2019.

Frey, M., Sha, M. K., Hase, F., Kiel, M., Blumenstock, T., Harig, R., Surawicz, G., Deutscher, N. M., Shiomi, K., Franklin, J. E., Bösch, H., Chen, J., Grutter, M., Ohyama, H., Sun, Y., Butz, A., Mengistu Tsidu, G., Ene, D., Wunch, D., Cao, Z., Garcia, O., Ra1035 monet, M., Vogel, F., and Orphal, J.: Building the COllaborative Carbon Column Observing Network (COCCON): long-term stability and ensemble performance of the EM27/SUN Fourier transform spectrometer, Atmospheric Measurement Techniques, 12, 1513–1530, https://doi.org/10.5194/amt-12-1513-2019, 2019.

García, O.E., M. Schneider, F. Hase, T. Blumenstock, E. Sepúlveda, E. Cuevas, A. Redondas, A. Gómez-Peláez and J.J. Bustos, Effect of updates in LINEFTIT and PROFFIT at Izaña ground-based FTIR: improvement of the ILS characterisation and intra-day pressure and temperature variability, NDACC-IRWG/TCCON meeting 2014, 12-16 May, Bad Sulza (Germany), 2014a.

García, O. E., Schneider, M., Hase, F., Blumenstock, T., Sepúlveda, E., Gómez-Peláez, A., Barthlott, S., Dohe, S., González, Y., Meinhardt, F., and Steinbacher, M.: Monitoring of N2O by ground-based FTIR: optimisation of retrieval strategies and comparison to GAW insitu observations, in: NDACC-IRWG/TCCON meeting 2014, Bad Sulza, Germany, 2014b.

García, O. E., Schneider, M., Ertl, B., Sepúlveda, E., Borger, C., Diekmann, C., Wiegele, A., Hase, F., Barthlott, S., Blumenstock, T., Raffalski, U., Gómez-Peláez, A., Steinbacher, M., Ries, L., and de Frutos, A. M.: The MUSICA IASI CH4 and N2O products and their comparison to HIPPO, GAW and NDACC FTIR references, Atmospheric Measurement Techniques, 11, 4171–4215, https://doi.org/10.5194/amt11-4171-2018, 2018.

García, O. E., Sanromá, E., Schneider, M., Hase, F., León-Luis, S. F., Blumenstock, T., Sepúlveda, E., Redondas, A., Carreño, V., Torres, C., and Prats, N.: Improved ozone monitoring by ground-based FTIR spectrometry, Atmos. Meas. Tech. Discuss., https://doi.org/10.5194/amt-2021-67, in review, 2021.

Hase, F., Blumenstock, T., Schneider, M., and Sepúlveda, E. Interactive comment on "Strategy for high-accuracy-and-precision retrieval of atmospheric methane from the mid-infrared FTIR network" by R. Sussmann et al., Atmos. Meas. Tech. Discuss., 4, C1048–C1048, 2011.

IRWG: Infrared Working Group Uniform Retrieval Parameter Summary, Tech. rep., http://www.acom.ucar.edu/irwg/IRWG_Uniform_RP_Summary-3.pdf, 2014.

Pollard, D. F., Sherlock, V., Robinson, J., Deutscher, N. M., Connor, B., and Shiona, H.: The Total Carbon Column Observing Network site description for Lauder, New Zealand, Earth Syst. Sci. Data, 9, 977–992, https://doi.org/10.5194/essd-9-977-2017, 2017.

Sepúlveda, E., Schneider, M., Hase, F., García, O. E., Gómez-Peláez, A., Dohe, S., Blumenstock, T., and Guerra, J. C.: Long-term validation of tropospheric column-averaged CH4 mole fractions obtained by mid-infrared ground-based FTIR spectrometry, Atmospheric Measurement Techniques, 5, 1425–1441, https://doi.org/10.5194/amt-5-1425-2012, 2012.

Sepúlveda, E., Schneider, M., Hase, F., Barthlott, S., Dubravica, D., García, O. E., Gómez-Peláez, A., González, Y., Guerra, J. C., Gisi, M., Kohlhepp, R., Dohe, S., Blumenstock, T., Strong, K., Weaver, D., Palm, M., Sadeghi, A., Deutscher, N. M., Warneke, T., Notholt, J., Jones, N., Griffith, D. W. T., Smale, D., Brailsford, G. W., Robinson, J., Meinhardt, F., Steinbacher, M., Aalto, T., and Worthy, D.: Tropospheric CH4 signals as observed by NDACC FTIR at globally distributed sites and comparison to GAW surface in situ measurements, Atmospheric Measurement Techniques, 7, 2337–2360, https://doi.org/10.5194/amt-7-2337-2014, 2014.

Wunch, D., Toon, G. C., Blavier, J.-F. L., Washenfelder, R. A., Notholt, J., Connor, B. J., Griffith, D. W. T., Sherlock, V., and Wennberg, 1300 P. O.: The total carbon column observing network, Philosophical Transactions of the Royal Society - Series A: Mathematical, Physical and Engineering Sciences, https://doi.org/doi:10.1098/rsta.2010.0240, 2011.

Wunch, D., Toon, G. C., Sherlock, V., Deutscher, N. M., Liu, X., Feist, D. G., and Wennberg, P. O.: The Total Carbon Column Observing Network's GGG2014 Data Version, https://doi.org/10.14291/TCCON.GGG2014.DOCUMENTATION.R0, 2015.

---

## Author Comment (AC3)

**Comment on acp-2021-307**

**Anonymous Referee #2**

**Referee comment on "Twenty years of ground-based NDACC FTIR spectrometry at Izaña Observatory – overview and long-term comparison to other techniques" by Omaira E. García et al., Atmos. Chem. Phys. Discuss., https://doi.org/10.5194/acp-2021-307-RC3, 2021**

**Response to Referee#2**

The authors would like to thank the Referee for reviewing carefully this paper and provide valuable and constructive comments that have improved this work. In the following text, the Referee suggestions (in blue italics) are addressed in detail (the authors' responses are in plain text).

*General comment*

*Manuscript by Omaira E. García et al. titled "Twenty years of ground-based NDACC FTIR spectrometry at Izaña Observatory - overview and long-term comparison to other techniques" presents a comprehensive analysis of the long-term FTIR-monitoring which is being carried out at Izaña Observatory. Izaña Observatory whose history dates back to 1916, has a strategic location for the investigation of atmospheric processes and contributes to numerous international programmes and observational networks (GAWWMO, WDCGG, WOUDC, NDACC, TCCON, AERONET, BSRN, MPLNET, E-GVAP, NOAA/ESRL/GMD CCGG, etc.).*

*Authors provided a thorough description of the unique FTIR-experiment which was started in 1999. The abstract clearly presents the subject matter and findings of the paper. The scientific basis of the results reported in the paper is the reliable and recognized technique of atmospheric FTIR-spectrometry, and widely used inverse methods for atmospheric sounding (formalism by Rodgers(2000)). Both, the acquisition of MIR spectra of direct solar radiation using FTIR-system installed at Izaña Observatory and the following FTIRspectra processing are described by authors in detail.*

*The investigated time series of C2H6, CH4, ClONO2, CO, HCl, HCN, H2CO, HF, HNO3, N2O, NO2, NO, O3, OCS, and three isotopologues of water vapour (H216O, H218O, and HDO) are of fundamental importance to the atmospheric studies including the interactions of atmospheric composition and climate, the investigation of trace gases temporal variations and processes driven these variations, the verification of modern CTMs (chemical transport models) and the validation of satellite observations. The manuscript is well-written and structured, contains new results that can be of interest to scientific community. Bibliography, in general, provides a relevant list of references, nevertheless, according to referee's opinion, the number of references could be reduced because the bibliography section occupies about one six of the whole paper volume.*

The authors agree with the Referee in that the number of references is excessive, therefore they will be revised and reduced accordingly in the revised manuscript.

*Specific comments*

*1) Lines 131-134: "By evaluating spectral signatures of vibrational-rotational transitions contained in the solar absorption spectra measured, the FTIR technique allows total column amounts and low-resolution vertical profiles of different atmospheric trace gases to be retrieved with a high degree of precision." The "degree of precision" is expected to be different for different trace gases and not necessarily "high" for those retrieved species which have weak absorption signatures in the analyzed FTIR-spectra. If statistical errors/uncertainties can be considered as a measure of "degree of precision", we can see in Table 3 that these errors can reach ~50% for H2CO and ~100% for ClONO2.*

As stated by the Referee, the FTIR technique provides high-quality observations for many trace gases, but its quality indeed depends on the target gas. To avoid confusions, the statement "with a high degree of precision" will be removed in the description of the FTIR technique in the Introduction section.

*2) Section 3.1: Whether FTIR-instruments (an IFS 120M and an IFS 120/5HR) at Izaña Observatory have been operated remotely or by an operator/technician? Could you please specify?*

The IFS 120M instrument (1999-2005) was operated manually by a technician between 1999-2005, while the IFS 120/5HR was also operated manually until 2012 when the instrument was adapted to be controlled remotely. Although this instrument works remotely, it is not an automatic system (i.e. a technician has to run the whole system, albeit not necessarily on site). This information will be added to Section 3.1. and Table 1.

*3) Lines 209-211: "The only quality filter applied on public FTIR products is that observations taken at high solar zenith angles (≥85°) have been excluded to avoid imprecise retrievals (mainly caused by misalignments of the solar tracker or spectroscopic issues). These data represent less than 1% of the total data set." It is expected that clouds are one of the most important factor leading to the outliers in retrieval results. Are the FTIR-observations at Izaña Observatory free of this effect?*

At Izaña Observatory (IZO) the FTIR spectra are only recorded when the line of sight between the instrument and sun is cloud free. Note that, as mentioned above, the FTIR system at IZO is not an automatic system, so the operator decides when the instrument takes measurements. However, to avoid possible contamination of thin clouds, the FTIR observations are, in a second step, filtered according to coincident global solar radiation observations taken at IZO in the framework of the Baseline Solar Radiation Network (BSRN, http://bsrn.awi.de). By using a cloud detection method on the coincident solar radiation measurements (based on Long and Ackerman, 2000, and adapted for IZO by García et al., 2014a), the cloud-free periods in the FTIR records are easily identified. Finally, during the operational analysis, unstable or imprecise FTIR retrievals usually lead to non-convergence of the inversion procedure, which is likely due to remaining thin clouds or other local factors.

*4) Lines 191-193: "Most relevant changes are those related to CH4, for which the spectral micro-windows are adopted from Sepúlveda et al. (2014), and the spectroscopy parameters correspond to the improved linelist provided by Dubravica et al. (2013)." What are the principle differences between CH4 retrieval strategies reported in Sepúlveda et al. (2014) and Sussmann et al. (2011)? Does the modified CH4 retrieval strategy by Sepúlveda et al. (2014) provide the*

*homogeneous results with other IRWG-NDACC sites which make retrievals according to Sussmann et al. (2011)? Please, clarify this.*

*Reference: Sussmann, R., Forster, F., Rettinger, M., and Jones, N.: Strategy for highaccuracy- and-precision retrieval of atmospheric methane from the mid-infrared FTIR network, Atmos. Meas. Tech., 4, 1943–1964, https://doi.org/10.5194/amt-4-1943-2011, 2011.*

The methane ($CH_4$) retrieval strategy used at IZO, described in detail in Sepúlveda et al. (2014), is essentially the same as the one described in Sepúlveda et al. (2012), where the $CH_4$ profile retrievals from mid-infrared FTIR spectra were presented for the relatively-dry high-mountain site of Izaña. Sepúlveda et al. (2014) further broadened that work by proving that this retrieval strategy could be successfully applied on different NDACC sites covering different environments (altitude, latitude, and humidity).

The main differences between the Sepúlveda and Sussmann approaches are related to the spectral micro-windows selected, the methane spectroscopic linelist used, and the treatment of water vapour, as summarised in Table 1 below.

The Sussmann strategy (so-called MIR-GBM v1.0 in Sussmann et al., 2011) uses the 2000 version of HITRAN database (including the 2001 update release), three spectral micro windows between 2613-2922 $cm^{-1}$ (see Table 1), and the water vapour profiles, which are scaled from a climatological profile. As stated in Sussmann et al. (2011), the MIR-GBM v1.0 is the optimum of 24 tested retrieval strategies, covering different spectral micro-window selections and HITRAN version (2000, 2004 and 2008), as it provides the best performance according to the absolute $H_2O$/HDO-$CH_4$ interference error. Note that dominant errors of the non-optimum retrieval strategies were found to be the systematic $H_2O$/HDO-$CH_4$ interference errors leading to a seasonal bias of up to ≈5%.

| $CH_4$ Retrieval Strategy | Sussmann et al. (2011) | Sepúlveda et al. (2014) |
|---|---|---|
| Micro-windows [$cm^{-1}$] | 2613.70–2615.40 2835.50–2835.80 2921.00– 2921.60 | 2611.60-2613.35 2613.70-2615.40 2835.55-2835.80 2903.82-2903.925 2914.70-2915.15 2941.51-2942.22 |
| Spectroscopic database | HITRAN 2000 including 2001 update release | HITRAN 2008 including 2009 update release for $H_2O$, and Dubravica et al. (2013) update for $CH_4$ |
| Water vapour treatment | Scale of climatological profiles | Simultaneous $H_2O$ and HDO profile retrieval |

Table 1. Main differences between the $CH_4$ retrieval strategies presented in Sussmann et al. (2011) and Sepúlveda et al. (2014).

On the other hand, the Sepúlveda work proposes six micro-windows (see Table 1 below), which contain strong, not saturated, and well-isolated $CH_4$ absorption lines as well as $H_2O$ and HDO lines, in order to better account for the $H_2O$ and HDO interferences. The $H_2O$ and HDO profiles are simultaneously retrieved with $CH_4$ using a dedicated profiling retrieval. This approach seeks to minimise the impact of water vapour interferences, which might play a key role for humid low-altitude sites, as documented in Sussmann et al. (2011). Finally, the Sepúlveda strategy uses the HITRAN 2008 spectroscopy database for the forward simulations (with 2009 updates, Rothman et al., 2009), except for the target species $CH_4$, for which they use the improved $CH_4$ line

parameters presented in Dubravica et al. (2013). The latter was found to provide lower spectroscopic residuals than the HITRAN 2008 linelist.

As discussed in detail in Hase et al. (2011), the Sepúlveda method generates $CH_4$ columns in agreement with the Sussman approach for different humidity conditions (subtropical high-mountain Izaña, mid-latitude Karlsruhe, and polar Kiruna sites). However, Hase et al. (2011) also documented that the Sepulveda strategy could be less dependent on humidity conditions as it minimises perturbing $H_2O$/HDO absorptions and it handles the problematic interference species $H2O$/HDO in a rigorous manner. Additionally, Sepúlveda et al. (2012) showed that their approach proved advantageous for reproducing the in-situ annual cycle and yearly mean time series when tropospheric FTIR products were compared to in-situ $CH_4$ records acquired in the framework of GAW-WMO programme at IZO. The difference is presumed to be due to different treatment of water vapour and the use of different line lists for $CH_4$.

This explanation will be introduced briefly in Section 3.3 of the revised manuscript.

*5) Fig.14 (page 37) and Fig.15 (page 41): This is not easy to distinguish between the sizes of dots which correspond to R2=0.5 and R2=0.3.*

The area of dots in Figure 14 and 15 will be modified to make them easier to evaluate.

*6) Maybe, it is worth adding to manuscript a table summarizing all the long-term trends reported and discussed in the text in Sections 5, 6, and 8. Such a table will simplify reading and navigation through the manuscript.*

According to the Referee's suggestion, a new table summarising the trend values discussed in the text will be added to the revised manuscript.

*7) Section 5, Fig.7, and Appendix B: Which methods and/or criteria were implemented for the selection of an optimal set of frequencies used for the construction of multi-regression fit presented in Fig.7? Evaluation of statistical significance, cross-validation, etc.?*

On the one hand, the multi-regression model used accounts for the intra-annual variation (season cycle), for which frequencies up to $2yr^{-1}$ (P=2) have been selected. Numerous works in the literature have proved that seasonal cycle variations can be properly described with this number of frequencies for many trace gases. See, for example, $CH_4$ (Gardiner et al., 2008; Sepúlveda et al., 2012, 2014; García et al., 2018), CO (Gardiner et al., 2008), HCHO (Vigoroux et al., 2018), $N_2O$ (Gardiner et al., 2008; García et al., 2014b), $NO_2$ (Yela et al., 2017), OCS (Hannigan et al., 2021), and $O_3$ (Gardiner et al. 2008; García et al., 2012; Vigouroux et al., 2015).

On the other hand, the inter-annual variations are modelled with a Fourier series that considers all frequencies between 1 and N – 1, where N is the total number of years covered by the whole time series (Sepúlveda et al., 2014). This selection ensures that the fitted curve is able to adapt to relatively fast changes of the measured time series and waves such as those induced by dynamical variations of atmospheric transport circulation (see, for example, the NO, $HNO_3$ or OCS time series in Figure 7 of the preprint).

To ensure that the multi-regression model used properly captures the evolution of measured FTIR observations at the different timescales, the normality of the residuals (differences between the measured data series and the modelled one) was analysed. To do so, the one-sample Kolmogorov-Smirnov test (Kolmogorov, 1933; Smirnov, 1948) was applied to all residual time series, confirming residuals are normally distributed for all the trace gas analysed.

**References**

Dubravica, D., Birk, M., Hase, F., Loos, J., Palm, M., Sadeghi, A., and Wagner, G.: Improved spectroscopic parameters of methane in the MIR for atmospheric remote sensing, in: High Resolution Molecular Spectroscopy 2013 meeting, availableat:http://lmsd.chem.elte.hu/ 1025 hrms/abstracts/D16.pdf, 2013.

García, R. D., García, O. E., Cuevas, E., Cachorro, V. E., RomeroCampos, P. M., Ramos, R., and de Frutos, A. M.: Solar radiation measurements compared to simulations at the BSRN Izaña station, Mineral dust radiative forcing and efficiency study, J. Geophys. Res., 119, 179–194, doi:10.1002/2013JD020301, 2014.

García, O. E., Schneider, M., Redondas, A., González, Y., Hase, F., Blumenstock, T., and Sepúlveda, E.: Investigating the long-term evolution of subtropical ozone profiles applying ground-based FTIR spectrometry, Atmospheric Measurement Techniques, 5, 2917–2931, 1045 https://doi.org/10.5194/amt-5-2917-2012, 2012.

García, O. E., Schneider, M., Hase, F., Blumenstock, T., Sepúlveda, E., Gómez-Peláez, A., Barthlott, S., Dohe, S., González, Y., Meinhardt, F., and Steinbacher, M.: Monitoring of N2O by ground-based FTIR: optimisation of retrieval strategies and comparison to GAW insitu observations, in: NDACC-IRWG/TCCON meeting 2014, Bad Sulza, Germany, 2014b.

García, O. E., Schneider, M., Ertl, B., Sepúlveda, E., Borger, C., Diekmann, C., Wiegele, A., Hase, F., Barthlott, S., Blumenstock, T., Raffalski, U., Gómez-Peláez, A., Steinbacher, M., Ries, L., and de Frutos, A. M.: The MUSICA IASI CH4 and N2O products and their comparison to HIPPO, GAW and NDACC FTIR references, Atmospheric Measurement Techniques, 11, 4171–4215, https://doi.org/10.5194/amt11-4171-2018, 2018.

Gardiner, T., Forbes, A., de Mazière, M., Vigouroux, C., Mahieu, E., Demoulin, P., Velazco, V., Notholt, J., Blumenstock, T., Hase, F., Kramer, I., Sussmann, R., Stremme, W., Mellqvist, J., Strandberg, A., Ellingsen, K., and Gauss, M.: Trend analysis of greenhouse gases over Europe measured by a network of ground-based remote FTIR instruments, Atmos. Chem. Phys., 8, 6719–6727, doi:10.5194/acp-8-6719-2008, 2008

Hase, F., Blumenstock, T., Schneider, M., and Sepúlveda, E. Interactive comment on "Strategy for high-accuracy-and-precision retrieval of atmospheric methane from the mid-infrared FTIR network" by R. Sussmann et al., Atmos. Meas. Tech. Discuss., 4, C1048–C1048, 2011.

Hannigan, J.W., I. Ortega, S. B. Shams, T. Blumenstock, J.E. Campbell, S. A. Conway, V. Flood, O. García, M. Grutter, F. Hase, N.B. Jones, P. Jeseck, E. Mahieu, M. Makarova, M. De Maziere, I. Morino, I. Murata, T. Nagahama, H. Nakajima, J. Notholt, M. Palm, A. Poberovskii, M. Rettinger, J. Robinson, M. Schneider, A. Röhling, C. Servais, D. Smale, W. Stremme, K. Strong, R. Sussmann, Y. Té, C. Vigouroux, T. Wizenberg, Global Atmospheric OCS Trend Analysis from 22 NDACC Stations, submitted to J. Geophys. Res., 2021.

Long, C. and Ackerman, T.: Identification of clear skies from broadband pyranometer measurements and calculation of downwelling shortwave cloud effects, J. Geophys. Res., 105, 15609–15626, doi:10.1029/2000JD900077, 2000.

Kolmogorov, A., Sulla determinazione empirica di una legge di distribuzione, G. Ist. Ital. Attuari. 4: 83–91, 1933

Smirnov, N., Table for estimating the goodness of fit of empirical distributions, Annals of Mathematical Statistics. 19 (2): 279–281. doi:10.1214/aoms/1177730256, 1948.

Sepúlveda, E., Schneider, M., Hase, F., García, O. E., Gómez-Peláez, A., Dohe, S., Blumenstock, T., and Guerra, J. C.: Long-term validation of tropospheric column-averaged CH4

mole fractions obtained by mid-infrared ground-based FTIR spectrometry, Atmospheric Measurement Techniques, 5, 1425–1441, https://doi.org/10.5194/amt-5-1425-2012, 2012.

Sepúlveda, E., Schneider, M., Hase, F., Barthlott, S., Dubravica, D., García, O. E., Gómez-Peláez, A., González, Y., Guerra, J. C., Gisi, M., Kohlhepp, R., Dohe, S., Blumenstock, T., Strong, K., Weaver, D., Palm, M., Sadeghi, A., Deutscher, N. M., Warneke, T., Notholt, J., Jones, N., Griffith, D. W. T., Smale, D., Brailsford, G. W., Robinson, J., Meinhardt, F., Steinbacher, M., Aalto, T., and Worthy, D.: Tropospheric CH4 signals as observed by NDACC FTIR at globally distributed sites and comparison to GAW surface in situ measurements, Atmospheric Measurement Techniques, 7, 2337–2360, https://doi.org/10.5194/amt-7-2337-2014, 2014.

Sussmann, R., Forster, F., Rettinger, M., and Jones, N.: Strategy for highaccuracy-and-precision retrieval of atmospheric methane from the mid-infrared FTIR network, Atmos. Meas. Tech., 4, 1943–1964, https://doi.org/10.5194/amt-4-1943-2011, 2011.

Vigouroux, C., Blumenstock, T., Coffey, M., Errera, Q., García, O., Jones, N. B., Hannigan, J. W., Hase, F., Liley, B., Mahieu, E., Mellqvist, J., Notholt, J., Palm, M., Persson, G., Schneider, M., Servais, C., Smale, D., Thölix, L., and De Mazière, M.: Trends of ozone total columns and vertical distribution from FTIR observations at eight NDACC stations around the globe, Atmospheric Chemistry and Physics, 15, 2915–2933, 2015.

Vigouroux, C., Bauer Aquino, C. A., Bauwens, M., Becker, C., Blumenstock, T., De Mazière, M., García, O., Grutter, M., Guarin, C., Hannigan, J., Hase, F., Jones, N., Kivi, R., Koshelev, D., Langerock, B., Lutsch, E., Makarova, M., Metzger, J.-M., Müller, J.-F., Notholt, J., Ortega, I., Palm, M., Paton-Walsh, C., Poberovskii, A., Rettinger, M., Robinson, J., Smale, D., Stavrakou, T., Stremme, W., Strong, K., Sussmann, R., Té, Y., and Toon, G.: NDACC harmonized formaldehyde time series from 21 FTIR stations covering a wide range of column abundances, Atmospheric Measurement Techniques, 11, 5049–5073, https://doi.org/10.5194/amt-11-5049-2018, 2018.

Yela, M., Gil-Ojeda, M., Navarro-Comas, M., Gonzalez-Bartolomé, D., Puentedura, O., Funke, B., Iglesias, J., Rodríguez, S., García, O., Ochoa, H., and Deferrari, G.: Hemispheric asymmetry in stratospheric NO2 trends, Atmospheric Chemistry and Physics, 17, 13 373– 13 389, https://doi.org/10.5194/acp-17-13373-2017, 2017.